

# Open XXZ chain and boundary modes at zero temperature

**Sebastián Grijalva[1], Jacopo De Nardis[2]\* and Véronique Terras[1]**

**1** LPTMS, UMR 8626, CNRS, Univ. Paris-Sud, Université Paris-Saclay, 91405 Orsay, France
**2** Department of Physics and Astronomy, University of Ghent,
Krijgslaan 281, 9000 Gent, Belgium.

\* jacopo.de.nardis@phys.ens.fr

## Abstract

We study the open XXZ spin chain in the anti-ferromagnetic regime and for generic longitudinal magnetic fields at the two boundaries. We discuss the ground state via the Bethe ansatz and we show that, for a chain of even length $L$ and in a regime where both boundary magnetic fields are equal and bounded by a critical field, the spectrum is gapped and the ground state is doubly degenerate up to exponentially small corrections in $L$. We connect this degeneracy to the presence of a boundary root, namely an excitation localized at one of the two boundaries. We compute the local magnetization at the left edge of the chain and we show that, due to the existence of a boundary root, this depends also on the value of the field at the opposite edge, even in the half-infinite chain limit. Moreover we give an exact expression for the large time limit of the spin autocorrelation at the boundary, which we explicitly compute in terms of the form factor between the two quasi-degenerate ground states. This, as we show, turns out to be equal to the contribution of the boundary root to the local magnetization. We finally discuss the case of chains of odd length.

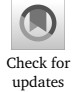

# 1 Introduction and main results

The study of condensed matter theory involves understanding many-body systems starting from their elementary constituents. This protocol, which is in general notoriously hard, can be sometimes carried out in systems of one-dimensional spin chains. These constitute one the main theoretical playgrounds for the emergent physics of strongly correlated quantum systems, see for example the seminal work of Haldane [1]. In particular, in the past years, a class of interacting spin chains which can be exactly solved by the so-called Bethe Ansatz [2,3] have been successfully applied to understand the dynamical response of real compounds [4,5] or to develop better numerical techniques [6].

While the bulk physics of spin chains can be usually studied by considering the large-size limit of systems with periodic boundary conditions, a richer phenomenology can be observed in the presence of open boundaries, as for example in doped spin chains [7–9]. By tuning different parameters at the boundaries one can explore different phase transitions (also experimentally [10]), as well as probing the existence of boundary modes. Notoriously, in topological superconducting systems, the Majorana zero modes [11] are boundary modes and they consist of two decoupled Majorana fermions localized at the two edges of the system, and that can be combined to form a zero-energy regular fermion. As a consequence of their existence, all many-particle states are degenerate. While Majorana zero modes are present in the so-called Kitaev chain, which becomes the XY chain with a transverse field after a Jordan-Wigner transformation, it was recently shown by Fendley [12] that the gapped (massive) XYZ chain contains also *strong zero modes*, namely operators defined at the two edges of the chain that commute with the Hamiltonian up to exponentially small corrections with the size of the chain. These

operators, instead of being exactly localized at the two edges, are characterized by exponential tails that decay away from the edges and are related to the $\mathbb{Z}_2$ symmetry of the model. Their existence also implies an extensive number of degeneracies between the different many-body states in the spectrum.

From the physical point of view it is interesting to study the spin autocorrelation at the edge of the chain. Due to the presence of the aforementioned boundary modes, the latter should not decay to zero even at finite temperature $T$, in the thermodynamic limit $L \to \infty$. Namely, given the Pauli spin operator $\sigma_1^z$ at the left edge of the chain and its time evolution $\sigma_1^z(t) = e^{iHt} \sigma_1^z e^{-iHt}$ with the Hamiltonian $H$ containing a strong zero mode, one should find that, for any temperature $T$,

$$\lim_{t \to \infty} \lim_{L \to \infty} \langle \sigma_1^z(t) \sigma_1^z \rangle_T^c \neq 0, \tag{1.1}$$

where $\langle O_1 O_2 \rangle_T^c = \langle O_1 O_2 \rangle_T - \langle O_1 \rangle_T \langle O_2 \rangle_T$ denotes the connected correlator and $\langle O \rangle_T = \text{Tr}\left(e^{-\beta H} O\right)/\text{Tr}\left(e^{-\beta H}\right)$ the thermal expectation value. This prediction constituted a starting point of an active research field focused on the study of coherence time of edge spins in the open XXZ chain. When the Hamiltonian is perturbed by additional terms that do not preserve the symmetry of the zero mode, namely when the system is perturbed away from the integrable limit, it was shown [13–16] that the dephasing time can still get very large and that the spin autocorrelation remains on a long-living plateau at large intermediate times.

We here consider the open XXZ Hamiltonian with anisotropy parameter $\Delta$ and boundary longitudinal magnetic fields $h_-$ and $h_+$,

$$H = \sum_{j=1}^{L-1} \left[ \sigma_j^x \sigma_{j+1}^x + \sigma_j^y \sigma_{j+1}^y + \Delta \left( \sigma_j^z \sigma_{j+1}^z - 1 \right) \right] + h_- \sigma_1^z + h_+ \sigma_L^z, \tag{1.2}$$

in the massive anti-ferromagnetic regime with $\Delta = \cosh \zeta > 1$ ($\zeta > 0$), which is indeed the regime of existence of the strong zero modes [12]. We particularly focus on the case of a chain with an even number of sites $L$. There are two critical values of the magnetic field at the boundary, $h_{\text{cr}}^{(1)} = \Delta - 1$ and $h_{\text{cr}}^{(2)} = \Delta + 1$, where different crossings between eigenstates occur, see Fig. 3. In the regime where $|h_\pm| < h_{\text{cr}}^{(1)}$, as we shall see, the spectrum is gapped and the ground state is doubly degenerate in the large $L$ limit whenever $h_+ = h_- = h$, so that the zero-temperature spin-spin boundary autocorrelation function is expected to converge for large time to the form factor of the spin operator between these two quasi-degenerate ground states. Namely, by denoting with $\langle \text{GS}_i, h |, i = 1, 2$, the two *normalized* quasi-degenerate ground states of the open chain with boundary magnetic fields $h_- = h_+ = h$, we expect that

$$\lim_{t \to \infty} \lim_{L \to \infty} \lim_{h_- \to h_+ = h} \langle \sigma_1^z(t) \sigma_1^z \rangle_{T=0}^c = \lim_{L \to \infty} |\langle \text{GS}_1, h | \sigma_1^z | \text{GS}_2, h \rangle|^2 \neq 0. \tag{1.3}$$

In this paper, we explicitly compute the thermodynamic and large-time limit (1.3) of the boundary auto-correlation function at zero-temperature from the study of the open chain (1.2) in the algebraic Bethe ansatz (ABA) framework [17]. By considering the large $L$ limit of the solutions of the Bethe equations and controlling the finite-size corrections up to exponentially small order in $L$, we show that the difference of energy between the ground state and the first excited state becomes exponentially small in $L$ when $h_+ = h_-$ ( $|h_\pm| < h_{\text{cr}}^{(1)}$). Each of these two states is characterized by a Fermi sea of $\frac{L}{2} - 1$ real Bethe roots and an isolated complex Bethe root which corresponds to a boundary mode and that we call *boundary root*. The latter is localized, up to exponentially small corrections in $L$, at the zero of one of the two boundary factors appearing in the Bethe equations, and represents a collective magnonic excitation pinned at one of the two edges of the chain, whose wave function has exponentially decreasing tails away from the boundary [18, 19]. We show that this boundary mode is responsible for the ground state degeneracy, which in particular has two main physical consequences:

1. The boundary magnetization in the ground state for even size $L$ depends on the value of *both* boundary fields, even in the infinite chain limit $L \to \infty$ (thermodynamic limit). This is due to the fact that the presence of the boundary root in the Bethe solution for the ground state and its localization at one of the two (zeroes corresponding to one of the two) edges of the chain depends on the values of both boundary fields. Moreover, when one of the fields is inside the interval $(-h_{\mathrm{cr}}^{(1)}, h_{\mathrm{cr}}^{(1)})$, the boundary magnetization becomes a discontinuous function of the other field at $h_- = h_+$, point at which the localization of the ground state boundary root changes from one edge of the chain to the other. We here provide an analytical derivation for the boundary magnetization at the left edge of the chain, and notably for the value of its discontinuity at $h_- = h_+ = h$ ($|h| < h_{\mathrm{cr}}^{(1)}$). The latter is given by $\langle \sigma_1^z \rangle_{\mathrm{BR}}$, the (thermodynamic limit of the) contribution to the boundary magnetization carried by the boundary root at the left edge, which is non-zero only when $h_- \geq h_+$:

$$\lim_{\substack{h_- \to h_+ \\ h_- < h_+}} \lim_{L \to \infty} \langle \sigma_1^z \rangle - \lim_{\substack{h_- \to h_+ \\ h_- > h_+}} \lim_{L \to \infty} \langle \sigma_1^z \rangle = -\lim_{\substack{h_- \to h_+ \\ h_- > h_+}} \langle \sigma_1^z \rangle_{\mathrm{BR}} = -2 \langle \sigma_1^z \rangle_{\mathrm{BR}} \Big|_{h_- = h_+}, \qquad (1.4)$$

see eq. (5.16) and (5.17) for an exact expression in terms of the parameters of the model. At exactly $h_- = h_+$, the boundary root becomes delocalized between the two edges of the chain and contributes equally to the left or the right boundary magnetization, hence the factor 2 in (1.4). In the particular case $h_+ = 0$, we recover that [20, 21]

$$\lim_{h_- \to 0^\pm} \lim_{h_+ \to 0} \lim_{L \to \infty} \langle \sigma_1^z \rangle = \mp s_0^2, \qquad (1.5)$$

where $s_0 = \prod_{n=1}^\infty \left( \frac{1 - e^{-2n\zeta}}{1 + e^{-2n\zeta}} \right)^2$ is the bulk magnetization [22].

2. The degeneracy of the ground state at $h_- = h_+ = h$ ($|h| < h_{\mathrm{cr}}^{(1)}$) implies that the infinite time limit of the boundary spin-spin autocorrelation function in the thermodynamic limit is given by the square of the norm of the matrix element of the spin operator $\sigma_1^z$ in the first site of the chain between the two quasi-degenerate ground states $\langle \mathrm{GS}_1, h |$ and $| \mathrm{GS}_2, h \rangle$, see equation (1.3). We here exactly compute this matrix element in the ABA framework and explain how to derive its thermodynamic limit $L \to \infty$. We show that, when $h_- = h_+$, it is directly related to the contribution to the boundary magnetization carried by the boundary root at the left edge as

$$\lim_{L \to \infty} \langle \mathrm{GS}_1, h | \sigma_1^z | \mathrm{GS}_2, h \rangle = -\langle \sigma_1^z \rangle_{\mathrm{BR}} \Big|_{h_- = h_+ = h}, \qquad (1.6)$$

for any $|h| < h_{\mathrm{cr}}^{(1)}$, so that it is given by half of the boundary magnetization discontinuity (1.4) at $h_+ = h_-$. For $h_+ = h_- = h$ and $|h| < h_{\mathrm{cr}}^{(1)}$ the quantity $\langle \sigma_1^z \rangle_{\mathrm{BR}}$, and so the matrix element (1.6), is non-zero, see (5.48). When both fields are zero ($h = 0$) this reduces to the value (1.5):

$$\lim_{L \to \infty} \langle \mathrm{GS}_1, 0 | \sigma_1^z | \mathrm{GS}_2, 0 \rangle = s_0^2. \qquad (1.7)$$

Instead, as soon as $h_+ \neq h_-$, the matrix element of $\sigma_1^z$ between the two states of lowest energy decreases exponentially fast with the size $L$ of the chain, so that the thermodynamic limit of the boundary autocorrelation function vanishes in the large time limit:

$$\lim_{t \to \infty} \lim_{L \to \infty} \langle \sigma_1^z(t) \sigma_1^z \rangle_{\substack{T=0 \\ h_+ \neq h_-}}^c = 0. \qquad (1.8)$$

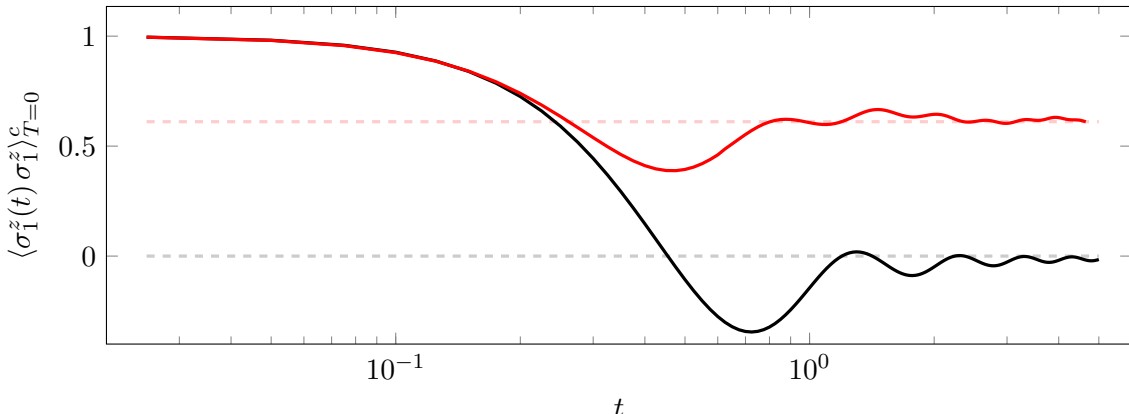

Figure 1: Real part of the spin autocorrelation function $\langle \sigma_1^z(t)\sigma_1^z \rangle_{T=0}^c$ vs. time $t$, obtained by tDMRG. *In red*: $\Delta = 3$. *In black*: $\Delta = 1$. System size $L = 32$, boundary fields $h_+ = h_- = 0$. Corresponding exact thermodynamic limit values, eq. (1.7) are shown with dashed lines.

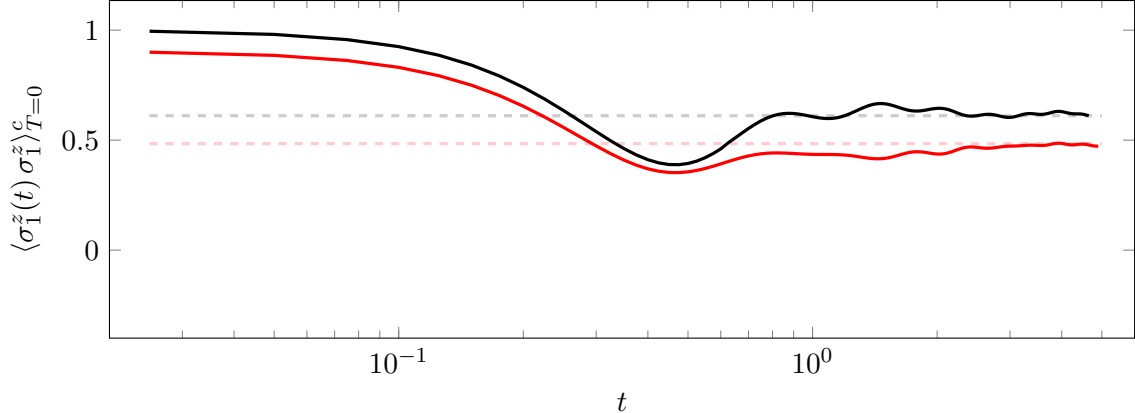

Figure 2: Real part of the spin autocorrelation function $\langle \sigma_1^z(t)\sigma_1^z \rangle_{T=0}^c$ vs. time $t$ obtained by tDMRG. *In black*: $h_\pm = 0$. *In red*: $h_\pm = 1$. System size $L = 32$, anisotropy $\Delta = 3$. Corresponding exact thermodynamic limit values from eq. (1.6) and eq. (1.7) are shown dashed.

In Fig. 1 and Fig. 2 the boundary spin autocorrelation function $\langle \sigma_1^z(t)\sigma_1^z \rangle_{T=0}^c$ is computed numerically by tDMRG as a function of time for a chain with finite even size $L$ at $h_+ = h_-$: at large times the correlation attains the value given by (1.3)-(1.6).

It is interesting to compare these effects to what happens in open chains of odd length: there is in this case an exact $\mathbb{Z}_2$ degeneracy of the spectrum at $h_- = -h_+$, and the discontinuity of the boundary magnetization at this point is simply due to a crossing of levels between the two states of lowest energy which belong to different magnetization sectors. Note that the description of the ground state of the odd $L$ chain in the regime $|h_\pm| < h_{\rm cr}^{(1)}$ and for $h_- + h_+ < 0$ only involves real Bethe roots (see section 6).

This article is organized as follows. In section 2, we recall the diagonalization of the Hamiltonian (1.2) in the framework of the boundary algebraic Bethe ansatz introduced by Sklyanin in [17]. In section 3, we explain how to derive, in this framework, compact determinant representations for the finite-size matrix elements (form factors) of the $\sigma_1^z$ operator between two Bethe eigenstates. In section 4 we study the solutions of the Bethe equations in the thermodynamic limit $L \to +\infty$ and explain how to control their finite-size corrections up to exponen-

tially small order in $L$. We more precisely consider the case of a chain of even length $L$, and we identify the solution corresponding to the ground state for the different values of the boundary magnetic fields $h_+$ and $h_-$. In the regime where both fields are between $-h_{cr}^{(1)}$ and $h_{cr}^{(1)}$, with $h_{cr}^{(1)} = \Delta - 1$, we show that the two states of lowest energy are given by a particular solution of the Bethe equations with $\frac{L}{2} - 1$ real Bethe roots and one complex Bethe root which has to be chosen between the two possible *boundary roots* given in terms of the boundary parameter at the left or the right end of the chain. We moreover show that, when $h_+ = h_-$, the deviation between the two boundary roots becomes exponentially small in $L$, and so does the difference of energy between the two corresponding states. In section 5, we compute the thermodynamic limit of the determinant representation that we obtained in section 3 in two particular cases: the mean value of $\sigma_1^z$ in the ground state, which gives the boundary magnetization, and the $\sigma_1^z$ form factor between the two states of lowest energy identified in section 4, which gives the infinite time limit of the boundary autocorrelation function. Finally, in section 6, we consider the case of a chain with an odd number of sites $L$, and explain how our computations should be modified in this case.

## 2 The integrable open XXZ spin chain

The Hamiltonian (1.2) is integrable [23] and can be diagonalized in the framework of the representation theory of the reflection algebra [24], by means of the boundary version of algebraic Bethe ansatz introduced by Sklyanin in [17].

The key object in this approach is the boundary monodromy matrix $\mathcal{U}(\lambda) \in \text{End}(\mathbb{C}^2 \otimes \mathcal{H})$ where $\mathcal{H}$ is the space of states of the system. It is such that $\mathcal{V}(\lambda) \equiv \mathcal{U}^t(-\lambda)$ satisfies the reflection equation[1],

$$R_{12}(\lambda - \mu)\mathcal{V}_1(\lambda)R_{12}(\lambda + \mu + i\zeta)\mathcal{V}_2(\mu) = \mathcal{V}_2(\mu)R_{12}(\lambda + \mu + i\zeta)\mathcal{V}_1(\lambda)R_{12}(\lambda - \mu), \quad (2.1)$$

where $R \in \text{End}(\mathbb{C}^2 \otimes \mathbb{C}^2)$ is the 6-vertex trigonometric $R$-matrix,

$$R_{12}(\lambda) = \begin{pmatrix} \sin(\lambda - i\zeta) & 0 & 0 & 0 \\ 0 & \sin(\lambda) & \sin(-i\zeta) & 0 \\ 0 & \sin(-i\zeta) & \sin(\lambda) & 0 \\ 0 & 0 & 0 & \sin(\lambda - i\zeta) \end{pmatrix}. \quad (2.2)$$

The relation (2.1) has to be understood $\mathbb{C}^2 \otimes \mathbb{C}^2 \otimes \mathcal{H}$, and the subscripts parameterize the subspaces of $\mathbb{C}^2 \otimes \mathbb{C}^2$ on which the corresponding operators act non-trivially. The parameter $\zeta$ is related to the anisotropy parameter $\Delta$ of (1.2) as $\Delta = \cosh \zeta$.

In the case of the spin chain (1.2) with longitudinal boundary fields, the boundary monodromy matrix solution of (2.1) can be constructed from the bulk monodromy matrix $T(\lambda)$ and a diagonal scalar solution of the reflection equation (2.1),

$$K(\lambda; \xi) = \begin{pmatrix} \sin(\lambda + i\zeta/2 + i\xi) & 0 \\ 0 & \sin(i\xi - \lambda - i\zeta/2) \end{pmatrix}. \quad (2.3)$$

More precisely, we introduce two such boundary scalar matrices,

$$K_-(\lambda) = K(\lambda; \xi_-), \qquad K_+(\lambda) = K(\lambda - i\zeta; \xi_+), \quad (2.4)$$

where $\xi_\pm$ are some complex parameters which parameterize the left and right boundary fields $h_\pm$ as $h_\pm = -\sinh \zeta \coth \xi_\pm$. The boundary monodromy matrix $\mathcal{U}(\lambda)$ is then constructed as

$$\mathcal{U}^t(\lambda) = T^t(\lambda)K_+^t(\lambda)\widehat{T}^t(\lambda) = \begin{pmatrix} \mathcal{A}(\lambda) & \mathcal{C}(\lambda) \\ \mathcal{B}(\lambda) & \mathcal{D}(\lambda) \end{pmatrix}, \quad (2.5)$$

---

[1]The monodromy matrix $\mathcal{U}(\lambda)$ that we consider here corresponds to the matrix $U_+(\lambda)$ of [17].

where the bulk monodromy matrix $T(\lambda)$ is itself constructed as a product of $R$-matrices (2.2) as

$$T(\lambda) \equiv T_a(\lambda) = R_{aL}(\lambda - \xi_L) \ldots R_{a1}(\lambda - \xi_1), \tag{2.6}$$

$$\widehat{T}(\lambda) = (-1)^L \sigma^y T^t(-\lambda) \sigma^y. \tag{2.7}$$

Here the index $a$ denotes the so-called auxilliary space $V_a \simeq \mathbb{C}^2$, and $\xi_1, \ldots, \xi_L$ are a set of inhomogeneity parameters which may be introduced for technical convenience.

One then define a one-parameter family of commuting transfer matrices as

$$\mathcal{T}(\lambda) = \mathrm{tr}\left\{ K_+(\lambda) T(\lambda) K_-(\lambda) \widehat{T}(\lambda) \right\} = \mathrm{tr}\left\{ K_-(\lambda) \mathcal{U}(\lambda) \right\}. \tag{2.8}$$

In the homogeneous limit in which $\xi_\ell = -i\zeta/2$, $\ell = 1, \ldots, L$, the Hamiltonian (1.2) of the spin-1/2 open chain can be obtained as

$$H = \frac{-i \sinh\zeta}{\mathcal{T}(\lambda)} \frac{d}{d\lambda} \mathcal{T}(\lambda)\Big|_{\lambda=-i\zeta/2} + \frac{1}{\cosh\zeta} - 2L \cosh\zeta. \tag{2.9}$$

In the algebraic Bethe ansatz framework, the common eigenstates of the transfer matrices can be constructed in the form

$$|\{\lambda\}\rangle = \prod_{j=1}^N \mathcal{B}(\lambda_j)|0\rangle, \qquad \langle\{\lambda\}| = \langle 0|\prod_{j=1}^N \mathcal{C}(\lambda_j), \tag{2.10}$$

where $|0\rangle$ (respectively $\langle 0|$) is the reference state (respectively the dual reference state) with all spins up. By using the commutation relations issued from (2.1), it can be shown that states of the form (2.10) are eigenstates of the transfer matrix (2.8) provided the set of spectral parameters $\{\lambda\} \equiv \{\lambda_1, \ldots, \lambda_N\}$ satisfies the system of Bethe equations

$$\mathbf{A}(\lambda_j) \prod_{k=1}^N \mathfrak{s}(\lambda_j + i\zeta, \lambda_k) + \mathbf{A}(-\lambda_j) \prod_{k=1}^N \mathfrak{s}(\lambda_j - i\zeta, \lambda_k) = 0, \quad j = 1, \ldots, N, \tag{2.11}$$

where

$$\mathbf{A}(\mu) = (-1)^L \frac{\sin(2\mu - i\zeta)}{\sin(2\mu)} \mathbf{a}(\mu), \tag{2.12}$$

$$\mathbf{a}(\mu) = (-1)^L a(\mu) d(-\mu) \sin(\mu + i\xi_+ + i\zeta/2) \sin(\mu + i\xi_- + i\zeta/2), \tag{2.13}$$

with

$$a(\mu) = \prod_{\ell=1}^L \sin(\mu - \xi_\ell - i\zeta), \qquad d(\mu) = \prod_{\ell=1}^L \sin(\mu - \xi_\ell). \tag{2.14}$$

Here and in the following, we use the shortcut notations:

$$\mathfrak{s}(\lambda, \mu) = \sin(\lambda + \mu) \sin(\lambda - \mu) = \sin^2\lambda - \sin^2\mu. \tag{2.15}$$

The corresponding transfer matrix eigenvalue is

$$\tau(\mu, \{\lambda\}) = (-1)^L \left[ \mathbf{A}(\mu) \prod_{i=1}^N \frac{\mathfrak{s}(\mu + i\zeta, \lambda_i)}{\mathfrak{s}(\mu, \lambda_i)} + \mathbf{A}(-\mu) \prod_{i=1}^N \frac{\mathfrak{s}(\mu - i\zeta, \lambda_i)}{\mathfrak{s}(\mu, \lambda_i)} \right]. \tag{2.16}$$

From (2.9), in the homogeneous limit in which $\xi_\ell = -i\zeta/2$, $\ell = 1, \ldots, L$ the transfer matrix eigenstates (2.10) become eigenstates of the Hamiltonian (1.2) with energy

$$E(\{\lambda\}) = h_+ + h_- + \sum_{j=1}^N \varepsilon_0(\lambda_j), \tag{2.17}$$

where the bare energy $\varepsilon_0(\lambda)$ is defined as

$$\varepsilon_0(\lambda) = -\frac{2\sinh^2\zeta}{\sin(\lambda - i\zeta/2)\sin(\lambda + i\zeta/2)} = -\frac{4\sinh^2\zeta}{\cosh\zeta - \cos(2\lambda_j)}. \tag{2.18}$$

Eigenstates of the form (2.10) are called on-shell Bethe states. States of the form (2.10) for which the parameters $\{\lambda\}$ do not satisfy the Bethe equations are instead called off-shell Bethe states. The study of the solutions of Bethe equations, and in particular of the ground state of the Hamiltonian (1.2) in the thermodynamic limit, has been performed in [18,19,23].

Building on this ABA description of the spectrum and eigenstates, it is possible to compute the zero-temperature correlation functions of the open spin chain [21,25]. However, this program has not yet reached the level of achievement as what has been done for the bulk correlation functions [26–37]. In the latter case, it was indeed possible to derive the large distance and long time asymptotic behavior of the two-point (or even multi-point) correlation functions in the thermodynamic limit from their exact representations on the lattice. At the root of this approach was the fact that there exist some compact and simple determinant formulas for the form factors of local operators in the finite periodic chain [26]. Such determinant representations were also of uttermost importance for the numerical studies of the correlation functions [38–41]. They were obtained thanks to two main ingredients: a determinant representation of the scalar product of an off-shell and an on-shell Bethe states [42], and the fact that the local spin operators could be expressed as a simple element of the monodromy matrix dressed by a product of transfer matrices (solution of the quantum inverse problem) [26,43,44].

In the open case, however, such nice determinant representations for the form factors do not exist in general. It is still possible to express the scalar product of an off-shell and an on-shell Bethe states of the form (2.10) as a generalized version of the Slavnov determinant [21,45], but a convenient expression of the local spin operators in terms of the boundary monodromy matrix elements dressed by a product of *boundary* transfer matrices is presently not known, except at the first (or last) site of the chain [46]. In fact, the formulas obtained in [21,25] relied on a cumbersome use of the *bulk* inverse problem, which resulted into multiple integral formulas for the zero-temperature correlation functions in the thermodynamic limit (half-infinite chain) similar to the one that were previously obtained in [20] from a different approach.

At the first (or last) site of the chain, however, the situation is different. Indeed, the solution of the quantum inverse problem proposed in [46] is in that case sufficient, together with the determinant representation for the scalar products, to obtain determinant representations for the form factors of local operators at site 1 which are very similar to the bulk ones. Hence, we may expect to be able to study their thermodynamic limit similarly as what has been done in [30,31,36,47]. In particular, we are in position to compute and study the thermodynamic limit of the form factors which are relevant for the long-time limit of the boundary autocorrelation (1.1). This is the purpose of the next sections.

## 3 The $\sigma_1^z$ form factor in the finite-size open chain

The finite-size form factor of local spin operators on the first site of the chain can be computed similarly as in the periodic case [26], by using the solution of the quantum inverse problem on the first site of the chain [46] together with the determinant representation for the scalar product of an on-shell $\langle\{\lambda\}|$ with an off-shell $|\{\mu\}\rangle$ Bethe states (2.10). For $\{\lambda\} \equiv \{\lambda_1, \ldots, \lambda_N\}$ a solution of the Bethe equations and $\{\mu\} \equiv \{\mu_1, \ldots, \mu_N\}$ and arbitrary set of parameters, the

latter is given by [21, 45]

$$\langle\{\lambda\}|\{\mu\}\rangle = \prod_{j=1}^{N}\left[a(\lambda_j)d(-\lambda_j)\frac{\sin(2\lambda_j-i\zeta)\sin(2\mu_j-i\zeta)}{\sin(2\mu_j)}\frac{\sin(\lambda_j+i\xi_++i\frac{\zeta}{2})}{\sin(\lambda_j-i\xi_--i\frac{\zeta}{2})}\right]$$
$$\times(-1)^{NL}\prod_{j<k}\left[\frac{\sin(\lambda_j+\lambda_k-i\zeta)}{\sin(\lambda_j+\lambda_k+i\zeta)}\frac{1}{\mathfrak{s}(\lambda_j,\lambda_k)\mathfrak{s}(\mu_k,\mu_j)}\right]\det_N\left[H(\boldsymbol{\lambda},\boldsymbol{\mu})\right], \quad (3.1)$$

where the elements of the $N\times N$ matrix $H(\boldsymbol{\lambda},\boldsymbol{\mu})$ are

$$\left[H(\boldsymbol{\lambda},\boldsymbol{\mu})\right]_{jk}=\frac{\sin(-i\zeta)}{\mathfrak{s}(\mu_k,\lambda_j)}\left[\mathbf{a}(\mu_k)\prod_{\ell\neq j}\mathfrak{s}(\mu_k+i\zeta,\lambda_\ell)-\mathbf{a}(-\mu_k)\prod_{\ell\neq j}\mathfrak{s}(\mu_k-i\zeta,\lambda_\ell)\right], \quad (3.2)$$

for $\boldsymbol{\lambda}\equiv(\lambda_1,\dots,\lambda_N)$ and $\boldsymbol{\mu}\equiv(\mu_1,\dots\mu_N)$. The reconstruction of the $\sigma_1^z$ operator in terms of the boundary monodromy matrix elements reads [46]

$$\sigma_1^z=[\sin(i\xi_-+\xi_1+i\zeta/2)\mathcal{A}(\xi_1)-\sin(i\xi_--\xi_1-i\zeta/2)\mathcal{D}(\xi_1)]\mathcal{T}(\xi_1)^{-1} \quad (3.3)$$
$$=2\sin(i\xi_-+\xi_1+i\zeta/2)\mathcal{A}(\xi_1)\mathcal{T}(\xi_1)^{-1}-1, \quad (3.4)$$

where $\xi_1$ is a generic inhomogeneity parameter that should be sent to $-i\zeta/2$ at the end of the computation. We also recall the action of the boundary monodromy matrix element $\mathcal{A}(\xi_1)$ on an off-Bethe state (2.10), which follows from the commutations relations issued from (2.1):

$$\mathcal{A}(\xi_1)\prod_{j=1}^{N}\mathcal{B}(\mu_j)|0\rangle=\Omega(\xi_1|\{\mu\})\prod_{j=1}^{N}\mathcal{B}(\mu_j)|0\rangle+\sum_{j=1}^{N}\Omega_j(\xi_1|\boldsymbol{\mu})\mathcal{B}(\xi_1)\prod_{\substack{k=1\\k\neq j}}^{N}\mathcal{B}(\mu_j)|0\rangle, \quad (3.5)$$

with

$$\Omega(\xi_1|\{\mu\})=\frac{2\,\tau(\xi_1|\{\mu\})}{\sin(i\xi_-+\xi_1+i\zeta/2)}, \quad (3.6)$$

$$\Omega_j(\lambda|\boldsymbol{\mu})=\frac{\sin(-i\zeta)\sin(2\mu_j-i\zeta)}{\mathfrak{s}(\lambda,\mu_j)\sin(2\mu_j)}\left[\frac{\mathbf{a}(\mu_j)\sin(\lambda+\mu_j+i\zeta)}{\sin(\mu_j+i\xi_-+i\zeta/2)}\prod_{\substack{k=1\\k\neq j}}^{N}\frac{\mathfrak{s}(\mu_j+i\zeta,\mu_k)}{\mathfrak{s}(\mu_j,\mu_k)}\right.$$

$$\left.+\frac{\mathbf{a}(-\mu_j)\sin(\lambda-\mu_j+i\zeta)}{\sin(\mu_j-i\xi_--i\zeta/2)}\prod_{\substack{k=1\\k\neq j}}^{N}\frac{\mathfrak{s}(\mu_j-i\zeta,\mu_k)}{\mathfrak{s}(\mu_j,\mu_k)}\right]. \quad (3.7)$$

It follows from (3.4), (3.5) and (3.1) that the matrix element of the $\sigma_1^z$ operator between two eigenstates $\langle\{\lambda\}|$ and $|\{\mu\}\rangle$ is

$$\langle\{\lambda\}|\sigma_1^z|\{\mu\}\rangle=\frac{2\sin(i\xi_-+\xi_1+i\zeta/2)}{\tau(\xi_1|\{\mu\})}\langle\{\lambda\}|\mathcal{A}(\xi_1)|\{\mu\}\rangle-\langle\{\lambda\}|\{\mu\}\rangle$$

$$=2\sum_{j=1}^{N}\frac{\Omega_j(\xi_1|\boldsymbol{\mu})}{\Omega(\xi_1|\{\mu\})}\langle\{\lambda\}|\{\mu_k\}_{k\neq j}\cup\{\xi_1\}\rangle+\langle\{\lambda\}|\{\mu\}\rangle$$

$$=\prod_{j=1}^{N}\left[(-1)^L a(\lambda_j)d(-\lambda_j)\frac{\sin(2\lambda_j-i\zeta)\sin(2\mu_j-i\zeta)}{\sin(2\mu_j)}\frac{\sin(\lambda_j+i\xi_++i\frac{\zeta}{2})}{\sin(\lambda_j-i\xi_--i\frac{\zeta}{2})}\right]$$

$$\times\prod_{j=1}^{N}\frac{\mathfrak{s}(\lambda_j,\xi_1+i\zeta)}{\mathfrak{s}(\mu_j,\xi_1+i\zeta)}\prod_{j<k}\left[\frac{\sin(\lambda_j+\lambda_k-i\zeta)}{\sin(\lambda_j+\lambda_k+i\zeta)}\frac{1}{\mathfrak{s}(\lambda_j,\lambda_k)\mathfrak{s}(\mu_k,\mu_j)}\right]$$

$$\times\det_N\left[H(\boldsymbol{\lambda},\boldsymbol{\mu})-2P(\boldsymbol{\lambda},\boldsymbol{\mu})\right], \quad (3.8)$$

where $H(\boldsymbol{\lambda}, \boldsymbol{\mu})$ is the matrix (3.2) and $P(\boldsymbol{\lambda}, \boldsymbol{\mu})$ is a rank one matrix with elements

$$
\left[P(\boldsymbol{\lambda}, \boldsymbol{\mu})\right]_{jk} = \mathbf{a}(-\mu_k) \prod_{\ell \neq k} \mathfrak{s}(\mu_k - i\zeta, \mu_\ell) \left[ \frac{\sin(\mu_k - \xi_1 - i\zeta)}{\sin(\mu_k - i\xi_- - i\frac{\zeta}{2})} - \frac{\sin(\mu_k + \xi_1 + i\zeta)}{\sin(\mu_k + i\xi_- + i\frac{\zeta}{2})} \right]
$$
$$
\times \sin(\xi_1 + i\xi_- + i\zeta/2) \frac{\sin^2(-i\zeta)}{\mathfrak{s}(\xi_1 + i\zeta, \lambda_j) \mathfrak{s}(\xi_1, \lambda_j)}. \quad (3.9)
$$

So as to express the determinant in a more convenient form for taking the thermodynamic limit, let us introduce, as in [47], an $N \times N$ matrix $\mathcal{X}$ with elements

$$
\mathcal{X}_{ij} = \frac{1}{\mathfrak{s}(\mu_i, \lambda_j)} \frac{\prod_{\ell=1}^{N} \mathfrak{s}(\lambda_j, \mu_\ell)}{\prod_{\ell \neq j} \mathfrak{s}(\lambda_j, \lambda_\ell)}. \quad (3.10)
$$

Its determinant is

$$
\det \mathcal{X} = (-1)^N \prod_{j > k} \frac{\mathfrak{s}(\mu_k, \mu_j)}{\mathfrak{s}(\lambda_k, \lambda_j)}. \quad (3.11)
$$

Multiplying and dividing (3.8) by $\det \mathcal{X}$, computing the matrices $\mathcal{X}H$ and $\mathcal{X}P$, and factorizing the quantity

$$
i^N \prod_{k=1}^{N} \frac{\mathbf{a}(-\mu_k) \prod_{\ell=1}^{N} \mathfrak{s}(\mu_k - i\zeta, \mu_\ell)}{\sin(2\mu_k) \sin(2\mu_k - i\zeta)}, \quad (3.12)
$$

outside of the determinant, we obtain

$$
\langle \{\lambda\} | \sigma_1^z | \{\mu\} \rangle = \prod_{j=1}^{N} \left[ (-1)^L a(\lambda_j) d(-\lambda_j) \sin(2\lambda_j - i\zeta) \frac{\sin(\lambda_j + i\xi_+ + i\frac{\zeta}{2})}{\sin(\lambda_j - i\xi_- - i\frac{\zeta}{2})} \right]
$$
$$
\times \prod_{j < k} \frac{\sin(\lambda_j + \lambda_k - i\zeta)}{\sin(\lambda_j + \lambda_k + i\zeta)} \prod_{k=1}^{N} \frac{\mathbf{a}(-\mu_k) \prod_{\ell=1}^{N} \mathfrak{s}(\mu_k - i\zeta, \mu_\ell)}{i \sin^2(2\mu_k) \prod_{\ell \neq k} \mathfrak{s}(\mu_k, \mu_\ell)}
$$
$$
\times \det_N \left[ \mathcal{M}(\boldsymbol{\lambda}, \boldsymbol{\mu}) - 2\mathcal{P}(\boldsymbol{\lambda}, \boldsymbol{\mu}) \right], \quad (3.13)
$$

with

$$
\left[\mathcal{M}(\boldsymbol{\lambda}, \boldsymbol{\mu})\right]_{jk} = i\delta_{jk} \sin(2\mu_j) \frac{\prod_{\ell \neq j} \mathfrak{s}(\mu_j, \mu_\ell)}{\prod_{\ell=1}^{N} \mathfrak{s}(\mu_j, \lambda_\ell)} \prod_{\ell=1}^{N} \frac{\mathfrak{s}(\mu_j - i\zeta, \lambda_\ell)}{\mathfrak{s}(\mu_j - i\zeta, \mu_\ell)} \left[ \mathfrak{a}(\mu_j | \{\lambda\}) - 1 \right]
$$
$$
- i \sin(2\mu_j) \left[ \frac{\mathfrak{a}(\mu_k | \{\mu\})}{\mathfrak{s}(\mu_k - i\zeta, \mu_j)} - \frac{1}{\mathfrak{s}(\mu_k + i\zeta, \mu_j)} \right], \quad (3.14)
$$
$$
\left[\mathcal{P}(\boldsymbol{\lambda}, \boldsymbol{\mu})\right]_{jk} = -i \sin(\xi_1 + i\xi_+ + i\zeta/2) \left[ \frac{\sin(\mu_k - \xi_1 - i\zeta)}{\sin(\mu_k - i\xi_- - i\frac{\zeta}{2})} - \frac{\sin(\mu_k + \xi_1 + i\zeta)}{\sin(\mu_k + i\xi_- + i\frac{\zeta}{2})} \right]
$$
$$
\times \frac{\sin(2\mu_j)}{\sin(2\xi_1 + i\zeta)} \left[ \frac{\prod_{\ell \neq j} \mathfrak{s}(\xi_1, \mu_\ell)}{\prod_{\ell=1}^{N} \mathfrak{s}(\xi_1, \lambda_\ell)} - \frac{\prod_{\ell \neq j} \mathfrak{s}(\xi_1 + i\zeta, \mu_\ell)}{\prod_{\ell=1}^{N} \mathfrak{s}(\xi_1 + i\zeta, \lambda_\ell)} \right], \quad (3.15)
$$

in which we have defined

$$
\mathfrak{a}(\mu | \{\nu\}) = \frac{\mathbf{a}(\mu)}{\mathbf{a}(-\mu)} \frac{\sin(i\zeta - 2\mu)}{\sin(i\zeta + 2\mu)} \prod_{\ell=1}^{N} \frac{\mathfrak{s}(\mu + i\zeta, \nu_\ell)}{\mathfrak{s}(\mu - i\zeta, \nu_\ell)}. \quad (3.16)
$$

Using the Bethe equations for $\{\mu\}$ and taking the limit $\xi_1 \to -i\zeta/2$, we can rewrite (3.14) and (3.15) as

$$
\big[\mathcal{M}(\boldsymbol{\lambda},\boldsymbol{\mu})\big]_{jk} = i\,\delta_{jk}\,\sin(2\mu_j)\frac{\prod_{\ell\neq j}\mathfrak{s}(\mu_j,\mu_\ell)}{\prod_{\ell=1}^{N}\mathfrak{s}(\mu_j,\lambda_\ell)}\prod_{\ell=1}^{N}\frac{\mathfrak{s}(\mu_j-i\zeta,\lambda_\ell)}{\mathfrak{s}(\mu_j-i\zeta,\mu_\ell)}\big[\mathfrak{a}(\mu_j|\{\lambda\})-1\big]
$$
$$
-2\pi\big[K(\mu_j-\mu_k)-K(\mu_j+\mu_k)\big], \tag{3.17}
$$

$$
\big[\mathcal{P}(\boldsymbol{\lambda},\boldsymbol{\mu})\big]_{jk} = -i\sinh\xi_-\left[\frac{\sin(\mu_k-i\frac{\zeta}{2})}{\sin(\mu_k-i\xi_--i\frac{\zeta}{2})}-\frac{\sin(\mu_k+i\frac{\zeta}{2})}{\sin(\mu_k+i\xi_-+i\frac{\zeta}{2})}\right]
$$
$$
\times\frac{\sin(2\mu_j)}{\mathfrak{s}(\mu_j,i\frac{\zeta}{2})}\prod_{\ell=1}^{N}\frac{\mathfrak{s}(\mu_\ell,i\frac{\zeta}{2})}{\mathfrak{s}(\lambda_\ell,i\frac{\zeta}{2})}\left[\sum_{\ell=1}^{N}\big[p'(\mu_\ell)-p'(\lambda_\ell)\big]-p'(\mu_j)\right], \tag{3.18}
$$

in which we have set

$$
K(\lambda) = \frac{\sinh(2\zeta)}{2\pi\,\sin(\lambda+i\zeta)\,\sin(\lambda-i\zeta)}, \tag{3.19}
$$

$$
p'(\lambda) = \frac{\sinh\zeta}{\sin(\lambda+i\frac{\zeta}{2})\,\sin(\lambda-i\frac{\zeta}{2})}. \tag{3.20}
$$

It remains to take into account the normalization of a Bethe state, which is given by the formula

$$
\langle\{\lambda\}|\{\lambda\}\rangle = \prod_{j=1}^{N}\left[(-1)^L\,a(\lambda_j)\,d(-\lambda_j)\,\sin(2\lambda_j-i\zeta)\frac{\sin(\lambda_j+i\xi_++i\frac{\zeta}{2})}{\sin(\lambda_j-i\xi_--i\frac{\zeta}{2})}\right]
$$
$$
\times\prod_{j<k}\frac{\sin(\lambda_j+\lambda_k-i\zeta)}{\sin(\lambda_j+\lambda_k+i\zeta)}\prod_{k=1}^{N}\frac{\mathbf{a}(-\lambda_k)\prod_{\ell=1}^{N}\mathfrak{s}(\lambda_k-i\zeta,\lambda_\ell)}{i\sin^2(2\lambda_k)\prod_{\ell\neq k}\mathfrak{s}(\lambda_k,\lambda_\ell)}\det_N\big[\mathcal{M}(\boldsymbol{\lambda},\boldsymbol{\lambda})\big]. \tag{3.21}
$$

The matrix $\mathcal{M}(\boldsymbol{\lambda},\boldsymbol{\lambda})$ reads explicitly

$$
\big[\mathcal{M}(\boldsymbol{\lambda},\boldsymbol{\lambda})\big]_{jk} = -2L\,\delta_{jk}\,\widehat{\xi}'(\lambda_j|\{\lambda\})-2\pi\big[K(\lambda_j-\lambda_k)-K(\lambda_j+\lambda_k)\big], \tag{3.22}
$$

in which $\widehat{\xi}'(\mu|\{\lambda\})$ is the following meromorphic function:

$$
\widehat{\xi}'(\mu|\{\lambda\}) = p'(\mu)+\frac{g'(\mu)}{2L}+\frac{2\pi K(2\mu)}{L}-\frac{\pi}{L}\sum_{k=1}^{N}\big[K(\mu-\lambda_k)+K(\mu+\lambda_k)\big], \tag{3.23}
$$

with $p'$ and $K$ given respectively by (3.20) and (3.19), and with

$$
g'(\lambda) = -\sum_{\sigma=\pm}\frac{\sinh(2\xi_\sigma+\zeta)}{\sin(\lambda+i\xi_\sigma+i\zeta/2)\,\sin(\lambda-i\xi_\sigma-i\zeta/2)}. \tag{3.24}
$$

## 4 The ground state(s) in the thermodynamic limit

In this section, we explain how to characterize the configuration of Bethe roots for the ground state(s) of the open XXZ Hamiltonian (1.2) in the regime $\Delta > 1$. As we shall see, the total number of these Bethe roots and their pattern in the complex plane for large $L$ depend non-trivially on the values of the magnetic fields at the boundaries, and so does the presence of an energy gap and of an exponential double degeneracy at $h_+ = h_-$ for even $L$, see Fig. 3.

Hence, we now focus on the regime $\Delta > 1$. We use the following parametrization for the anisotropy parameter $\Delta$ and the boundary fields $h_\sigma$ ($\sigma \in \{+,-\}$) in this regime:

$$\Delta = \cosh\zeta \qquad \text{with} \quad \zeta > 0, \tag{4.1}$$

$$h_\sigma = -\sinh\zeta \coth\xi_\sigma \qquad \text{with} \quad \xi_\sigma = -\tilde{\xi}_\sigma + i\delta_\sigma \frac{\pi}{2}, \tag{4.2}$$

where $\tilde{\xi}_\sigma \in \mathbb{R}$, and

$$\delta_\sigma = \begin{cases} 1 & \text{if } |h_\sigma| < \sinh\zeta, \\ 0 & \text{if } |h_\sigma| > \sinh\zeta. \end{cases} \tag{4.3}$$

The Bethe equations (2.11) can be conveniently rewritten[2] as

$$\mathfrak{a}(\lambda_k|\{\lambda\}) = 1, \qquad k = 1, \ldots, N, \tag{4.4}$$

in terms of the function (3.16). In the homogeneous limit $\xi_n \to -i\zeta/2$, $n = 1, \ldots, L$, the latter reads explicitly

$$\mathfrak{a}(\alpha|\{\lambda\}) = \left(\frac{\sin(\alpha - i\zeta/2)}{\sin(\alpha + i\zeta/2)}\right)^{2L} \frac{\sin(\alpha + i\xi_- + i\zeta/2)\sin(\alpha + i\xi_+ + i\zeta/2)}{\sin(\alpha - i\xi_- - i\zeta/2)\sin(\alpha - i\xi_+ - i\zeta/2)}$$
$$\times \frac{\sin(i\zeta - 2\alpha)}{\sin(i\zeta + 2\alpha)} \prod_{k=1}^{N} \frac{\mathfrak{s}(\alpha + i\zeta, \lambda_k)}{\mathfrak{s}(\alpha - i\zeta, \lambda_k)}. \tag{4.5}$$

Due to the parity and periodicity properties of the problem, we can in fact restrict ourselves to the roots which are contained in the following subspace of the complex plane:

$$D_{\text{sol}} = \left\{\lambda \;\middle|\; 0 < \text{Re}(\lambda) < \frac{\pi}{2} \quad \text{or} \quad \left(\text{Re}(\lambda) = 0, \frac{\pi}{2} \text{ and } \text{Im}(\lambda) < 0\right)\right\}. \tag{4.6}$$

The ground state for the open XXZ chain in the regime $\Delta > 1$ was studied in [19]. It is given by a solution of the Bethe equations where all Bethe roots are real, except a possible isolated complex root. In the thermodynamic limit $L \to +\infty$, the real roots $\alpha_j$ of the Bethe equations for the ground state form a dense distribution on the interval $(0, \frac{\pi}{2})$ (which can be extended by parity on the interval $(-\frac{\pi}{2}, \frac{\pi}{2})$), with density $\rho(\alpha)$ solution of the integral equation:

$$\rho(\alpha) + \int_{-\frac{\pi}{2}}^{\frac{\pi}{2}} K(\alpha - \beta)\rho(\beta)\,d\beta = \frac{p'(\alpha)}{\pi}. \tag{4.7}$$

The latter can be solved explicitly as

$$\rho(\alpha) = \frac{1}{\pi}\sum_{k\in\mathbb{Z}} \frac{e^{2ik\alpha}}{\cosh(k\zeta)} = \frac{1}{\pi}\frac{\vartheta_1'}{\vartheta_2}\frac{\vartheta_3(\alpha, q)}{\vartheta_4(\alpha, q)}, \qquad q = e^{-\zeta}, \tag{4.8}$$

where $\vartheta_i(\alpha, q)$, $i \in \{1, 2, 3, 4\}$, are the Theta functions of nome $q$ defined as in [48], with $\vartheta_1' \equiv \vartheta_1'(0, q)$, $\vartheta_2 \equiv \vartheta_2(0, q)$. It was moreover argued in [19] that the possible additional complex root was issued from the presence of the boundary factors in (4.5). More precisely, according to the study of [19], the latter should correspond to a root which approaches, in the large $L$ limit, one of the two zeroes of the boundary factors of the Bethe equations (4.5):

$$\alpha_{\text{BR}}^\sigma = -i(\zeta/2 + \xi_\sigma + \epsilon_\sigma) = -i(\zeta/2 - \tilde{\xi}_\sigma + \epsilon_\sigma) + \delta_\sigma \frac{\pi}{2}, \qquad \sigma \in \{+, -\}, \tag{4.9}$$

---

[2] When doing this, we have to exclude the possible roots $0$ and $\frac{\pi}{2}$ which are always solutions of (4.4) but should actually correspond to a zero of order 2 in the numerator of (2.11). By treating them apart, it is in fact easy to see that low-energy states do not contain these roots for large $L$.

with exponentially small correction $\epsilon_\sigma = O(L^{-\infty})$ so as to compensate the exponentially large factor in $L$ in the first line of (4.5). Such a complex root $\alpha^\sigma_{\mathrm{BR}}$, which in the following will be called *boundary complex root* or more simply *boundary root*[3], was predicted [19] to exist only if $\tilde{\xi}_\sigma < \zeta/2$, i.e. when the corresponding boundary field $h_\sigma$ is *not* in the interval delimited by the two boundary critical fields $h^{(1)}_{\mathrm{cr}}$ and $h^{(2)}_{\mathrm{cr}}$ defined as [19,20]

$$h^{(1)}_{\mathrm{cr}} = \Delta - 1, \qquad h^{(2)}_{\mathrm{cr}} = \Delta + 1. \tag{4.10}$$

The presence of this kind of boundary root in the ground state was also discussed in [19], in particular in the regime $h_- > 0$, $h_+ < 0$.

It is however not completely clear, from [19], what is the accurate configuration of the Bethe roots for the ground state and the first low-energy states according to the values of the two boundary fields $h_\pm$, notably in our case of interest $h_+ = h_-$ for $L$ even (see Fig. 3) for which we may a priori expect a degeneracy. In the remaining part of the present section, we therefore perform a more detailed study of these configurations, so as to make more precise (and sometimes slightly correct) the predictions of [19]. We in particular show how to control the finite-size corrections up to exponentially small order in $L$, which enables us to discuss the degeneracy at $h_+ = h_-$.

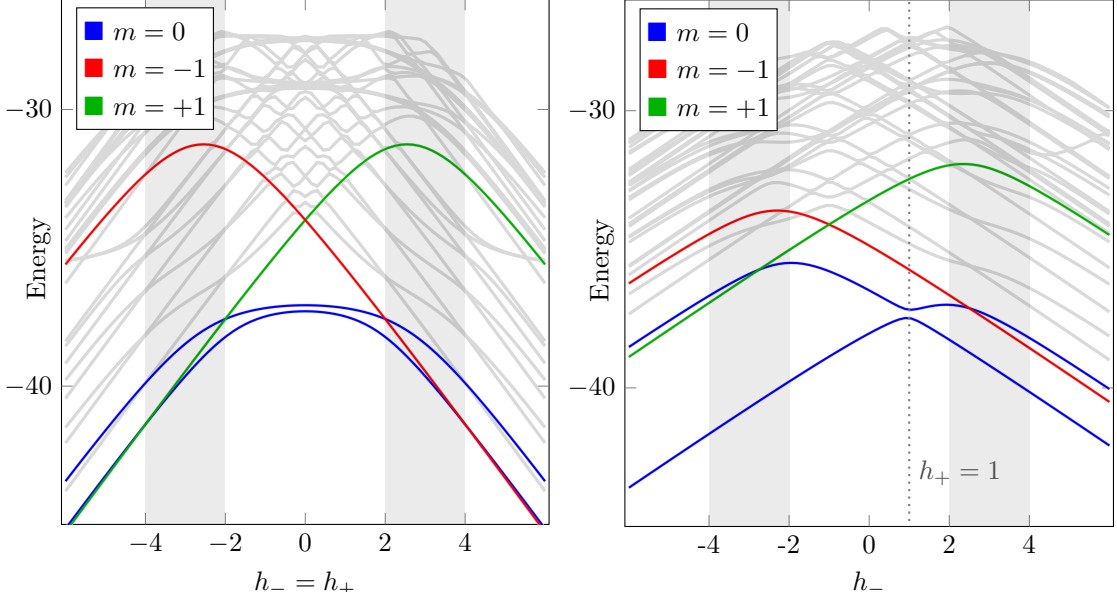

Figure 3: Low energy spectrum for the open boundary XXZ chain ($L = 12$ and $\Delta = 3$) with respect to the boundary fields $h_-, h_+$ from numerical exact diagonalization. *Left*: Applying equal boundary fields, $h_+ = h_-$. The shaded regions correspond to the values between the critical fields $\pm h^{(1)}_{\mathrm{cr}}, \pm h^{(2)}_{\mathrm{cr}}$ and the gray states are other states with various values of total magnetization. The spectrum is gapped and there are two quasi-degenerate ground states (with difference of energy being exponentially small with the system size) in the region $-h^{(1)}_{\mathrm{cr}} < h < h^{(1)}_{\mathrm{cr}}$, while in all other regions the spectrum is gapless and there is no exponential degeneracy of the ground state. *Right*: Spectrum after fixing the boundary field at the right to $h_+ = 1$. The lowest energy states are shown in color with their corresponding magnetization.

---

[3]It was denoted as 'boundary 1-string' in [19].

## 4.1 Properties of low-energy states for large $L$

Low-energy states are given in the thermodynamic limit $L \to \infty$ by an infinite number of real roots (i.e of order $L/2$) and a finite number of complex roots. Using the same argument as in [49], we can show that, if a set of solutions $\{\lambda\} \equiv \{\lambda_1, \ldots, \lambda_N\}$ of (4.4) contains a complex root $\lambda_\ell$ such that $\text{Re}(\lambda_\ell) \neq 0, \frac{\pi}{2}$, then it also contains the conjugate root $\bar{\lambda}_\ell$. Hence complex roots appear by pairs $\lambda_\ell, \bar{\lambda}_\ell$, except possible isolated imaginary roots $\lambda_\ell$ such that $\text{Re}(\lambda_\ell) = 0, \frac{\pi}{2}$.

### 4.1.1 Bethe equations for real roots and counting function

Let us consider a real root $\lambda_j \in \{\lambda\}$. It is convenient to rewrite the corresponding Bethe equation in logarithmic form,

$$\widehat{\xi}(\lambda_j|\{\lambda\}) = \frac{\pi n_j}{L}, \tag{4.11}$$

where $n_j$ is an integer and $\widehat{\xi}(\alpha|\{\lambda\})$ is the counting function. The latter is defined, for the given set of Bethe roots $\{\lambda\}$, as

$$\widehat{\xi}(\alpha|\{\lambda\}) = p(\alpha) + \frac{g(\alpha)}{2L} - \frac{\theta(2\alpha)}{2L} + \frac{1}{2L} \sum_{k=1}^{N} \Theta(\alpha, \lambda_k), \tag{4.12}$$

in terms of the functions

$$p(\alpha) = \int_0^\alpha \varphi'(\mu, \zeta/2) \, d\mu, \tag{4.13}$$

$$\theta(\alpha) = -\int_0^\alpha \varphi'(\mu, \zeta) \, d\mu, \tag{4.14}$$

$$\Theta(\alpha, \lambda_k) = -\frac{1}{2} \int_0^\alpha \Big[ \varphi'(\mu - \lambda_k, \zeta) + \varphi'(\mu - \bar{\lambda}_k, \zeta)$$
$$+ \varphi'(\mu + \lambda_k, \zeta) + \varphi'(\mu + \bar{\lambda}_k, \zeta) \Big] \, d\mu, \tag{4.15}$$

$$g(\alpha) = -\int_0^\alpha \Big[ \varphi'(\mu, \zeta/2 + \xi_+) + \varphi'(\mu, \zeta/2 + \xi_-) \Big] \, d\mu, \tag{4.16}$$

where we have set

$$\varphi'(\mu, \gamma) = \frac{\sinh(2\gamma)}{\sin(\mu + i\gamma) \sin(\mu - i\gamma)}. \tag{4.17}$$

Here we have used the fact that the complex roots $\lambda_k$ always appear in pairs $\lambda_k, \bar{\lambda}_k$, except if $\text{Re}(\lambda_k) \in \{0, \frac{\pi}{2}\}$. Note that the functions $p'$ (3.20) and $g'$ (3.24) that appeared in the expression of the form factor correspond indeed to the derivatives of $p$ and $g$, and that the function $K$ (3.19) is related to the derivative of $\theta$ as $K(\alpha) = -\frac{\theta'(\alpha)}{2\pi}$, so that the function $\widehat{\xi}'$ (3.23) is indeed the derivative of (4.12).

It is possible to determine the range of allowed quantum numbers $n_j$ for the real roots in (4.11) by continuity arguments from the Ising limit $\zeta \to +\infty$. This is done in appendix A. We obtain that $1 \leq n_j \leq M-1$, where $M$ is given by (A.7). Hence we can rewrite the logarithmic Bethe equations (4.11) for the real roots as

$$\widehat{\xi}(\lambda_j|\{\lambda\}) = \frac{\pi j}{L}, \qquad j \in \{1, \ldots, M-1\} \setminus \{h_1, \ldots, h_n\}, \tag{4.18}$$

where $M$ is given by (A.7) and where $h_1, \ldots, h_n$ are the positions of the holes in the adjacent set of quantum numbers for the real roots. It is also convenient to define the rapidities $\check{\lambda}_{h_k}$ of

the holes from the relation

$$\widehat{\xi}(\check{\lambda}_{h_k}|\{\lambda\}) = \frac{\pi h_k}{L}, \qquad k \in \{1, \dots, n\}. \tag{4.19}$$

In the thermodynamic limit $L \to +\infty$, the derivative (3.23) of the counting function (4.12) tends to the density function (4.8) solution of (4.7) multiplied by $\pi$. This comes from the fact that, in the thermodynamic limit, the sums over real Bethe roots turn into integrals with measure given by the density function (4.8). As explained in appendix B, it is possible to control more precisely the finite-size corrections to this transformation sum-integral, in the spirit of what was done in [47,50] (see Proposition B.1 and Corollary B.1), and to decompose the counting function (4.12) in the large $L$ limit according to the different contributions of the real roots, complex roots and holes up to exponentially small corrections in $L$:

$$\widehat{\xi}(\alpha|\{\lambda\}) = \widehat{\xi}_0(\alpha) + \frac{1}{L}\sum_{k \in \mathcal{Z}}\widehat{\xi}_{\lambda_k}(\alpha) - \frac{1}{L}\sum_{j=1}^{n}\widehat{\xi}_{\check{\lambda}_{h_j}}(\alpha) + O(L^{-\infty}), \tag{4.20}$$

in which the first sum runs over the set $\mathcal{Z}$ of indices corresponding to the complex roots (i.e. $\mathrm{Im}(\lambda_k) \neq 0$ if $k \in \mathcal{Z}$), whereas the second sum runs over the positions of the holes. In (4.20), the term

$$\widehat{\xi}_0(\alpha) = \pi \int_0^\alpha \rho(\beta)\,d\beta + \frac{1}{L}\widehat{\xi}_{\mathrm{open}}(\alpha) \tag{4.21}$$

stands for the contribution of the "Fermi sea" of real roots, taking into account the finite-size corrections $\frac{1}{L}\widehat{\xi}_{\mathrm{open}}(\alpha)$ which are common to all low-energy states, see (B.18)-(B.20). The corrections due to the presence of a complex root or a hole with rapidity $\mu$ are given by the corresponding term $\widehat{\xi}_\mu(\alpha)$, see (B.21)-(B.24).

### 4.1.2 Bethe equations for complex roots, boundary roots and wide roots

We now investigate the large $L$ behaviour of the complex solutions of the Bethe equations, and more particularly of the isolated complex roots which, according to [19] and to the study of appendix A, may appear in the set of solutions corresponding to the ground state. Hence, let us now suppose that $\lambda_j \in \{\lambda\}$ is a complex root. One can still use Corollary B.1 to rewrite the sum over real Bethe roots as integrals in the corresponding Bethe equation for large $L$, which gives

$$
\begin{aligned}
\exp\Bigg\{ & 2L\,F(\lambda_j) + \frac{i}{\pi}\int_{-\frac{\pi}{2}}^{\frac{\pi}{2}}\theta(\lambda_j - \mu)\Bigg[\widehat{\xi}'_{\mathrm{open}}(\mu) + \sum_{\ell \in \mathcal{Z}}\widehat{\xi}'_{\lambda_\ell}(\mu) - \sum_{\ell=1}^{n}\widehat{\xi}'_{\check{\lambda}_{h_\ell}}(\mu)\Bigg]d\mu \\
& + \frac{\theta(\lambda_j - \frac{\pi}{2}) + \theta(\lambda_j + \frac{\pi}{2}) + 2\theta(\lambda_j)}{2i} + \sum_{\ell=1}^{n}\frac{\theta(\lambda_j - \check{\lambda}_{h_\ell}) + \theta(\lambda_j + \check{\lambda}_{h_\ell})}{i} + O(L^{-\infty})\Bigg\} \\
& \times \frac{\sin(\lambda_j + i\xi_- + i\zeta/2)\sin(\lambda_j + i\xi_+ + i\zeta/2)}{\sin(\lambda_j - i\xi_- - i\zeta/2)\sin(\lambda_j - i\xi_+ - i\zeta/2)}\prod_{\substack{k \in \mathcal{Z} \\ k \neq j}}\frac{\mathfrak{s}(\lambda_j + i\zeta, \lambda_k)}{\mathfrak{s}(\lambda_j - i\zeta, \lambda_k)} = 1. \tag{4.22}
\end{aligned}
$$

In (4.22) we have set

$$F(z) = ip(z) + \frac{i}{2}\int_{-\frac{\pi}{2}}^{\frac{\pi}{2}}\theta(z - \mu)\rho(\mu)\,d\mu, \tag{4.23}$$

and the functions $p$ and $\theta$ are defined such that

$$e^{ip(\alpha)} = \frac{\sin(i\zeta/2 - \alpha)}{\sin(i\zeta/2 + \alpha)}, \qquad e^{i\theta(\alpha)} = \frac{\sin(i\zeta + \alpha)}{\sin(i\zeta - \alpha)}, \tag{4.24}$$

and such that they coincide with the definitions (4.13) and (4.14) for $\alpha$ real.

It is interesting to investigate the behaviour of (4.23) so as to see how the first line of (4.22) behaves with $L$. Using the terminology of [49], we find that

$$F(\lambda_j) = i\pi \int_0^{\lambda_j} \rho(\mu)\, d\mu, \qquad (4.25)$$

if $\lambda_j$ is a close root, i.e. if $|\mathrm{Im}(\lambda_j)| < \zeta$. The real part of (4.25) is moreover positive if $-\zeta < \mathrm{Im}(\lambda_j) < 0$ (see Fig. 4), which means that, in that case, the first factor in (4.22) is exponentially diverging in $L$. Hence, for (4.22) to be satisfied, $\lambda_j$ has to approach a zero of the expression with exponentially small corrections in $L$. If we suppose moreover that $\lambda_j$ is the only complex root of the set $\{\lambda\}$, i.e. that all other roots are real, this means that $\lambda_j$ has to approach one of the two zeros of the boundary factor in the last line of (4.22), i.e. that $\lambda_j$ is indeed a boundary root of the form (4.9). This can of course only be possible if $-\zeta < \mathrm{Im}(-i\zeta/2 - i\xi_\sigma) < 0$ for some $\sigma \in \{+,-\}$, i.e. if $|\tilde{\xi}_\sigma| < \zeta/2$, which corresponds to

$$h_\sigma \notin [-h_{\mathrm{cr}}^{(2)}, -h_{\mathrm{cr}}^{(1)}] \cup [h_{\mathrm{cr}}^{(1)}, h_{\mathrm{cr}}^{(2)}]. \qquad (4.26)$$

If instead $\lambda_j$ is a wide root, i.e. if $|\mathrm{Im}(\lambda_j)| > \zeta$, then, using similar arguments as in [49], we find that

$$\mathrm{Re}(F(\lambda_j)) = 0, \qquad (4.27)$$

so that the first factor in (4.22) remains finite. If we suppose moreover that $\lambda_j$ is the only complex root of the set $\{\lambda\}$, i.e. that all other roots are real, this means that $\lambda_j$ does no longer converge exponentially fast towards one of the zeros (or poles) of the boundary factor in the last line of (4.22), and therefore is not strictly speaking a boundary root as defined in (4.9).

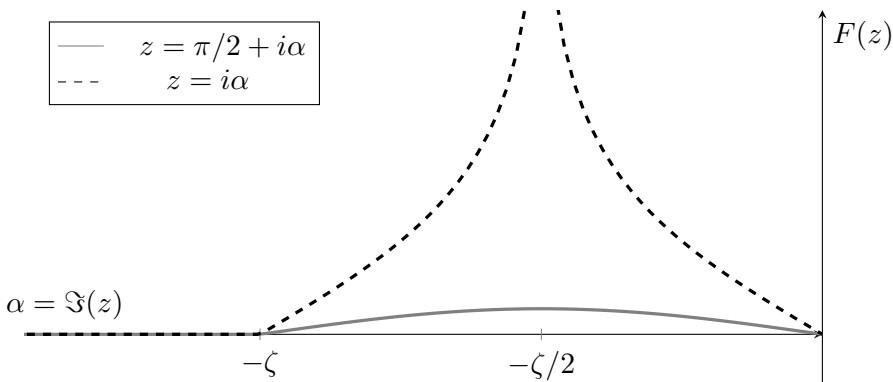

Figure 4: The function $F(z)$ (4.23) for values of $z = i\alpha$ (dashed) and $z = \pi/2 + i\alpha$, (continuous), $\alpha < 0$.

Let us finally remark that we have here found a domain of existence of the boundary root, given by (4.26), which is more narrow than the one ($h_\sigma \notin [h_{\mathrm{cr}}^{(1)}, h_{\mathrm{cr}}^{(2)}]$) found in [19]. This comes from the fact that the reasoning of the authors of [19] did not take into account the full exponential factor given by (4.23), but only the part given by $p(z)$[4].

### 4.1.3 Expression of the energy

Proposition B.1 can also be applied, together with (4.20), to obtain an asymptotic expansion of the energy (2.17) of the corresponding Bethe state in the large $L$ limit up to exponentially

---

[4]In other words, the argument of [19], which is the standard one given by the string hypothesis, is valid only for states in sectors close to $N = 0$ (see also the related work [51]), and not for states in sectors close to $N = \frac{L}{2}$ as those we consider here.

small order in $L$:

$$E(\{\lambda\}) = E_0 + \sum_{k \in \mathcal{Z}} \varepsilon(\lambda_k) - \sum_{j=1}^{n} \varepsilon(\check{\lambda}_{h_j}) + O(L^{-\infty}), \qquad (4.28)$$

where $E_0$ is the contribution of the real roots taking into account the finite-size corrections which are common to all low-energy states, see (B.25), whereas $\varepsilon(\mu)$ is the dressed energy of an excitation with rapidity $\mu$, defined as

$$\varepsilon(\mu) = \varepsilon_0(\mu) + \frac{1}{2\pi} \int_{-\frac{\pi}{2}}^{\frac{\pi}{2}} \varepsilon_0(\beta) \, \widehat{\xi}'_\mu(\beta) \, d\beta. \qquad (4.29)$$

Explicitly,

$$\varepsilon(\mu) = -\pi \sinh \zeta \, [\rho(\mu) + \rho(\bar{\mu})], \qquad (4.30)$$

in terms of the meromorphic elliptic function $\rho(\alpha)$ given by the ratio of Theta functions (4.8) if $\mu$ stands for the rapidity of a hole or of a close root (i.e. if $|\mathrm{Im}(\mu)| < \zeta$), whereas

$$\varepsilon(\mu) = 0, \qquad (4.31)$$

in the case of a wide root (i.e. if $|\mathrm{Im}(\mu)| > \zeta$), see (B.27).

In particular, the dressed energy of a hole with rapidity $\check{\lambda}_h \in (0, \frac{\pi}{2})$ is

$$\varepsilon_h(\check{\lambda}_h) = -\varepsilon(\check{\lambda}_h) = 2 \sinh \zeta \sum_{k \in \mathbb{Z}} \frac{e^{2ik\check{\lambda}_h}}{\cosh(k\zeta)} = 2\pi \sinh \zeta \, \rho(\check{\lambda}_h), \qquad (4.32)$$

as found in [19,49]. Note that (4.32) is a positive and decreasing function of $\check{\lambda}_h$ on the interval $[0, \frac{\pi}{2}]$, see Fig. 5.

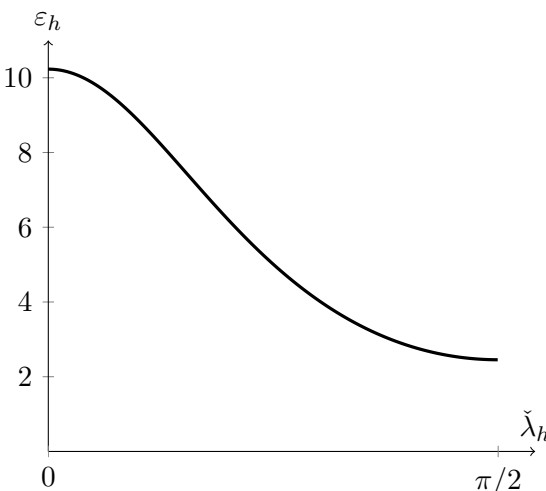

Figure 5: Dressed energy (4.32) of a hole as a function of its hole rapidity $\check{\lambda}_h \in (0, \pi/2)$ for a chain at $\Delta = 3$.

The dressed energy of the boundary root (4.9) is

$$\varepsilon(\alpha_{\mathrm{BR}}^\sigma) = -2\pi \sinh \zeta \, \rho(\alpha_{\mathrm{BR}}^\sigma) = -2\pi \sinh \zeta \, \rho\left(i\tilde{\xi}_\sigma - i\frac{\zeta}{2} + \delta_\sigma \frac{\pi}{2}\right) + O(L^{-\infty}), \qquad (4.33)$$

in its domain of validity (4.26), as found in [19]. We recall that $\delta_\sigma$ is given by (4.3). Note that the expression (4.33) is an odd function of $\tilde{\xi}_\sigma$ (and therefore of $h_\sigma$). It is moreover a

decreasing function of $h_\sigma$ if $|h_\sigma| \notin [h_{cr}^{(1)}, h_{cr}^{(2)}]$, and we have

$$\varepsilon(\alpha_{BR}^\sigma) \xrightarrow[\substack{h_\sigma \to -h_{cr}^{(2)} \\ h_\sigma < -h_{cr}^{(2)}}]{} \varepsilon_h(0), \qquad\qquad \varepsilon(\alpha_{BR}^\sigma) \xrightarrow[\substack{h_\sigma \to -h_{cr}^{(1)} \\ h_\sigma > -h_{cr}^{(1)}}]{} \varepsilon_h(\pi/2), \qquad (4.34)$$

so that the dressed energy (4.33) of the boundary root can be compared to the dressed energy (4.32) of a hole with rapidity $\check{\lambda}_h$ as

$$\varepsilon(\alpha_{BR}^\sigma) > \varepsilon_h(\check{\lambda}_h), \qquad \forall \check{\lambda}_h \in \left(0, \frac{\pi}{2}\right) \qquad \text{if} \quad h_\sigma < -h_{cr}^{(2)}, \qquad (4.35)$$

$$|\varepsilon(\alpha_{BR}^\sigma)| < \varepsilon_h(\check{\lambda}_h), \qquad \forall \check{\lambda}_h \in \left(0, \frac{\pi}{2}\right) \qquad \text{if} \quad |h_\sigma| < h_{cr}^{(1)}. \qquad (4.36)$$

The dressed energy of the boundary root $\alpha_{BR}^\sigma$ is plotted as a function of the field $h_\sigma$ for a specific value of $\Delta$ in Fig. 6.

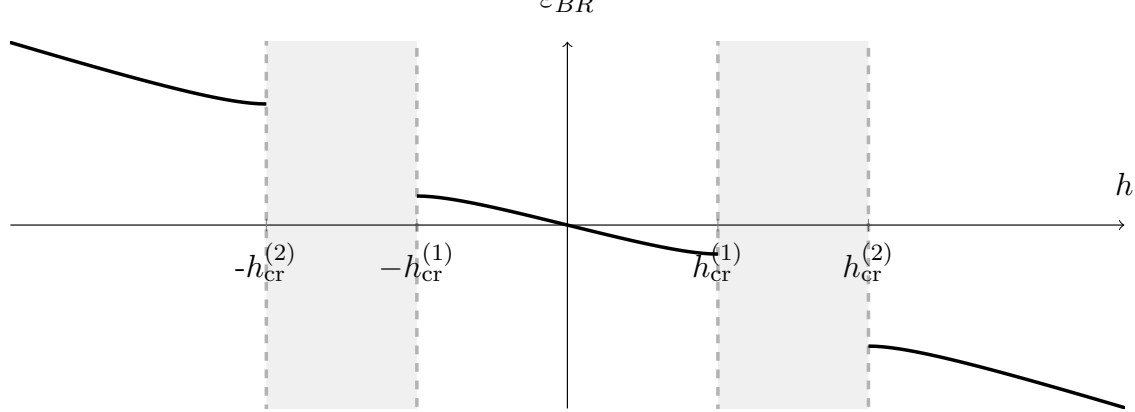

Figure 6: Dressed energy $\varepsilon_{BR} = \varepsilon(\alpha_{BR}^\sigma)$ of the boundary root as a function of the boundary field $h = h_\sigma$ at $\Delta = 3$, as given by (4.33), in its domain of existence (4.26). The two critical fields are here $h_{cr}^{(1)} = 2$ and $h_{cr}^{(2)} = 4$. The dressed energy $\varepsilon_{BR}$ is an odd and decreasing function of $h$. Moreover, the dressed energy of the boundary root tends to the one of a hole with rapidity $\check{\lambda}_h = \frac{\pi}{2}$ when $h \to -h_{cr}^{(1)}$, $h > -h_{cr}^{(1)}$, and it tends to the one of a hole with rapidity $\check{\lambda}_h = 0$ when $h \to -h_{cr}^{(2)}$, $h < -h_{cr}^{(2)}$.

We finally recall that, according to [49], the bulk close complex roots are arranged either in 2-strings or in quartets whose dressed energy vanishes.

## 4.2 Configuration of Bethe roots in the ground state

We now discuss the configuration of the Bethe roots for the ground state according to the values of the two boundary fields $h_+$ and $h_-$. We focus here on the case of a chain with an even number of sites $L$ (for the case of $L$ odd, see section 6).

Let us first suppose that the boundary field of maximal absolute value is non-positive, so as to ensure that the magnetization of the ground state is non-negative[5]. It follows from the study

---

[5]The number of Bethe roots of a given Bethe state constructed as in (2.10) is related to its total magnetization $m = \langle \sum_{n=1}^{L} S_n^z \rangle$ as $m = \frac{L}{2} - N$. In this framework, the states with negative magnetization would correspond to "going beyond the equator", with a number of Bethe roots exceeding the value $L/2$. To avoid this, one has to construct the corresponding Bethe states from the multiple action of $\mathcal{C}$ on the reference state $|\underline{0}\rangle$ with all spins down (or in the dual space from the multiple action of $\mathcal{B}$ on $\langle\underline{0}|$). It is also possible to reach these sectors with negative magnetization by simply using the invariance of the model under the reversal of all spins together with a change of sign of the boundary fields $h_\pm$.

of the previous subsection that the ground state should be given by a configuration of Bethe roots that minimizes the number of holes, except if it is at the cost of containing a boundary root with higher energy, see (4.35). The allowed configurations of Bethe roots according to the values of the boundary fields in the different magnetization sectors can be deduced from the study of appendix A. Combining the results of appendix A with those of subsection 4.1, we can therefore distinguish seven different cases [6] .

**Case 1:** $|h_{\sigma_1}| < h_{cr}^{(1)}$ with $h_{\sigma_1} > h_{\sigma_2}$ ($\{\sigma_1, \sigma_2\} = \{+, -\}$).

The ground state is in the sector with magnetization 0 (i.e. with number of Bethe roots $N = \frac{L}{2}$). It corresponds to the state with $\frac{L}{2} - 1$ real roots with adjacent quantum numbers $n_j = 1, \ldots, \frac{L}{2} - 1$ (no hole) and the boundary root $\alpha_{BR}^{\sigma_1}$.

Indeed, if $|h_{\sigma_1}| < h_{cr}^{(1)}$, it follows from the study of appendix A that all other configurations with $N \leq \frac{L}{2}$ contain either the boundary root $\alpha_{BR}^{\sigma_2}$ with higher dressed energy than $\alpha_{BR}^{\sigma_1}$, or one or more hole(s), and therefore from (4.36) have higher energy. Note that the conclusion still holds even if the boundary field of maximal absolute eigenvalue is positive (but less that $h_{cr}^{(1)}$), since by symmetry of the model under the reversal of all spins together with a change of sign of the boundary fields $h_{\pm}$ we know that the ground state is in this case still in the sector with magnetization 0.

**Case 2:** $h_{\sigma_2} < h_{\sigma_1} < -h_{cr}^{(2)}$ ($\{\sigma_1, \sigma_2\} = \{+, -\}$).

The consideration of the Ising limit indicates that the ground state is in the sector with magnetization 1 (i.e. with number of Bethe roots $N = \frac{L}{2} - 1$). It therefore corresponds to the state with $\frac{L}{2} - 1$ real roots with adjacent quantum numbers $n_j = 1, \ldots, \frac{L}{2} - 1$ (one hole at position $h = \frac{L}{2}$).

Indeed, in that case, the dressed energy of the hole is smaller than the dressed energy of the boundary root, see (4.35), so that the aforementioned state has indeed lower energy than a state with $\frac{L}{2} - 1$ real roots and a boundary root in the sector $\frac{L}{2}$. One can moreover notice that the latter is not even the lowest energy state in its sector, since any state with one hole (and a bulk 2-string or possibly a wide root) would also have a lower energy.

**Case 3:** $-h_{cr}^{(2)} < h_{\sigma_1} < -h_{cr}^{(1)}$ with $h_{\sigma_1} > h_{\sigma_2}$ ($\{\sigma_1, \sigma_2\} = \{+, -\}$).

The ground state has to be found within the states with minimal number of holes $n_h = 1$, which may have the following configurations:

(i) the state, in the sector $N = \frac{L}{2} - 1$, with $\frac{L}{2} - 1$ real roots and a hole at position $h = \frac{L}{2}$;

(ii) a state, in the sector $N = \frac{L}{2}$, with $\frac{L}{2} - 1$ real roots with adjacent quantum numbers $n_j = 1, \ldots, \frac{L}{2} - 1$, a wide root, and a hole at position $h = \frac{L}{2}$;

(iii) a state, in the sector $N = \frac{L}{2}$, with $\frac{L}{2} - 2$ real roots with adjacent quantum numbers $n_j = 1, \ldots, \frac{L}{2} - 2$, a pair of close roots (2-string), and a hole at position $h = \frac{L}{2} - 1$.

Since the dressed energy of a wide root or of a 2-string vanishes, the difference of energy between these states is only given at leading order by the small shift between the hole rapidities. Considering the large $\zeta$ limit (see appendix A), we find that the rapidity of the hole for the configuration (iii) is given at leading order in $\zeta$ by

$$\check{\lambda}_{(iii)} \underset{\zeta \to +\infty}{\sim} \frac{\pi}{2} - \frac{\pi}{L}, \tag{4.37}$$

---

[6]Notice that a similar classification has been recently proposed in [52] in a slightly different context.

whereas the rapidities of the hole for the configurations (i) or (ii) are given at leading order in $\zeta$ by

$$\check{\lambda}_{(i)} \underset{\zeta \to +\infty}{\sim} \check{\lambda}_{(ii)} \underset{\zeta \to +\infty}{\sim} \frac{\pi}{2} - \frac{\pi}{L+2}, \tag{4.38}$$

which seems to indicate that the ground state has to be found within configurations (i) or (ii) only. To conclude further would require a more advanced study of the solutions of the Bethe equations, which is anyway unnecessary for the purpose of the present paper. Let us just mention here that the numerical results from exact diagonalization suggest that, when $h_{\sigma_1} > -\Delta$ the ground state remains in the sector $m = 0$ whatever the value of $h_{\sigma_2}$, whereas when $h_{\sigma_1}$ approaches $-h_{\text{cr}}^{(2)}$, there exists a certain value $h(h_{\sigma_1}) < -h_{\text{cr}}^{(2)}$ at which a transition from the $m = 0$ (for $h_{\sigma_2} > h(h_{\sigma_1})$) to the $m = +1$ (for $h_{\sigma_2} < h(h_{\sigma_1})$) sector occurs, see Fig. 7.

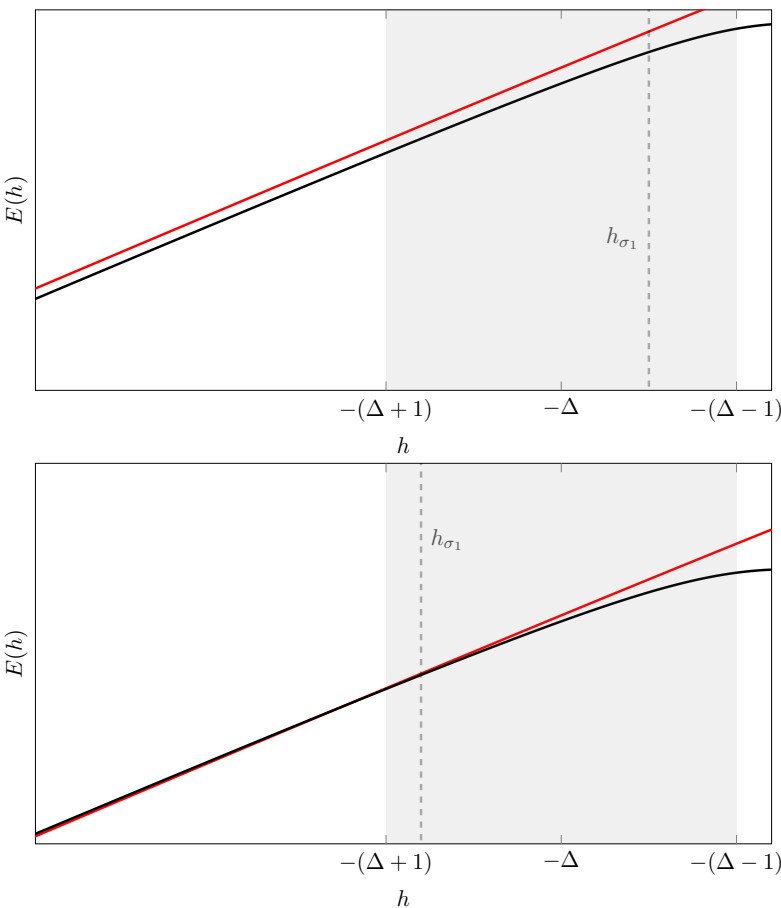

Figure 7: Energies of the lowest states obtained by exact diagonalization, for two values of $h_{\sigma_1}$ in the critical region $(-h_{\text{cr}}^{(2)}, -h_{\text{cr}}^{(1)})$, and varying $h_{\sigma_2}$. In black: $m = 0$. In red: $m = +1$. Notice that, as we pick $h_{\sigma_1}$ closer to $-h_{\text{cr}}^{(2)}$, a crossing of levels appears when $h_{\sigma_2}$ becomes small enough. Here $\Delta = 3$, $L = 14$.

**Case 4:** $h_{\sigma_2} < h_{\text{cr}}^{(1)} < h_{\sigma_1} < h_{\text{cr}}^{(2)}$ ($\{\sigma_1, \sigma_2\} = \{+, -\}$).

The ground state is in the sector with magnetization 0 (i.e. with number of Bethe roots $N = \frac{L}{2}$), even if $|h_{\sigma_1}| > |h_{\sigma_2}|$ (this follows by symmetry from the study of previous cases). It therefore corresponds to the state with $\frac{L}{2}$ real roots with adjacent quantum numbers $n_j = 1, \ldots, \frac{L}{2}$ (no hole).

**Case 5:** $h_{\sigma_2} < h_{\mathrm{cr}}^{(1)} < h_{\mathrm{cr}}^{(2)} < h_{\sigma_1}$ ($\{\sigma_1, \sigma_2\} = \{+,-\}$).

The ground state is in the sector with magnetization 0 (i.e. with number of Bethe roots $N = \frac{L}{2}$), even if $|h_{\sigma_1}| > |h_{\sigma_2}|$ (this follows by symmetry from the study of previous cases). It corresponds to the state with $\frac{L}{2}-1$ real roots with adjacent quantum numbers $n_j = 1, \ldots, \frac{L}{2}-1$ (no hole) and the boundary root $\alpha_{\mathrm{BR}}^{\sigma_1}$.

**Case 6:** $h_{\mathrm{cr}}^{(2)} < h_{\sigma_2} < h_{\sigma_1}$ ($\{\sigma_1, \sigma_2\} = \{+,-\}$).

This case can be obtained by symmetry from Case 2. The ground state is in the sector with magnetization $-1$ and hence is beyond the equator.

**Case 7:** $h_{\mathrm{cr}}^{(1)} < h_{\sigma_2} < h_{\mathrm{cr}}^{(2)}$ with $h_{\sigma_1} > h_{\sigma_2}$ ($\{\sigma_1, \sigma_2\} = \{+,-\}$).

This case can be obtained by symmetry from Case 3. Depending on the values of the magnetic fields, it may be:

(i) a state, in the sector $N = \frac{L}{2}$ of magnetization 0, with one hole and either

- $\frac{L}{2}$ real roots (if $h_{\sigma_1} \in [h_{\mathrm{cr}}^{(1)}, h_{\mathrm{cr}}^{(2)}]$),
- $\frac{L}{2}-1$ real roots and the boundary root $\alpha_{\mathrm{BR}}^{\sigma_1}$ (if $h_{\sigma_1} > h_{\mathrm{cr}}^{(2)}$, the energy of the boundary root being in that case negative and bigger, in absolute value, than the energy of the hole);

(ii) a state with magnetization $-1$, which is beyond the equator.

It also follows from the previous study that, in the thermodynamic limit, the ground state is separated by a gap of energy from the excited states only in Cases 1, 4 and 5.

## 4.3 The two states of lowest energy in the regime $|h_\pm| < h_{\mathrm{cr}}^{(1)}$

Still for chains of even size $L$, we now focus on the regime where both boundary fields $h_\pm$ are such that $|h_\pm| < h_{\mathrm{cr}}^{(1)}$, and investigate more thoroughly the ground state and the first excited state in this regime.

### 4.3.1 The case $h_+ \neq h_-$

If $h_{\sigma_1} > h_{\sigma_2}$ (with $\{\sigma_1, \sigma_2\} = \{+,-\}$), we have seen that the ground state is the state in the sector $N = \frac{L}{2}$ with $\frac{L}{2}-1$ real roots with adjacent quantum numbers $n_j = 1, \ldots, \frac{L}{2}-1$ (no hole) and the boundary root $\alpha_{\mathrm{BR}}^{\sigma_1}$. Moreover, it is easy to see from similar arguments that the excited state with lowest energy is the state in the sector $N = \frac{L}{2}$ with $\frac{L}{2}-1$ real roots with adjacent quantum numbers $n_j = 1, \ldots, \frac{L}{2}-1$ (no hole) and the boundary root $\alpha_{\mathrm{BR}}^{\sigma_2}$.

Let us denote by $\{\alpha^+\}$ and $\{\alpha^-\}$ the two sets of $N = \frac{L}{2}$ Bethe roots corresponding to each of these two states, with $\alpha_1^\pm, \ldots, \alpha_{N-1}^\pm$ being real roots, and

$$\alpha_N^\pm = \alpha_{\mathrm{BR}}^\pm = -i(\zeta/2 + \xi_\pm + \epsilon_\pm), \tag{4.39}$$

being the corresponding boundary root. It follows from (4.22) that the deviation $\epsilon_\pm$ is exponentially small in $L$:

$$\epsilon_\pm = e^{-2LF(-i\zeta/2 - i\xi_\pm) + O(1)} = O(L^{-\infty}). \tag{4.40}$$

Hence, since $\xi_+ \neq \xi_-$, these two boundary roots remain at finite distance from each other:

$$\alpha_N^+ - \alpha_N^- = \alpha_{\mathrm{BR}}^+ - \alpha_{\mathrm{BR}}^- = -i(\xi_+ - \xi_- + O(L^{-\infty})). \tag{4.41}$$

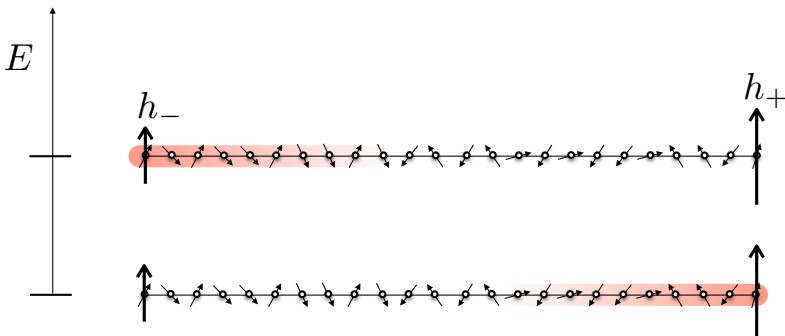

Figure 8: Pictorial representation of the ground state and the first excited state in the regime $|h_\pm| < h_{\text{cr}}^{(1)}$. The boundary root is a localized excitation (here represented in red) pinned, in the ground state, at the edge of the chain where the field is the largest. Here $h_+ > h_-$, and the ground state has the boundary root localized around the right edge, (with exponential tails), i.e with $\alpha_{\text{BR}}^+ = \pi/2 - i(\zeta/2 - \tilde{\xi}_+) + O(L^{-\infty})$, while the first excited state has it localised around the left edge, i.e with $\alpha_{\text{BR}}^- = \pi/2 - i(\zeta/2 - \tilde{\xi}_-) + O(L^{-\infty})$.

From (4.28), the difference of energy between these two states remains finite in the thermodynamic limit:

$$E_+ - E_- = \varepsilon(-i\zeta/2 - i\xi_+) - \varepsilon(-i\zeta/2 - i\xi_-) + O(L^{-\infty}). \tag{4.42}$$

Note that there also exists a finite gap of energy between these two states and the remaining part of the spectrum, the latter corresponding to Bethe states with one or more hole(s) and therefore leading to continuous distributions of energy in the thermodynamic limit.

If we denote by $\widehat{\xi}_\pm$ the counting functions corresponding to these two states, we obtain from (4.20) that

$$\widehat{\xi}_+(\alpha) - \widehat{\xi}_-(\alpha) = \frac{1}{L}\left(\widehat{\xi}_{\alpha_{\text{BR}}^+}(\alpha) - \widehat{\xi}_{\alpha_{\text{BR}}^-}(\alpha)\right) + O(L^{-\infty}). \tag{4.43}$$

Hence, using the fact that $\widehat{\xi}_+(\alpha_j^+) = \widehat{\xi}_-(\alpha_j^-)$ for $j = 1, \dots, N-1$,

$$\widehat{\xi}_+(\alpha_j^-) - \widehat{\xi}_-(\alpha_j^-) = (\alpha_j^- - \alpha_j^+)\widehat{\xi}_+'(\alpha_j^+) + O\left((\alpha_j^- - \alpha_j^+)^2\right)$$

$$= \frac{1}{L}\left(\widehat{\xi}_{\alpha_{\text{BR}}^+}(\alpha_j^-) - \widehat{\xi}_{\alpha_{\text{BR}}^-}(\alpha_j^-)\right) + O(L^{-\infty}), \tag{4.44}$$

so that the deviation $\delta_j$ between the real Bethe roots of the two states is of order $1/L$:

$$\delta_j = \alpha_j^- - \alpha_j^+ = \frac{1}{L}\frac{\widehat{\xi}_{\alpha_{\text{BR}}^+}(\alpha_j^-) - \widehat{\xi}_{\alpha_{\text{BR}}^-}(\alpha_j^-)}{\widehat{\xi}_+'(\alpha_j^+)} + O(L^{-2}) = O(L^{-1}). \tag{4.45}$$

### 4.3.2 The ground state degeneracy at $h_+ = h_-$

Let us now consider the particular case $h_- = h_+ = h$, at which the ground state becomes degenerate in the thermodynamic limit. When $h_+ = h_- = h$, namely $\xi_- = \xi_+ = \xi$, the Bethe equations (3.16) contain a zero of second order which is given by the product of the two field-dependent factors:

$$\left(\frac{\sin(\alpha + i\xi_- + i\zeta/2)}{\sin(\alpha - i\xi_- - i\zeta/2)}\right)\left(\frac{\sin(\alpha + i\xi_+ + i\zeta/2)}{\sin(\alpha - i\xi_+ - i\zeta/2)}\right) = \left(\frac{\sin(\alpha + i\xi + i\zeta/2)}{\sin(\alpha - i\xi - i\zeta/2)}\right)^2. \tag{4.46}$$

Let us consider a state, in the sector $N = \frac{L}{2}$, with $N-1$ real roots $\alpha_1, \ldots, \alpha_{N-1}$ with adjacent quantum numbers $n_j = 1, \ldots, N-1$ and a complex root $\alpha_{\mathrm{BR}}$ at

$$\alpha_{\mathrm{BR}} = -i(\zeta/2 + \xi + \epsilon) = -i(\zeta/2 - \tilde{\xi} + \epsilon) + \frac{\pi}{2}, \tag{4.47}$$

and let us evaluate more precisely the deviation $\epsilon$ of this complex root with respect to the position of the double zero in the large $L$ limit. The Bethe equation (3.16) for the complex root is

$$\left(\frac{\sin(\alpha_{\mathrm{BR}} - i\zeta/2)}{\sin(\alpha_{\mathrm{BR}} + i\zeta/2)}\right)^{2L} \left(\frac{\sin(\alpha_{\mathrm{BR}} + i\xi + i\zeta/2)}{\sin(\alpha_{\mathrm{BR}} - i\xi - i\zeta/2)}\right)^2$$
$$\times \prod_{k=1}^{N-1} \frac{\sin(\alpha_{\mathrm{BR}} - \alpha_k + i\zeta)\sin(\alpha_{\mathrm{BR}} + \alpha_k + i\zeta)}{\sin(\alpha_{\mathrm{BR}} - \alpha_k - i\zeta)\sin(\alpha_{\mathrm{BR}} + \alpha_k - i\zeta)} = 1. \tag{4.48}$$

Hence, using (4.47) and keeping the leading order terms in $\epsilon$, we obtain

$$\left(\frac{\sinh(\zeta + \xi)}{\sinh\xi}\right)^{2L} \left(\frac{\sinh\epsilon}{\sinh(2\xi + \zeta)}\right)^2 \exp\left[L\,O(\epsilon)\right]$$
$$\times \exp\left\{-i\sum_{k=1}^{N-1}\left[\theta\big(i(\zeta/2 + \xi) + \alpha_k\big) + \theta\big(i(\zeta/2 + \xi) - \alpha_k\big)\right]\right\} = 1. \tag{4.49}$$

We can now use Corollary B.1 so as to replace the sum over the real roots in (4.48) by an integral in the large $L$ limit by means of (B.13). It leads to

$$\epsilon \exp\left[L\,O(\epsilon)\right] = \pm\left\{\sinh^2(\zeta + 2\xi)\left(\frac{\sinh\xi}{\sinh(\zeta + \xi)}\right)^{2L}\right.$$
$$\times \exp\left[\frac{\theta\big(i(\zeta/2 + \xi) + \frac{\pi}{2}\big) + \theta\big(i(\zeta/2 + \xi) - \frac{\pi}{2}\big) + 2\theta\big(i(\zeta/2 + \xi)\big)}{2i}\right]$$
$$\left.\times \exp\left[\frac{iL}{\pi}\int_{-\frac{\pi}{2}}^{\frac{\pi}{2}} \theta\big(i(\zeta/2 + \xi) - x\big)\widehat{\xi}'(x)\,dx\right]\right\}^{1/2}\left(1 + O(L^{-\infty})\right)$$
$$= \pm\exp\left[-L\,F(-i\zeta/2 - i\xi) + G(\xi)\right]\left(1 + O(L^{-\infty})\right), \tag{4.50}$$

where $F(-i\zeta/2 - i\xi)$ is given by (4.23) and $G(\xi)$ is a term of order 1 when $L \to +\infty$. We recall that $F(-i\zeta/2 - i\xi)$ is positive for $|h| < h_{\mathrm{cr}}^{(1)}$ (see Fig. 4, or more explicitly Fig. 9 in which this term is plotted as a function of the boundary magnetic field $h$), so that the deviation $\epsilon$ is exponentially decreasing in $L$ in this regime, as expected[7]. Moreover, we see from (4.50) that there are two possible choices $\epsilon_{\pm}$ for the deviation $\epsilon$, corresponding to the two possible choices of the sign in (4.50). Hence there are two different states with $N-1$ real roots $\alpha_1^{\pm} \ldots, \alpha_{N-1}^{\pm}$ and one boundary complex root $\alpha_{\mathrm{BR}}^{\pm}$, that we shall denote by superscripts $+$ or $-$ according to the sign of the leading correction of the complex root in (4.50) (note that the $+$ or $-$ denomination is here not related to the left or right boundary, but only to the fact that there are two different solutions for the complex root position corresponding to the two different signs in (4.50)).

From (4.50), the boundary roots for these two states are exponentially close in $L$. If we denote by $\widehat{\xi}_{\pm}$ the corresponding counting function, it follows from (4.20) and Appendix B.2 that

$$\widehat{\xi}_+(\alpha) - \widehat{\xi}_-(\alpha) = \frac{1}{L}\left(\widehat{\xi}_{\alpha_{\mathrm{BR}}^+}(\alpha) - \widehat{\xi}_{\alpha_{\mathrm{BR}}^-}(\alpha)\right) + O(L^{-\infty}) = O(L^{-\infty}). \tag{4.51}$$

---

[7]Note that we more generally recover from Fig. 9 the regimes (4.26) of existence of the boundary root $(h \notin [-h_{\mathrm{cr}}^{(2)}, -h_{\mathrm{cr}}^{(1)}] \cup [h_{\mathrm{cr}}^{(1)}, h_{\mathrm{cr}}^{(2)}])$ for which $F(-i\zeta/2 - i\xi) < 0$.

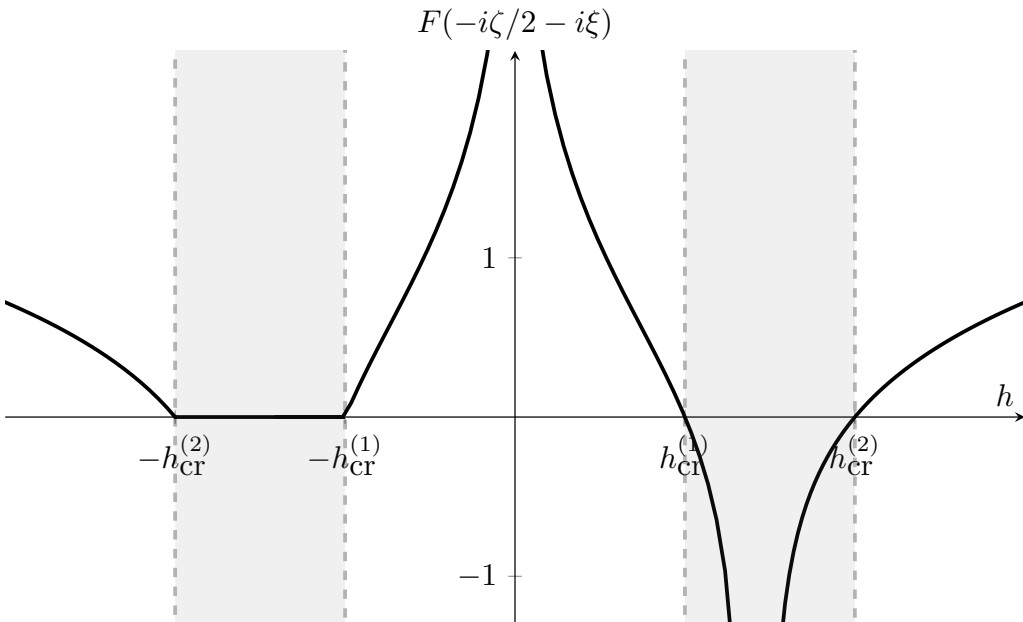

Figure 9: Values of the coefficient $F(-i\zeta/2 - i\xi)$ as a function of the magnetic field $h$ and for $\Delta = 3$ (therefore $h_{\mathrm{cr}}^{(1)} = 2$ and $h_{\mathrm{cr}}^{(2)} = 4$).

Hence, using the fact that $\widehat{\xi}_+(\alpha_j^+) = \widehat{\xi}_-(\alpha_j^-)$ for $j = 1, \ldots, N-1$, we can deduce from (4.51) that

$$\widehat{\xi}_+(\alpha_j^-) - \widehat{\xi}_-(\alpha_j^-) = (\alpha_j^- - \alpha_j^+)\widehat{\xi}'_+(\alpha_j^+) + o(\alpha_j^- - \alpha_j^+) = O(L^{-\infty}), \tag{4.52}$$

so that the real roots of these two states are also exponentially close in $L$. It moreover follows from (4.51) and (4.28) that the difference of energy between these two states is also exponentially small in $L$:

$$E_+ - E_- = O(L^{-\infty}). \tag{4.53}$$

Furthermore, since from appendix A other types of states are given by solutions of the Bethe equations with at least one hole, there is a gap of energy between these two quasi-degenerate ground states and the other excited states.

Let us finally remark that the exponential degeneracy at $h_+ = h_-$ and the gap in the spectrum are no longer present in the other regimes. Indeed, in the regimes $h \in (-h_{\mathrm{cr}}^{(2)}, -h_{\mathrm{cr}}^{(1)})$ and $h < -h_{\mathrm{cr}}^{(2)}$, it follows from our previous study that the lowest energy states contain one hole, and that their difference of energy is a direct consequence of the difference of rapidities of the hole. This is in agreement with numerical results obtained by exact diagonalization (see Fig. 3).

## 5 Form factors in the thermodynamic limit

In this section, we compute the thermodynamic limit $L \to \infty$ ($L$ even) of the expression for the boundary form factor of $\sigma_1^z$ obtained in section 3 in two particular cases. We first consider the case of the boundary magnetization (i.e. both Bethe states coincide with each other and with the ground state) for generic values of the boundary magnetic fields. We then consider the form factor between the two states of lowest energy in the regime $|h_\pm| < h_{\mathrm{cr}}^{(1)}$: when $h_+ \neq h_-$, we show that this form factor vanishes exponentially fast with $L$ whereas, for $h_+ = h_- = h$, it tends to a finite value which gives the large time limit of the boundary spin-spin autocorrelation function (1.3).

## 5.1 Boundary magnetization in the ground state

Let us first explain how to obtain from (3.13) the value of the boundary magnetization in the thermodynamic limit, namely the mean value $\langle \sigma_1^z \rangle$ in the ground state. This quantity has already been computed by different methods for $T = 0$ and $h_+ = 0$ in [20, 21, 25][8], and for finite $T$ in [53], together with [54] where the boundary free energy was obtained for generic boundary conditions at one edge of the chain. It is relevant to see how one can derive it directly from the finite-size form factor by taking into account the precise large-$L$ structure of the Bethe roots for the ground state that we have obtained in the previous section. We shall see in particular that, since this structure depends on *both* boundary fields (and therefore also on the right boundary field $h_+$ at infinity), so does the large-$L$ limit of the boundary magnetization.

From the expressions (3.13) and (3.21), the mean value of the operator $\sigma_1^z$ in an eigenstate $|\{\lambda\}\rangle$ is

$$\frac{\langle\{\lambda\}|\sigma_1^z|\{\lambda\}\rangle}{\langle\{\lambda\}|\{\lambda\}\rangle} = \frac{\det_N[\mathcal{M}(\boldsymbol{\lambda},\boldsymbol{\lambda}) - 2\mathcal{P}(\boldsymbol{\lambda},\boldsymbol{\lambda})]}{\det_N \mathcal{M}(\boldsymbol{\lambda},\boldsymbol{\lambda})} = 1 - 2\,\mathrm{tr}\left[\mathcal{M}(\boldsymbol{\lambda},\boldsymbol{\lambda})^{-1} \cdot \mathcal{P}(\boldsymbol{\lambda},\boldsymbol{\lambda})\right], \qquad (5.1)$$

in which $\mathcal{M}(\boldsymbol{\lambda},\boldsymbol{\lambda})$ is given by (3.22), and $\mathcal{P}(\boldsymbol{\lambda},\boldsymbol{\lambda})$ by (see (3.18))

$$\left[\mathcal{P}(\boldsymbol{\lambda},\boldsymbol{\lambda})\right]_{jk} = p''(\lambda_j)\mathsf{v}(\lambda_k), \qquad (5.2)$$

where $p''$ is the derivative of the function (3.20) and

$$\begin{aligned}
\mathsf{v}(\lambda) &= i \sinh \xi_- \left[\frac{\sin(\lambda + i\frac{\zeta}{2})}{\sin(\lambda + i\xi_- + i\frac{\zeta}{2})} - \frac{\sin(\lambda - i\frac{\zeta}{2})}{\sin(\lambda - i\xi_- - i\frac{\zeta}{2})}\right] \\
&= \frac{\sinh^2 \xi_- \sin(2\lambda)}{\sin(\lambda - i\xi_- - i\frac{\zeta}{2}) \sin(\lambda + i\xi_- + i\frac{\zeta}{2})}.
\end{aligned} \qquad (5.3)$$

Let us now particularise the state $|\{\lambda\}\rangle$ in (5.1) to be the ground state of the open XXZ spin chain. We denote by $\alpha_1, \dots, \alpha_N$ the corresponding Bethe roots, by $\widehat{\xi}(\mu) \equiv \widehat{\xi}(\mu|\{\alpha\})$ the corresponding counting function, and set $\boldsymbol{\alpha} \equiv (\alpha_1, \dots, \alpha_N)$. From the results of section 4.2, either all $N$ Bethe roots are real, or $N-1$ of them are real whereas one of them, say $\alpha_N$, is an isolated complex root. We need then to compute the following trace in the thermodynamic limit:

$$\begin{aligned}
\mathrm{tr}\left[\mathcal{M}(\boldsymbol{\alpha},\boldsymbol{\alpha})^{-1} \cdot \mathcal{P}(\boldsymbol{\alpha},\boldsymbol{\alpha})\right] &= \sum_{j,k=1}^{N} \left[\mathcal{M}(\boldsymbol{\alpha},\boldsymbol{\alpha})^{-1}\right]_{kj} \left[\mathcal{P}(\boldsymbol{\alpha},\boldsymbol{\alpha})\right]_{jk} \\
&= \sum_{j,k=1}^{N} \left[\mathcal{M}(\boldsymbol{\alpha},\boldsymbol{\alpha})^{-1}\right]_{kj} p''(\alpha_j)\mathsf{v}(\alpha_k) \\
&= \sum_{k=1}^{N} \mathsf{u}(\alpha_k)\mathsf{v}(\alpha_k),
\end{aligned} \qquad (5.4)$$

in which the vector $(\mathsf{u}(\alpha_1), \dots, \mathsf{u}(\alpha_N))$ is obtained as the result of the action of the matrix $\mathcal{M}(\boldsymbol{\alpha},\boldsymbol{\alpha})^{-1}$ on the vector $(p''(\alpha_1), \dots, p''(\alpha_N))$, i.e. is such that

$$\sum_{\ell=1}^{N} [\mathcal{M}(\boldsymbol{\alpha},\boldsymbol{\alpha})]_{j\ell}\, \mathsf{u}(\alpha_\ell) = p''(\alpha_j), \qquad 1 \le j \le N. \qquad (5.5)$$

---

[8]The influence of the right boundary field $h_+$ was not taken into account in the half-infinite chain limit $L \to \infty$ that was considered *a priori* in [20, 21, 25]. It means that the $L \to \infty$ results of [20, 21, 25] correspond in fact to the case of a free boundary condition at infinity, i.e. $h_+ = 0$.

Let us suppose that this vector can be obtained from an odd $\pi$-periodic function $\mathsf{u}$ (so that in particular $\mathsf{u}(0) = \mathsf{u}(\frac{\pi}{2}) = 0$) which is moreover $\mathcal{C}^\infty$ on the real axis. Then we can use Corollary B.1 to change the sum over real roots into an integral in the left hand side of (5.5). It gives

$$
\sum_{\ell=1}^{N}[\mathcal{M}(\boldsymbol{\alpha},\boldsymbol{\alpha})]_{j\ell}\,\mathsf{u}(\alpha_\ell)
$$

$$
= -2L\,\widehat{\xi}'(\alpha_j)\,\mathsf{u}(\alpha_j) - 2\pi\sum_{\ell=1}^{N}\bigl[K(\alpha_j-\alpha_\ell)-K(\alpha_j+\alpha_\ell)\bigr]\mathsf{u}(\alpha_\ell)
$$

$$
= -2L\Bigl\{\widehat{\xi}'(\alpha_j)\,\mathsf{u}(\alpha_j) + \int_{-\frac{\pi}{2}}^{\frac{\pi}{2}}K(\alpha_j-v)\,\widehat{\xi}'(v)\,\mathsf{u}(v)\,dv
$$

$$
+ \frac{\pi}{L}\sum_{\ell\in\mathcal{Z}}\bigl[K(\alpha_j-\alpha_\ell)-K(\alpha_j+\alpha_\ell)\bigr]\mathsf{u}(\alpha_\ell)
$$

$$
- \frac{\pi}{L}\sum_{\ell=1}^{n}\bigl[K(\alpha_j-\check{\alpha}_{h_\ell})-K(\alpha_j+\check{\alpha}_{h_\ell})\bigr]\mathsf{u}(\check{\alpha}_{h_\ell}) + O(L^{-\infty})\Bigr\}.
\tag{5.6}
$$

Note that, in the case of the ground state that we consider here, the set of complex roots is either empty or equal to $\alpha_N$, and the number $n$ of holes is either 0 or 1. It is easy to solve (5.5) at leading order in $L$, by noticing that the function $p''$ can be obtained as

$$
p''(\alpha) = \pi\rho'(\alpha) + \pi\int_{-\frac{\pi}{2}}^{\frac{\pi}{2}}K(\alpha-v)\rho'(v)\,dv,
\tag{5.7}
$$

in terms of the derivative $\rho'$ of the function (4.8), see (4.7). Therefore, the $\mathsf{u}$ solving (5.5) is of the form

$$
\mathsf{u}(\alpha) = -\frac{\pi}{2L\,\widehat{\xi}'(\alpha)}\bigl[\rho'(\alpha) + \mathsf{u}_1(\alpha)\bigr],
\tag{5.8}
$$

where $\mathsf{u}_1(\alpha)$ is a correction of order $O(\frac{1}{L})$ (or even of order $O(L^{-\infty})$ if the ground state does neither contain a complex root nor a hole). Note that the leading term in (5.8) is indeed an odd $\pi$-periodic meromorphic function with no pole on the real axis. Hence, combining this result with (5.4), we obtain that

$$
\mathrm{tr}\bigl[\mathcal{M}(\boldsymbol{\alpha},\boldsymbol{\alpha})^{-1}\cdot\mathcal{P}(\boldsymbol{\alpha},\boldsymbol{\alpha})\bigr] = -\frac{\pi}{2L}\sum_{k=1}^{N}\frac{\rho'(\alpha_k)+\mathsf{u}_1(\alpha_k)}{\widehat{\xi}'(\alpha_k)}\,\mathsf{v}(\alpha_k)
$$

$$
= -\frac{1}{4}\int_{-\frac{\pi}{2}}^{\frac{\pi}{2}}\bigl[\rho'(\alpha)+\mathsf{u}_1(\alpha)\bigr]\mathsf{v}(\alpha)\,d\alpha + \frac{\pi}{2L}\sum_{j=1}^{n}\frac{\rho'(\check{\alpha}_{h_j})+\mathsf{u}_1(\check{\alpha}_{h_j})}{\widehat{\xi}'(\check{\alpha}_{h_j})}\,\mathsf{v}(\check{\alpha}_{h_j})
$$

$$
- \frac{\pi}{2L}\sum_{k\in\mathcal{Z}}\frac{\rho'(\alpha_k)+\mathsf{u}_1(\alpha_k)}{\widehat{\xi}'(\alpha_k)}\,\mathsf{v}(\alpha_k) + O(L^{-\infty}),
\tag{5.9}
$$

in which we have again replaced the sum over real roots by integrals. Note that the contributions of the complex root and/or hole vanish in the thermodynamic limit $L\to\infty$, except for the case of the boundary root $\alpha_{\mathrm{BR}}^- = -i(\zeta/2 + \xi_- + \epsilon_-)$ for which the coefficient $\mathsf{v}(\alpha_{\mathrm{BR}}^-)$ diverges as the inverse of the boundary root deviation $\epsilon_-$:

$$
\mathsf{v}(\alpha_{\mathrm{BR}}^-) = i\,\frac{\sinh^2\xi_-}{\epsilon_-}\bigl(1 + O(\epsilon_-)\bigr).
\tag{5.10}
$$

This divergence is compensated in (5.9) by the fact that the function $2L\,\widehat{\xi}'$ itself diverges at $\alpha_{\mathrm{BR}}^-$, via the contribution $g'(\alpha_{\mathrm{BR}}^-)$, as the inverse of the boundary root deviation $\epsilon_-$:

$$2L\,\widehat{\xi}'(\alpha_{\mathrm{BR}}^-) = \frac{1+\delta_{\xi_+,\xi_-}}{\epsilon_-}\big(1+O(\epsilon_-)\big). \tag{5.11}$$

In other words, the divergence in (5.10) is compensated by a divergence of the same order in the last row of the matrix $\mathcal{M}(\boldsymbol{\alpha},\boldsymbol{\alpha})$ (3.22) if $\alpha_N = \alpha_{\mathrm{BR}}^-$:

$$[\mathcal{M}(\boldsymbol{\alpha},\boldsymbol{\alpha})]_{Nk} = -\frac{1}{\epsilon_-}\big[(1+\delta_{\xi_-,\xi_+})\delta_{Nk} + O(\epsilon_-)\big]. \tag{5.12}$$

The presence of the factor $(1+\delta_{\xi_-,\xi_+})$ in (5.11) or in (5.12), which is equal to 1 when the two boundary fields are different and to 2 when they are equal, is due to the fact that the term $g'$, see eq. (3.24), is summed over the two boundary fields: hence, when the latter are equal, the boundary root approaches a pole for both factors. Finally,

$$\lim_{L\to\infty}\mathrm{tr}\big[\mathcal{M}(\boldsymbol{\alpha},\boldsymbol{\alpha})^{-1}\cdot\mathcal{P}(\boldsymbol{\alpha},\boldsymbol{\alpha})\big] = -\frac{1}{4}\int_{-\frac{\pi}{2}}^{\frac{\pi}{2}}\rho'(\alpha)v(\alpha)\,d\alpha \quad -\delta_{\alpha_N,\alpha_{\mathrm{BR}}^-}\frac{i\pi\,\sinh^2\xi_-}{1+\delta_{\xi_-,\xi_+}}\rho'(\alpha_{\mathrm{BR}}^-), \tag{5.13}$$

in which the symbol $\delta_{\alpha_N,\alpha_{\mathrm{BR}}^-}$ indicates that the last term exists only when one of the Bethe roots (and by convention the last one) coincides with the boundary root $\alpha_{\mathrm{BR}}^-$.

Hence, the thermodynamic limit of the boundary magnetization in the ground state is given by

$$\lim_{L\to\infty}\langle\sigma_1^z\rangle = \langle\sigma_1^z\rangle_0 + \langle\sigma_1^z\rangle_{\mathrm{BR}}, \tag{5.14}$$

where $\langle\sigma_1^z\rangle_0$ denotes the contribution given by the dense distribution of real roots, which is

$$\langle\sigma_1^z\rangle_0 = 1 + \frac{\sinh^2\xi_-}{2}\int_{-\frac{\pi}{2}}^{\frac{\pi}{2}}\frac{\sin(2\alpha)}{\sin^2(\alpha)+\sinh^2(\frac{\zeta}{2}+\xi_-)}\rho'(\alpha)\,d\alpha, \tag{5.15}$$

whereas $\langle\sigma_1^z\rangle_{\mathrm{BR}}$ denotes the possible contribution from the boundary root $\alpha_{\mathrm{BR}}^-$ given by

$$\langle\sigma_1^z\rangle_{\mathrm{BR}} = H_{h_-,h_+}\frac{2\pi i\,\sinh^2\xi_-}{1+\delta_{\xi_-,\xi_+}}\rho'(-i(\zeta/2+\xi_-)). \tag{5.16}$$

Here we have introduced the function $H_{h_-,h_+}$ which is 1 when the boundary root $\alpha_{\mathrm{BR}}^-$ belongs to the set of Bethe roots parametrizing the ground state, and 0 otherwise. Note that the presence of the boundary root $\alpha_{\mathrm{BR}}^+$ does not play a direct role here, since it does not correspond to a divergence in the form factor. However, we have seen in section 4.2 that the presence of the boundary root $\alpha_{\mathrm{BR}}^-$ in the set of roots for the ground state depends in fact on the value of *both* boundary magnetic fields, so that the value of the boundary magnetization depends also indirectly on the boundary field $h_+$ at infinity in the thermodynamic limit through the function $H_{h_-,h_+}$ (see Fig. 10 and Fig. 11 for few specific evaluations and for a comparison with numerical data).

For instance, if $|h_+| < h_{\mathrm{cr}}^{(1)}$, then $H_{h_-,h_+} = 0$ if $h_- < h_+$ or if $h_- \in [h_{\mathrm{cr}}^{(1)}, h_{\mathrm{cr}}^{(2)}]$, and $H_{h_-,h_+} = 1$ otherwise. Hence the thermodynamic limit of the boundary magnetization presents, at $h_- = h_+$,

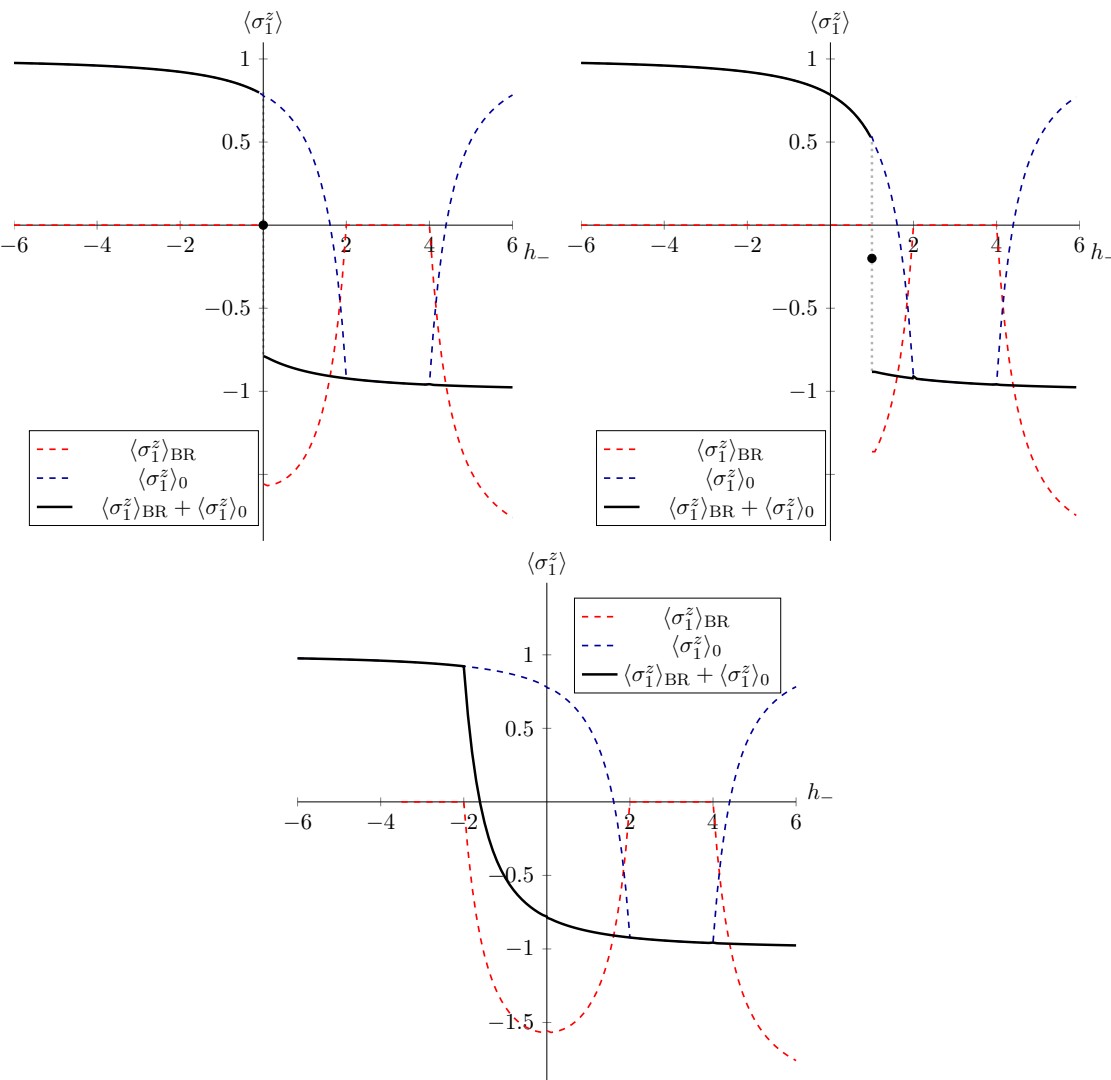

Figure 10: Plot of the magnetization at the left boundary $\langle \sigma_1^z \rangle$ in the thermodynamic limit $L \to \infty$ at $\Delta = 3$ as function of $h_-$ and for different values of $h_+$. The blue dashed line shows the contribution from the bulk rapidities $\langle \sigma_1^z \rangle_0$, while the red dashed line is the contribution from the boundary root $\langle \sigma_1^z \rangle_{\mathrm{BR}}$. The sum of the two is shown as a black line, giving the boundary magnetization. *Above Left*: $h_+ = 0$. The discontinuity at $h_- = 0$ is due to the boundary root moving to the other side of the chain when $h_- < h_+$ (see Fig. 8) and therefore not contributing to the magnetization on the left edge for $h_- < h_+$. *Above Right*: same as the left plot but with $h_+ = 1$, the discontinuity being now at $h_- = 1$. *Below*: same as above but with $h_+ = -3.5$. There is no discontinuity in this case since the ground state does not contain a boundary root when $h_+ = h_-$.

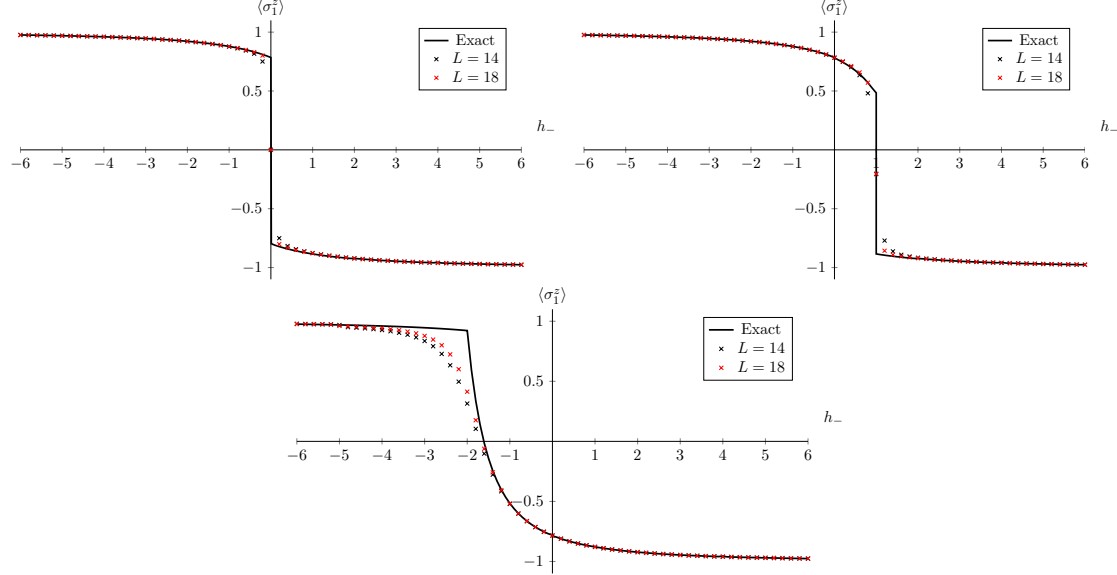

Figure 11: Comparison of the exact result for the magnetization at the boundary (5.14) in the thermodynamic limit, against numerical exact diagonalization results. *Above left*: $h_+ = 0$. *Above right*: $h_+ = 1$. *Below*: $h_+ = -3.5$. For a chain at $\Delta = 3$ and lengths 14 and 18. Note that the discontinuity of the boundary magnetization appears only in the thermodynamic limit, due to the closing of the gap between the ground state and the first excited state at $h_- = h_+$ in this limit, see Fig. 3.

a discontinuity corresponding to the boundary root contribution (5.16):

$$\lim_{h_- - h_+ \to 0^-} \lim_{L \to \infty} \langle \sigma_1^z \rangle - \lim_{h_- - h_+ \to 0^+} \lim_{L \to \infty} \langle \sigma_1^z \rangle = - \lim_{h_- - h_+ \to 0^+} \langle \sigma_1^z \rangle_{\mathrm{BR}}$$

$$= -2 \langle \sigma_1^z \rangle_{\mathrm{BR}} \Big|_{h_- = h_+}$$

$$= -2 i \pi \sinh^2 \xi_- \rho'(-i(\zeta/2 + \xi_-))$$

$$= 2 \prod_{n=1}^{\infty} \frac{\left(1 - q^{2n}\right)^4 \left(1 - e^{4\tilde{\xi}_-} q^{2(2n-1)}\right)\left(1 - e^{-4\tilde{\xi}_-} q^{2(2n-1)}\right)}{\left(1 - q^{2(2n-1)}\right)^2 \left(1 + e^{2\tilde{\xi}_-} q^{2n}\right)^2 \left(1 + e^{-2\tilde{\xi}_-} q^{2n}\right)^2}, \qquad (5.17)$$

which vanishes in the limit $h_+ \to \pm h_{\mathrm{cr}}^{(1)}$. We recall that $q = e^{-\zeta}$, and that the boundary fields are parametrized in this regime $|h_\pm| < h_{\mathrm{cr}}^{(1)}$ as $h_\pm = \sinh \zeta \tanh \tilde{\xi}_\pm$. Note that the difference between taking the limit of equal field and evaluating at exactly the same field is given by the factor $1 + \delta_{\xi_-, \xi_+}$ in the contribution (5.16) from the boundary root. In our convention we indeed have

$$\lim_{h_- - h_+ \to 0} \frac{1}{1 + \delta_{\xi_-, \xi_+}} = 1, \qquad \frac{1}{1 + \delta_{\xi_-, \xi_+}} \Big|_{h_- = h_+} = \frac{1}{2}. \qquad (5.18)$$

If instead $h_+ < -h_{\mathrm{cr}}^{(1)}$, then $H_{h_-, h_+} = 0$ for $h_- < 0$ or $h_- \in [h_{\mathrm{cr}}^{(1)}, h_{\mathrm{cr}}^{(2)}]$, and $H_{h_-, h_+} = 1$ otherwise. In that case the thermodynamic limit of the boundary magnetization is continuous at $h_- = h_+$. By symmetry of the model under the reversal of all spins and change of sign of the boundary fields, this is also the case when $h_+ > h_{\mathrm{cr}}^{(1)}$. In the latter case, we can more precisely use the symmetry relation:

$$\langle \sigma_1^z \rangle \Big|_{h_-, h_+} = -\langle \sigma_1^z \rangle \Big|_{-h_-, -h_+}, \qquad (5.19)$$

in particular when the ground state has negative magnetization (see the cases 6 and 7 of section 4.2).

The integral in (5.15) can be computed by closing the integration contour on the lower half-plane and evaluating the corresponding residues. It gives

$$\langle \sigma_1^z \rangle_0 = -i\pi \sinh^2 \xi_- \, \rho'(-i|\zeta/2 - \tilde{\xi}_-| + \delta_-\pi/2)$$
$$+ \sinh^2 \xi_- \sum_{n=1}^{+\infty} (-1)^n \left[ \frac{1}{\sinh^2(n\zeta + \xi_-)} - \frac{1}{\sinh^2(n\zeta - \xi_-)} \right]. \quad (5.20)$$

It follows in particular from (5.20) that

$$\langle \sigma_1^z \rangle_0 \Big|_{h_-} + \langle \sigma_1^z \rangle_0 \Big|_{-h_-} = -i\pi \sinh^2 \xi_- \Big[ \rho'(-i|\zeta/2 - \tilde{\xi}_-| + \delta_-\pi/2)$$
$$+ \rho'(-i|\zeta/2 + \tilde{\xi}_-| + \delta_-\pi/2) \Big], \quad (5.21)$$

so that the expression (5.14)-(5.16) can in fact be written in the following more compact form, which is valid for all values of the boundary magnetic fields $h_\pm$ (including cases for which the ground state has magnetization $-1$):

$$\lim_{L\to\infty} \langle \sigma_1^z \rangle = \langle \sigma_1^z \rangle_0 + \Theta_{h_-,h_+} \, 2\pi i \, \sinh^2 \xi_- \, \rho'(-i(\zeta/2 + \xi_-)), \quad (5.22)$$

where

$$\Theta_{h_-,h_+} = \begin{cases} 1 & \text{if} \quad \max(-h_{\mathrm{cr}}^{(1)}, h_+) < h_- < h_{\mathrm{cr}}^{(1)} \quad \text{or} \quad h_{\mathrm{cr}}^{(2)} < h_-, \\ \frac{1}{2} & \text{if} \quad h_- = h_+ \quad \text{and} \quad |h_\pm| < h_{\mathrm{cr}}^{(1)}, \\ 0 & \text{otherwise}. \end{cases} \quad (5.23)$$

Notice that, at $h_- = h_+ = 0$ (i.e. for $\xi_+ = \xi_- = i\pi/2$), we have

$$\langle \sigma_1^z \rangle_0 \Big|_{h_-=h_+=0} = -\langle \sigma_1^z \rangle_{\mathrm{BR}} \Big|_{h_-=h_+=0}, \quad (5.24)$$

so that

$$\lim_{L\to\infty} \langle \sigma_1^z \rangle \Big|_{h_-=h_+=0} = 0, \quad (5.25)$$

as it should be. Moreover, due to the factor $\delta_{\xi_-,\xi_+}$ in the contribution (5.16) of the boundary root, we have the relation

$$\lim_{h_-\to 0^\pm} \lim_{h_+\to 0} \lim_{L\to\infty} \langle \sigma_1^z \rangle = \pm \langle \sigma_1^z \rangle_{\mathrm{BR}} \Big|_{h_-=h_+=0}$$
$$= \mp i\pi \rho'(-i\zeta/2 + \pi/2) = \mp \prod_{n=1}^{+\infty} \left( \frac{1-q^{2n}}{1+q^{2n}} \right)^4, \quad (5.26)$$

which corresponds (up to the sign) to the square of the bulk magnetization [22], as already noticed in [20].

## 5.2 The form factor between the two states of lowest energy for $|h_\pm| < h_{\mathrm{cr}}^{(1)}$

We now consider the form factor of the $\sigma_1^z$ operator between the two states of lowest energy in the regime $|h_\pm| < h_{\mathrm{cr}}^{(1)}$. Since in this regime these two states are separated by a gap from the (continuum of the) other excited states in the thermodynamic limit, this form factor gives the only possible non-zero contribution to the large-time limit of the connected boundary autocorrelation function $\langle \sigma_1^z(t) \sigma_1^z \rangle_{T=0}^c$.

### 5.2.1 The case $h_- = h_+$

We here work directly in the regime $h_- = h_+ = h$ (namely $\xi_- = \xi_+ = \xi$) and we write the form factor between the two quasi-degenerate ground states as

$$\langle \mathrm{GS}_1, h \,|\, \sigma_1^z \,|\, \mathrm{GS}_2, h \rangle = \frac{\langle \{\alpha^+\} \,|\, \sigma_1^z \,|\, \{\alpha^-\}\rangle}{\langle \{\alpha^+\} \,|\, \{\alpha^+\}\rangle^{1/2} \langle \{\alpha^-\} \,|\, \{\alpha^-\}\rangle^{1/2}},$$

$$= \left( \frac{\langle \{\alpha^+\} \,|\, \{\alpha^+\}\rangle}{\langle \{\alpha^-\} \,|\, \{\alpha^-\}\rangle} \right)^{1/2} \frac{\langle \{\alpha^+\} \,|\, \sigma_1^z \,|\, \{\alpha^-\}\rangle}{\langle \{\alpha^+\} \,|\, \{\alpha^+\}\rangle}, \tag{5.27}$$

which can be expressed by means of (3.13) and (3.21). In (5.27), $\{\alpha_+\}$ and $\{\alpha_-\}$ denote the two sets of Bethe roots associated with the two quasi-degenerate ground states identified in subsection 4.3.2.

Let us first consider the first ratio. We recall that the Bethe roots of the two states only differ by exponentially small corrections in $L$,

$$\alpha_j^+ - \alpha_j^- = O(L^{-\infty}), \qquad 1 \le j \le N, \tag{5.28}$$

so that most of the prefactors in (3.21) simplify up to exponentially small corrections in $L$:

$$\frac{\langle \{\alpha^+\} \,|\, \{\alpha^+\}\rangle}{\langle \{\alpha^-\} \,|\, \{\alpha^-\}\rangle} = \frac{\sin(\alpha_N^+ + i\xi + i\frac{\zeta}{2})}{\sin(\alpha_N^- + i\xi + i\frac{\zeta}{2})} \frac{\det_N \left[ \mathcal{M}(\boldsymbol{\alpha}^+, \boldsymbol{\alpha}^+) \right]}{\det_N \left[ \mathcal{M}(\boldsymbol{\alpha}^-, \boldsymbol{\alpha}^-) \right]} + O(L^{-\infty}),$$

$$= -\frac{\det_N \left[ \mathcal{M}(\boldsymbol{\alpha}^+, \boldsymbol{\alpha}^+) \right]}{\det_N \left[ \mathcal{M}(\boldsymbol{\alpha}^-, \boldsymbol{\alpha}^-) \right]} + O(L^{-\infty}). \tag{5.29}$$

Here we have explicitly used that the two boundary complex roots $\alpha_N^\pm \equiv \alpha_{\mathrm{BR}}^\pm$ are of the form

$$\alpha_N^\pm = -i(\zeta/2 + \xi + \epsilon_\pm) \qquad \text{with} \quad \epsilon_\pm = \pm \epsilon \left( 1 + O(L^{-\infty}) \right), \tag{5.30}$$

see (4.50). Moreover, it follows from (3.22) that

$$\left[ \mathcal{M}(\boldsymbol{\alpha}^+, \boldsymbol{\alpha}^+) \right]_{jk} = \left[ \mathcal{M}(\boldsymbol{\alpha}^-, \boldsymbol{\alpha}^-) \right]_{jk} + O(L^{-\infty}), \tag{5.31}$$

for each row such that $\alpha_j^\pm$ are real roots, i.e. for $1 \le j \le N-1$. The $N$-th row has to be treated separately since in that case the complex root $\alpha_N^\pm$ approaches, with an exponentially small deviation $\epsilon_\pm \sim \pm\epsilon$, the double pole of the function $g'$ (3.24) so that the corresponding diagonal coefficient is exponentially diverging with $L$, see (5.12), and we have

$$\left[ \mathcal{M}(\boldsymbol{\alpha}^+, \boldsymbol{\alpha}^+) \right]_{NN} = -\left[ \mathcal{M}(\boldsymbol{\alpha}^-, \boldsymbol{\alpha}^-) \right]_{NN} \left( 1 + O(L^{-\infty}) \right), \tag{5.32}$$

whereas the off-diagonal coefficients $\left[ \mathcal{M}(\boldsymbol{\alpha}^\pm, \boldsymbol{\alpha}^\pm) \right]_{Nk}$ with $k \neq N$ remain finite (and therefore are exponentially subleading with respect to (5.32)). Finally, we obtain from (5.29), (5.31) and (5.32) that

$$\frac{\langle \{\alpha^+\} \,|\, \{\alpha^+\}\rangle}{\langle \{\alpha^-\} \,|\, \{\alpha^-\}\rangle} = 1 + O(L^{-\infty}). \tag{5.33}$$

Let us now consider the second ratio in (5.27). Using again (5.28) so as to simplify the prefactors, and the fact that $\mathcal{P}$ is a rank-one matrix so as to decompose the determinant in the numerator, we obtain that

$$\frac{\langle \{\alpha^+\} \,|\, \sigma_1^z \,|\, \{\alpha^-\}\rangle}{\langle \{\alpha^+\} \,|\, \{\alpha^+\}\rangle} = \frac{\det_N \left[ \mathcal{M}(\boldsymbol{\alpha}^+, \boldsymbol{\alpha}^-) - 2\mathcal{P}(\boldsymbol{\alpha}^+, \boldsymbol{\alpha}^-) \right]}{\det_N \left[ \mathcal{M}(\boldsymbol{\alpha}^+, \boldsymbol{\alpha}^+) \right]} + O(L^{-\infty})$$

$$= \frac{\det_N \left[ \mathcal{M}(\boldsymbol{\alpha}^+, \boldsymbol{\alpha}^-) \right]}{\det_N \left[ \mathcal{M}(\boldsymbol{\alpha}^+, \boldsymbol{\alpha}^+) \right]} - 2 \sum_{\ell=1}^{N} \frac{\det_N \left[ \widetilde{\mathcal{M}}^{(\ell)}(\boldsymbol{\alpha}^+, \boldsymbol{\alpha}^-) \right]}{\det_N \left[ \mathcal{M}(\boldsymbol{\alpha}^+, \boldsymbol{\alpha}^+) \right]}$$

$$+ O(L^{-\infty}), \tag{5.34}$$

where

$$\big[\widetilde{\mathcal{M}}^{(\ell)}(\boldsymbol{\alpha}^+, \boldsymbol{\alpha}^-)\big]_{jk} = \big[\mathcal{M}(\boldsymbol{\alpha}^+, \boldsymbol{\alpha}^-)\big]_{jk} \quad \text{if} \quad k \neq \ell, \tag{5.35}$$

$$\big[\widetilde{\mathcal{M}}^{(\ell)}(\boldsymbol{\alpha}^+, \boldsymbol{\alpha}^-)\big]_{j\ell} = \big[\mathcal{P}(\boldsymbol{\alpha}^+, \boldsymbol{\alpha}^-)\big]_{j\ell}. \tag{5.36}$$

Note that, from the orthogonality property of two different Bethe states, the first term in (5.34) should in fact vanish. Moreover, it follows from (5.28) and from (3.18) that

$$\big[\mathcal{P}(\boldsymbol{\alpha}^+, \boldsymbol{\alpha}^-)\big]_{jk} = \big[\mathcal{P}(\boldsymbol{\alpha}^-, \boldsymbol{\alpha}^-)\big]_{jk}\big(1 + O(L^{-\infty})\big) = p''(\alpha_j^+)v(\alpha_k^-)\big(1 + O(L^{-\infty})\big), \tag{5.37}$$

in which we have used the notations of (5.2), and from (5.28) and (3.17) that

$$\big[\mathcal{M}(\boldsymbol{\alpha}^+, \boldsymbol{\alpha}^-)\big]_{jk} = i\,\delta_{jk} \frac{\mathfrak{a}(\alpha_j^-|\{\alpha^+\}) - \mathfrak{a}(\alpha_j^+|\{\alpha^+\})}{\alpha_j^- - \alpha_j^+}\big(1 + O(L^{-\infty})\big)$$
$$- 2\pi\big[K(\alpha_j^- - \alpha_k^-) - K(\alpha_j^- + \alpha_k^-)\big]. \tag{5.38}$$

If $\alpha_j^{\pm}$ are real roots, i.e. for $j < N$, we therefore obtain that

$$\big[\mathcal{M}(\boldsymbol{\alpha}^+, \boldsymbol{\alpha}^-)\big]_{jk} = i\,\delta_{jk}\,\mathfrak{a}'(\alpha_j^+|\{\alpha^+\})\big(1 + O(L^{-\infty})\big)$$
$$- 2\pi\big[K(\alpha_j^- - \alpha_k^-) - K(\alpha_j^- + \alpha_k^-)\big]$$
$$= \big[\mathcal{M}(\boldsymbol{\alpha}^+, \boldsymbol{\alpha}^+)\big]_{jk} + O(L^{-\infty}), \tag{5.39}$$

so that we recover for the first $N-1$ rows the elements of the Gaudin matrix (3.22) up to exponentially small corrections in $L$. The row $j = N$ has to be treated separately since the two complex roots $\alpha_N^{\pm}$ are, in the leading order, symmetrically distributed around a zero of the function $\mathfrak{a}$ (see (5.30)). In that case,

$$\frac{\mathfrak{a}(\alpha_N^-|\{\alpha_\ell^+\}) - \mathfrak{a}(\alpha_N^+|\{\alpha_\ell^+\})}{\alpha_N^- - \alpha_N^+} = \epsilon^2 \frac{\widetilde{\mathfrak{a}}(\alpha_N^-|\{\alpha^+\}) - \widetilde{\mathfrak{a}}(\alpha_N^+|\{\alpha^+\})}{-2\epsilon}\big(1 + O(L^{-\infty})\big)$$
$$= \epsilon^2 \frac{\widetilde{\mathfrak{a}}'(\alpha_N^+|\{\alpha^+\})}{\widetilde{\mathfrak{a}}(\alpha_N^+|\{\alpha^+\})}\widetilde{\mathfrak{a}}(\alpha_N^+|\{\alpha^+\})\big(1 + O(L^{-\infty})\big),$$

in which we have used that $\alpha_N^- - \alpha_N^+ = -2\epsilon\big(1 + O(L^{-\infty})\big)$ and defined the regularized function:

$$\widetilde{\mathfrak{a}}(\alpha|\{\alpha^+\}) = \left(\frac{\sin(\alpha - i\zeta/2)}{\sin(\alpha + i\zeta/2)}\right)^{2L}\left(\frac{-i}{\sin(\alpha - i\xi - i\zeta/2)}\right)^2$$
$$\times \frac{\sin(i\zeta - 2\alpha)}{\sin(i\zeta + 2\alpha)}\prod_{k=1}^{N}\frac{\mathfrak{s}(\alpha + i\zeta, \alpha_k^+)}{\mathfrak{s}(\alpha - i\zeta, \alpha_k^+)}. \tag{5.40}$$

Now we use again the Bethe equations for $\alpha_N^+$, which state that

$$\epsilon^2\,\widetilde{\mathfrak{a}}(\alpha_N^+|\{\alpha^+\}) = 1 + O(L^{-\infty}). \tag{5.41}$$

Hence we obtain

$$\big[\mathcal{M}(\boldsymbol{\alpha}^+, \boldsymbol{\alpha}^-)\big]_{Nk} = i\,\delta_{Nk}\frac{\widetilde{\mathfrak{a}}'(\alpha_N^+|\{\alpha^+\})}{\widetilde{\mathfrak{a}}(\alpha_N^+|\{\alpha^+\})}\big(1 + O(L^{-\infty})\big)$$
$$- 2\pi\big[K(\alpha_N^- - \alpha_k^-) - K(\alpha_N^- + \alpha_k^-)\big]$$
$$= -\delta_{Nk}\Bigg\{2Lp'(\alpha_N^-) + \widetilde{g}'(\alpha_N^-) - 2\theta'(2\alpha_N^-)$$
$$+ \sum_{k=1}^{N}\big[\theta'(\alpha_N^- - \alpha_k^-) + \theta'(\alpha_N^- + \alpha_k^-)\big]\Bigg\}\big(1 + O(L^{-\infty})\big)$$
$$- 2\pi\big[K(\alpha_N^- - \alpha_k^-) - K(\alpha_N^- + \alpha_k^-)\big], \tag{5.42}$$

in which we have defined

$$\widetilde{g}'(\alpha) = 2i\,\frac{\cos(\alpha - i\xi - i\zeta/2)}{\sin(\alpha - i\xi - i\zeta/2)}. \tag{5.43}$$

Notice that, contrary to what happens for the Gaudin matrix $\mathcal{M}(\boldsymbol{\alpha}^+, \boldsymbol{\alpha}^+)$ in the denominator of (5.34) (see (5.12)), there is no singularity in this last row associated with the complex root. Hence, in (5.34), all terms but the one with $\ell = N$ vanish as $\epsilon$ (i.e. exponentially fast with $L$) in the large $L$ limit due to the fact that $\det_N\big[\mathcal{M}(\boldsymbol{\alpha}^+, \boldsymbol{\alpha}^+)\big]$ diverges as $1/\epsilon$. The only term in the sum (5.34) which does not vanish is the term with $\ell = N$, since the corresponding matrix elements of $\mathcal{P}(\boldsymbol{\alpha}^+, \boldsymbol{\alpha}^-)$ themselves diverge as $1/\epsilon$. Therefore

$$
\begin{aligned}
\frac{\langle\{\alpha^+\}|\sigma_1^z|\{\alpha^-\}\rangle}{\langle\{\alpha^+\}|\{\alpha^+\}\rangle} &= -2 \det_N\big[\mathcal{M}(\boldsymbol{\alpha}^+, \boldsymbol{\alpha}^+)^{-1} \cdot \widetilde{\mathcal{M}}^{(N)}(\boldsymbol{\alpha}^+, \boldsymbol{\alpha}^-)\big] + O(L^{-\infty}) \\
&= -2 \sum_{k=1}^{N}\big[\mathcal{M}(\boldsymbol{\alpha}^+, \boldsymbol{\alpha}^+)^{-1}\big]_{Nk}\big[\mathcal{P}(\boldsymbol{\alpha}^+, \boldsymbol{\alpha}^-)\big]_{kN} + O(L^{-\infty}) \\
&= -2 \sum_{k=1}^{N}\big[\mathcal{M}(\boldsymbol{\alpha}^+, \boldsymbol{\alpha}^+)^{-1}\big]_{Nk} p''(\alpha_k^+)\mathsf{v}(\alpha_N^-) + O(L^{-\infty}) \\
&= -2\,\mathsf{u}(\alpha_N^+)\mathsf{v}(\alpha_N^-) + O(L^{-\infty}),
\end{aligned} \tag{5.44}
$$

in which

$$\mathsf{u}(\alpha_N^+) \underset{L\to+\infty}{\sim} -\frac{\epsilon\,\pi}{2}\,\rho'(-i\zeta/2 - i\xi), \tag{5.45}$$

$$\mathsf{v}(\alpha_N^-) \underset{L\to+\infty}{\sim} -i\,\frac{\sinh^2\xi}{\epsilon}, \tag{5.46}$$

so that

$$\frac{\langle\{\alpha^+\}|\sigma_1^z|\{\alpha^-\}\rangle}{\langle\{\alpha^+\}|\{\alpha^+\}\rangle} \underset{L\to+\infty}{\sim} -\pi i \sinh^2\xi\,\rho'(-i\zeta/2 - i\xi). \tag{5.47}$$

Note that is equal (up to the sign) to the contribution $\langle\sigma_1^z\rangle_{\mathrm{BR}}$ to the boundary magnetization from the boundary root when $\xi_- = \xi_+ = \xi$, see eq. (5.16).

Finally,

$$
\begin{aligned}
\lim_{L\to\infty}\langle \mathrm{GS}_1, h\,|\,\sigma_1^z\,|\,\mathrm{GS}_2, h\rangle &= -\pi i \sinh^2\xi\,\rho'(-i\zeta/2 - i\xi) = -\langle\sigma_1^z\rangle_{\mathrm{BR}}\Big|_{\xi_+=\xi_-=\xi} \\
&= \prod_{n=1}^{\infty}\frac{\big(1-q^{2n}\big)^4\big(1-e^{4\bar\xi_-}q^{2(2n-1)}\big)\big(1-e^{-4\bar\xi_-}q^{2(2n-1)}\big)}{\big(1-q^{2(2n-1)}\big)^2\big(1+e^{2\bar\xi_-}q^{2n}\big)^2\big(1+e^{-2\bar\xi_-}q^{2n}\big)^2},
\end{aligned} \tag{5.48}
$$

which is exactly half of the discontinuity of the boundary magnetization at $h_+ = h_-$, see (5.17). In Fig. 12 and Fig. 13 we report this result at $h = 0$ and $h = 1$ and as function of $\Delta$ compared to the values of the form factors in a finite size chain.

### 5.2.2 The case $h_- \neq h_+$

As soon as the two boundary fields are different, the degeneracy of the ground state is broken and the two states with $\frac{L}{2} - 1$ real roots and one boundary root have different energy (see subsection 4.3.1). We show here that the form factor between these two states decays exponentially with the system size $L$, so that the thermodynamic limit of the boundary autocorrelation function (1.8) effectively vanishes in the large time limit.

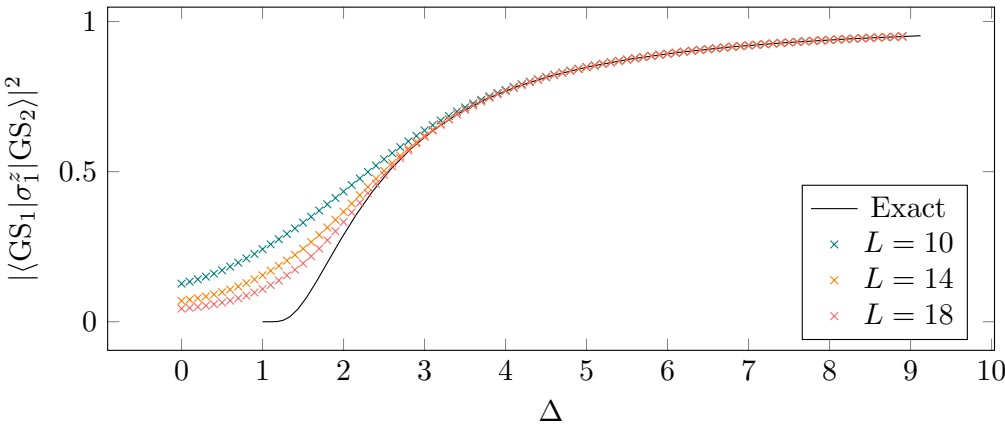

Figure 12: Form Factor between the two degenerate ground states at $h_- = h_+ = 0$ in the thermodynamic limit with respect to the anisotropy $\Delta$ from equation (5.48), given by the closed formula $s_0^4 = \prod_{n=1}^{\infty} \left( \frac{1-e^{-2n\zeta}}{1+e^{-2n\zeta}} \right)^8$, compared to its value at finite size obtained by numerical exact diagonalization.

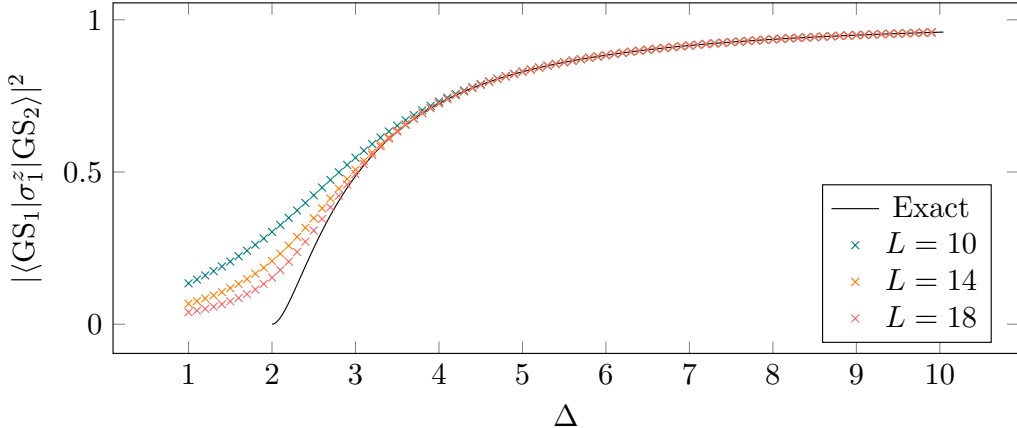

Figure 13: Form Factor between the two degenerate ground states at $h_- = h_+ = 1$ in the thermodynamic limit with respect to the anisotropy $\Delta$ from equation (5.48) compared to its value at finite size obtained by numerical exact diagonalization.

It is more convenient to consider the square of the form factor,

$$\frac{\langle \{\alpha^+\} | \sigma_1^z | \{\alpha^-\} \rangle \langle \{\alpha^-\} | \sigma_1^z | \{\alpha^+\} \rangle}{\langle \{\alpha^-\} | \{\alpha^-\} \rangle \langle \{\alpha^+\} | \{\alpha^+\} \rangle} = \frac{\det_N \left[ \mathcal{M}(\boldsymbol{\alpha}^+, \boldsymbol{\alpha}^-) - 2\mathcal{P}(\boldsymbol{\alpha}^+, \boldsymbol{\alpha}^-) \right]}{\det_N \left[ \mathcal{M}(\boldsymbol{\alpha}^-, \boldsymbol{\alpha}^-) \right]}$$
$$\times \frac{\det_N \left[ \mathcal{M}(\boldsymbol{\alpha}^-, \boldsymbol{\alpha}^+) - 2\mathcal{P}(\boldsymbol{\alpha}^-, \boldsymbol{\alpha}^+) \right]}{\det_N \left[ \mathcal{M}(\boldsymbol{\alpha}^+, \boldsymbol{\alpha}^+) \right]}, \quad (5.49)$$

which enters the expression for the spin auto-correlation function. Here, $\{\alpha_+\}$ and $\{\alpha_-\}$ denote the two sets of Bethe roots associated with the two states of lowest energy identified in subsection 4.3.1.

As previously, we use the fact that $\mathcal{P}(\boldsymbol{\alpha}^{-\sigma}, \boldsymbol{\alpha}^{\sigma})$ for $\sigma \in \{+, -\}$ is a rank-one matrix to write

$$
\frac{\det_N \left[ \mathcal{M}(\boldsymbol{\alpha}^{-\sigma}, \boldsymbol{\alpha}^{\sigma}) - 2\mathcal{P}(\boldsymbol{\alpha}^{-\sigma}, \boldsymbol{\alpha}^{\sigma}) \right]}{\det_N \left[ \mathcal{M}(\boldsymbol{\alpha}^{\sigma}, \boldsymbol{\alpha}^{\sigma}) \right]}
$$

$$
= \frac{\det_N \left[ \mathcal{M}(\boldsymbol{\alpha}^{-\sigma}, \boldsymbol{\alpha}^{\sigma}) \right]}{\det_N \left[ \mathcal{M}(\boldsymbol{\alpha}^{\sigma}, \boldsymbol{\alpha}^{\sigma}) \right]} - 2 \sum_{\ell=1}^N \frac{\det_N \left[ \widetilde{\mathcal{M}}^{(\ell)}(\boldsymbol{\alpha}^{-\sigma}, \boldsymbol{\alpha}^{\sigma}) \right]}{\det_N \left[ \mathcal{M}(\boldsymbol{\alpha}^{\sigma}, \boldsymbol{\alpha}^{\sigma}) \right]}, \quad (5.50)
$$

where

$$
\left[ \widetilde{\mathcal{M}}^{(\ell)}(\boldsymbol{\alpha}^{-\sigma}, \boldsymbol{\alpha}^{\sigma}) \right]_{jk} = \left[ \mathcal{M}(\boldsymbol{\alpha}^{-\sigma}, \boldsymbol{\alpha}^{\sigma}) \right]_{jk} \quad \text{if} \quad k \neq \ell, \quad (5.51)
$$

$$
\left[ \widetilde{\mathcal{M}}^{(\ell)}(\boldsymbol{\alpha}^{-\sigma}, \boldsymbol{\alpha}^{\sigma}) \right]_{j\ell} = \left[ \mathcal{P}(\boldsymbol{\alpha}^{-\sigma}, \boldsymbol{\alpha}^{\sigma}) \right]_{j\ell}, \quad (5.52)
$$

with the first term in the sum (5.50) vanishing due to the orthogonality property of two different Bethe states.

We need to evaluate the order of the different determinants appearing in (5.50). We recall that

$$
[\mathcal{M}(\boldsymbol{\alpha}^{\sigma}, \boldsymbol{\alpha}^{\sigma})]_{jk} = -2L\, \widehat{\xi}'_{\sigma}(\alpha_j^{\sigma}) \left\{ \delta_{jk} + \frac{\pi}{L} \frac{K(\alpha_j^{\sigma} - \alpha_k^{\sigma}) - K(\alpha_j^{\sigma} + \alpha_k^{\sigma})}{\widehat{\xi}'_{\sigma}(\alpha_j^{\sigma})} \right\}
$$

$$
+ O(L^{-\infty}), \qquad j \neq N, \quad (5.53)
$$

$$
[\mathcal{M}(\boldsymbol{\alpha}^{\sigma}, \boldsymbol{\alpha}^{\sigma})]_{Nk} = -\frac{1}{\epsilon_{\sigma}} \left[ (1 + \delta_{\xi_-, \xi_+}) \delta_{Nk} + O(\epsilon_{\sigma}) \right], \quad (5.54)
$$

so that the determinant of $\mathcal{M}(\boldsymbol{\alpha}^{\sigma}, \boldsymbol{\alpha}^{\sigma})$ is of order $\frac{L^{N-1}}{\epsilon_{\sigma}}$ in the large $L$ limit.

Let us now determine the behavior of the matrix elements of $\mathcal{M}(\boldsymbol{\alpha}^{-\sigma}, \boldsymbol{\alpha}^{\sigma})$, which is given by the expression (3.17). Its diagonal part reads

$$
i \sin(2\alpha_j^{\sigma}) \frac{\prod_{\ell \neq j} \mathfrak{s}(\alpha_j^{\sigma}, \alpha_\ell^{\sigma})}{\prod_{\ell=1}^N \mathfrak{s}(\alpha_j^{\sigma}, \alpha_\ell^{-\sigma})} \prod_{\ell=1}^N \frac{\mathfrak{s}(\alpha_j^{\sigma} - i\zeta, \alpha_\ell^{-\sigma})}{\mathfrak{s}(\alpha_j^{\sigma} - i\zeta, \alpha_\ell^{\sigma})} \left[ \mathfrak{a}(\alpha_j^{\sigma} | \{\alpha^{-\sigma}\}) - 1 \right]
$$

$$
= \frac{iL \sin(2\alpha_j^{\sigma})}{\phi_j^{\sigma}} \left[ \phi_+^{\sigma}(\alpha_j^{\sigma}) - \phi_-^{\sigma}(\alpha_j^{\sigma}) \right], \quad (5.55)
$$

in which we have defined, similarly as in [47],

$$
\phi_\pm^{\sigma}(\mu) = \prod_{\ell=1}^N \frac{\mathfrak{s}(\mu \pm i\zeta, \alpha_\ell^{-\sigma})}{\mathfrak{s}(\mu \pm i\zeta, \alpha_\ell^{\sigma})}, \qquad \phi_j^{\sigma} = L \frac{\prod_{\ell=1}^N \mathfrak{s}(\alpha_j^{\sigma}, \alpha_\ell^{-\sigma})}{\prod_{\ell \neq j}^N \mathfrak{s}(\alpha_j^{\sigma}, \alpha_\ell^{\sigma})}. \quad (5.56)
$$

To evaluate (5.55), it is convenient to separate the contribution of the complex root from the contribution of the real roots. We therefore also define

$$
\widetilde{\phi}_\pm^{\sigma}(\mu) = \prod_{\ell=1}^{N-1} \frac{\mathfrak{s}(\mu \pm i\zeta, \alpha_\ell^{-\sigma})}{\mathfrak{s}(\mu \pm i\zeta, \alpha_\ell^{\sigma})}, \quad (5.57)
$$

$$
\widetilde{\phi}_j^{\sigma} = L \frac{\prod_{\ell=1}^{N-1} \mathfrak{s}(\alpha_j^{\sigma}, \alpha_\ell^{-\sigma})}{\prod_{\ell \neq j}^{N-1} \mathfrak{s}(\alpha_j^{\sigma}, \alpha_\ell^{\sigma})} = \widetilde{\phi}^{\sigma}(\alpha_j^{\sigma}) \quad \text{if } j \neq N, \quad (5.58)
$$

with

$$
\widetilde{\phi}^{\sigma}(\mu) = \sin(2\mu) \prod_{\ell=1}^{N-1} \frac{\mathfrak{s}(\mu, \alpha_\ell^{-\sigma})}{\mathfrak{s}(\mu, \alpha_\ell^{\sigma})} \frac{\mathfrak{a}(\mu | \{\alpha^{\sigma}\}) - \mathfrak{a}(-\mu | \{\alpha^{\sigma}\})}{4i\, \widehat{\xi}'_{\sigma}(\mu)}. \quad (5.59)
$$

Since the shift between the $\alpha_j^\sigma$ and $\alpha_j^{-\sigma}$ is of order $1/L$ for $j = 1,\ldots,N-1$ (see (4.45) for a more precise expression) and of order 1 for $j = N$, it is easy to see that the functions $\widetilde{\phi}_\pm^\sigma(\mu)$ (and therefore $\phi_\pm^\sigma(\mu)$) remain finite: their limiting value can be computed by means of the general relation

$$\prod_{j=1}^{N-1} \frac{f(\alpha_j^-)}{f(\alpha_j^+)} \underset{L \to +\infty}{\longrightarrow} \exp\left\{ \frac{1}{2\pi} \int_{-\frac{\pi}{2}}^{\frac{\pi}{2}} \left[ \widehat{\xi}_{\alpha_{\mathrm{BR}}^+}(\beta) - \widehat{\xi}_{\alpha_{\mathrm{BR}}^-}(\beta) \right] \frac{f'(\beta)}{f(\beta)} \, d\beta \right\}, \qquad (5.60)$$

which can be derived from (4.45) for any regular function $f$. As for $\widetilde{\phi}_j^\sigma$, it can be computed for $j < N$ by means of the relation

$$\frac{1}{L} \sum_{k=1}^{N-1} \frac{\widetilde{\phi}_k^\sigma}{\mathfrak{s}(\alpha_k^\sigma, \alpha_j^{-\sigma} \pm i\zeta) \, \mathfrak{s}(\alpha_k^\sigma, \alpha_j^{-\sigma})} = \frac{\left[ \widetilde{\phi}_\pm^{-\sigma}(\alpha_j^{-\sigma}) \right]^{-1}}{\mathfrak{s}(\alpha_j^{-\sigma} \pm i\zeta, \alpha_j^{-\sigma})}, \qquad (5.61)$$

which, thanks to the fact that the function $\widetilde{\phi}^\sigma$ is an even $\pi$-periodic function, can be transformed into the following integral relation:

$$\frac{1}{2\pi} \int_{-\frac{\pi}{2}}^{\frac{\pi}{2}} \frac{\widetilde{\phi}^\sigma(\mu) \, \widehat{\xi}_\sigma'(\mu)}{\mathfrak{s}(\mu, \alpha_j^{-\sigma} \pm i\zeta) \, \mathfrak{s}(\mu, \alpha_j^{-\sigma})} \, d\mu = \frac{\left[ \widetilde{\phi}_\pm^{-\sigma}(\alpha_j^{-\sigma}) \right]^{-1}}{\mathfrak{s}(\alpha_j^{-\sigma} \pm i\zeta, \alpha_j^{-\sigma})} + O(1/L). \qquad (5.62)$$

It is however unnecessary for our purpose to compute precisely this quantity, we just need to know that it remains finite (and so does $\phi_j^\sigma$ for $j < N$), which is clear from the above equation. It is also easy to see that $\phi_N^\sigma$ is of order $L$. Hence, the diagonal element (5.55) is of order $L$ except for $j = N$ for which it remains finite.

Finally, it is also easy to see that the matrix elements $\left[ \mathcal{P}(\boldsymbol{\alpha}^{-\sigma}, \boldsymbol{\alpha}^\sigma) \right]_{jk}$ all remain finite, except for $\sigma = -$ and $k = N$ since $\left[ \mathcal{P}(\boldsymbol{\alpha}^+, \boldsymbol{\alpha}^-) \right]_{jN}$ diverges as $1/\epsilon_-$.

Therefore, all terms with $\ell < N$ in the sum (5.50) vanish exponentially fast with $L$ in the large $L$ limit, due to the extra divergence in $1/\epsilon_\sigma$ of the Gaudin determinant $\det_N \left[ \mathcal{M}(\boldsymbol{\alpha}^\sigma, \boldsymbol{\alpha}^\sigma) \right]$ in the denominator with respect to the numerator $\det_N \left[ \widetilde{\mathcal{M}}^{(\ell)}(\boldsymbol{\alpha}^{-\sigma}, \boldsymbol{\alpha}^\sigma) \right]$. The only term that does not vanish is the one with $\ell = N$ and for $\sigma = -$, since the corresponding matrix elements of $\mathcal{P}(\boldsymbol{\alpha}^+, \boldsymbol{\alpha}^-)$ also diverges as $1/\epsilon_-$, which compensates the divergence in the denominator. However, if $\sigma = +$, the extra divergence in $1/\epsilon_+$ in the denominator is not compensated even in the last term of (5.50), so that the product (5.49) vanishes as $\epsilon_+$, i.e. exponentially fast with $L$.

## 6 The odd-length open chain

Until now, we have only considered open chains with an even number of sites $L$. In this section, we briefly underline what changes in the case of a chain with an odd number of sites.

The description of the ground state and its degeneracies are in this case very different. The Bethe eigenstates (and therefore the ground state(s)) of a chain of odd length $L$ always have a finite magnetization. Moreover, we no longer have quasi-degenerate ground states for $h_+ = h_- \neq 0$; instead, due to the spin-flip symmetry, there exists an *exact* degeneracy of the whole spectrum at $h_+ = -h_-$, but the two ground states are in this case in different magnetization sectors (see Figure 14). Hence, the change of parity of the length of the chain has some drastic effect on the microscopic description of the spectrum. We can nevertheless expect to observe a similar behavior in the thermodynamic limit for the chain with $L$ odd and antiparallel boundary fields and for the chain with $L$ even and parallel boundary fields.

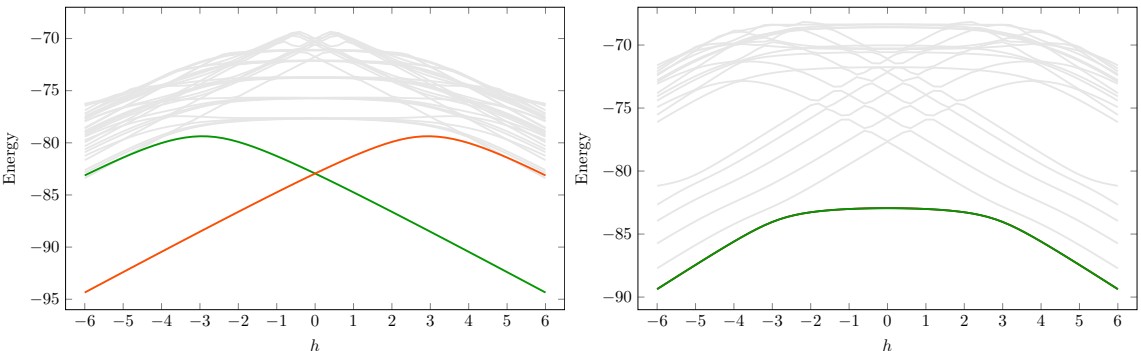

Figure 14: Lowest part of the spectrum from numerical exact diagonalization for a chain of odd length. *Left*: Parallel boundary fields ($h = h_\pm$). The red line has $m = +1/2$, the green one, $m = -1/2$. *Right*: Anti-parallel boundary fields ($h = h_- = -h_+$), where *all* states are doubly degenerate. For a chain of $\Delta = 4$ and $L = 11$.

## 6.1 Configuration of Bethe roots in the ground state

It is easy to repeat in the odd length case the classification that we have done in section 4.2 in the even length case.

Let us first suppose that $h_+ + h_- < 0$, so as to ensure that the ground state is in a sector of positive magnetization. Combining again the results of Appendix A with those of section 4.1, we have to distinguish the following different cases:

**Case 1:** $h_\pm < h_{\mathrm{cr}}^{(1)}$ with $h_+ + h_- < 0$.

It follows from the study of Appendix A (see Case A) that the number of vacancies for real Bethe roots in a given sector $N = \frac{L-1-2n}{2}$ ($n \in \mathbb{N}$) is $\frac{L-1}{2} + n$. Hence the ground state is the state, in the sector $N = \frac{L-1}{2}$ of magnetization $+1/2$, which is given by $N = \frac{L-1}{2}$ real Bethe roots (no hole).

**Case 2:** $h_\sigma > h_{\mathrm{cr}}^{(2)}$ and $h_{-\sigma} < -h_{\mathrm{cr}}^{(2)}$ ($\sigma \in \{+,-\}$) with $h_+ + h_- < 0$.

This case still corresponds to Case A of Appendix A. Hence, the ground state is in the sector $N = \frac{L-1}{2}$ of magnetization $+1/2$. It is given by $N-1$ real Bethe roots, one hole, and the boundary root $\alpha_{\mathrm{BR}}^{\sigma}$.

**Case 3:** $h_\sigma \in (h_{\mathrm{cr}}^{(1)}, h_{\mathrm{cr}}^{(2)})$ and $h_{-\sigma} < -h_{\mathrm{cr}}^{(1)}$ ($\sigma \in \{+,-\}$) with $h_+ + h_- < 0$.

This case corresponds to Case B of Appendix A. The ground state is still in the sector $N = \frac{L-1}{2}$ of magnetization $+1/2$. It is given by $N = \frac{L-1}{2}$ real Bethe roots and one hole.

It follows from the previous study that, in the thermodynamic limit, the ground state is separated by a gap of energy from the excited states in Case 1 and not in Cases 2 and 3.

All cases with $h_+ + h_- > 0$ can be obtained from the above cases by symmetry, using the invariance of the model under the reversal of all spins together with a change of sign of the boundary fields $h_\pm$. The ground state is in the sector of magnetization $-1/2$ and hence is beyond the equator.

Due to this symmetry, the spectrum of the finite-size model is doubly degenerate when $h_+ + h_- = 0$. The two ground states are in two different magnetization sectors ($+1/2$ and $-1/2$).

## 6.2 Boundary magnetization in the ground state

If $h_+ + h_- < 0$, i.e. $h_- < -h_+$, the boundary magnetization in the ground state is

$$\langle \sigma_1^z \rangle = \langle GS_+ | \sigma_1^z | GS_+ \rangle, \tag{6.1}$$

where $|GS_+\rangle$ is the normalized ground state with magnetization $+1/2$ which is described in Cases 1,2 and 3 above. The boundary magnetization is therefore in this case still given in the thermodynamic limit by the formulas (5.14), (5.15) and (5.16), the only difference being in the value of the factor $H_{h_-,h_+}$, i.e. in the dependance of the presence of the boundary root $\alpha_{BR}^-$ in the set of Bethe roots for the ground state with respect to the boundary fields $h_\pm$. In the present case, $H_{h_-,h_+} = 1$ only if $h_- > h_{cr}^{(2)}$, which may happen only if $h_+ < -h_{cr}^{(2)}$ (so that the condition $h_+ + h_- < 0$ is still satisfied).

One can obtain the value of the boundary magnetization in the case $h_- > -h_+$, by symmetry from the previous case by means of formula (5.19):

$$\langle \sigma_1^z \rangle \Big|_{\substack{h_-,h_+ \\ h_->-h_+}} = \langle GS_- | \sigma_1^z | GS_- \rangle \Big|_{h_-,h_+} = -\langle GS_+ | \sigma_1^z | GS_+ \rangle \Big|_{-h_-,-h_+} \tag{6.2}$$

where $|GS_-\rangle$ is the normalized state of magnetization $-1/2$ which is the ground state if $h_- + h_+ > 0$. We can therefore expect to have, even for finite odd $L$, a discontinuity of the boundary magnetization at $h_- = -h_+$ which is given by:

$$\lim_{\substack{h_-\to-h_+ \\ h_-<-h_+}} \langle \sigma_1^z \rangle - \lim_{\substack{h_-\to-h_+ \\ h_->-h_+}} \langle \sigma_1^z \rangle$$

$$= \langle GS_+ | \sigma_1^z | GS_+ \rangle \Big|_{h_-=-h_+,h_+} - \langle GS_- | \sigma_1^z | GS_- \rangle \Big|_{h_-=-h_+,h_+}$$

$$= \langle GS_+ | \sigma_1^z | GS_+ \rangle \Big|_{h_-=-h_+,h_+} + \langle GS_+ | \sigma_1^z | GS_+ \rangle \Big|_{-h_-=h_+,-h_+}, \tag{6.3}$$

in which we have used (6.2). This discontinuity can be evaluated in the thermodynamic limit by means of (5.20). It is easy to check that it vanishes if $|h_+| > h_{cr}^{(1)}$, whereas, if $|h_+| < h_{cr}^{(1)}$, it gives

$$\lim_{\substack{h_-\to-h_+ \\ h_-<-h_+}} \lim_{L\to\infty} \langle \sigma_1^z \rangle - \lim_{\substack{h_-\to-h_+ \\ h_->-h_+}} \lim_{L\to\infty} \langle \sigma_1^z \rangle = \langle \sigma_1^z \rangle_0 \Big|_{h_-} + \langle \sigma_1^z \rangle_0 \Big|_{-h_-}$$

$$= -i\pi \sinh^2 \xi_- \left[ \rho'(-i|\zeta/2 - \tilde{\xi}_-| + \pi/2) + \rho'(-i|\zeta/2 + \tilde{\xi}_-| + \pi/2) \right]$$

$$= -2i\pi \sinh^2 \xi_- \rho'(-i(\zeta/2 + \xi_-)), \tag{6.4}$$

and we recover the value of the discontinuity (5.17) that we had obtained for the thermodynamic limit of the boundary magnetization for even length $L$ at $h_- = h_+$.

Hence, as expected, it follows from the previous study that the thermodynamic behavior of the boundary magnetization coincides for even and odd $L$ (see fomula (5.22), and see also Figure 15), *provided we change the sign of the boundary field $h_+$ at infinity*[9]. The only possible discrepancy is when we are exactly at $h_- = -h_+$ for $L$ odd with respect to the case $h_- = h_+$ for $L$ even. Whereas we did not have an exact degeneracy at this point in the even $L$ case, so that the ground states can be defined without ambiguity from the consideration of the finite size corrections, this not the case for $L$ odd: even for finite size we have a two-dimensional eigenspace generated by the two degenerate normalized Bethe states $|GS_+\rangle$ and $|GS_-\rangle$. We

---

[9]In other words, the quantity $h$ which should be kept fixed when considering the thermodynamic limit is the combination $h \equiv (-1)^L h_+$. This effect is a direct consequence of the anti-ferromagnetic nature of the chain and can easily be understood by considering the Ising limit.

see that, in that case, to recover the factor $1/2$ that we had obtained at this point from the consideration of the boundary root in the even $L$ case (see (5.16) and (5.22)), we have to consider the mean value of $\sigma_1^z$ in a superposition

$$|\widetilde{GS}_{\pm}\rangle = \frac{|GS_{+}\rangle \pm |GS_{-}\rangle}{\sqrt{2}} \tag{6.5}$$

of these two ground Bethe states of different magnetization. Note that (6.5) corresponds to the two ground states which are also eigenstates of the spin-flip operator $\mathcal{F} = \bigotimes_{n=1}^{L} \sigma_n^x$: $\mathcal{F}|\widetilde{GS}_{\pm}\rangle = \pm|\widetilde{GS}_{\pm}\rangle$.

Figure 15: Boundary Magnetization in even and odd-length chains at anisotropy $\Delta = 3$ in terms of the left boundary field $h_-$ with fixed right boundary fields $h_+ = h$ in the even case, $h_+ = -h$ in the odd cases. *Above*: $h = 0$. *Middle*: $h = 1$. *Below*: $h = -3.5$.

### 6.3  The spin-spin autocorrelation function at $h_- = -h_+$

It is clear that the large $L$ limit of the connected autocorrelation function computed in the ground Bethe state $|\mathrm{GS}_\pm\rangle$ always vanishes at large time for $L$ odd, even in the case of a degeneracy of the ground state when $h_- = -h_+$:

$$\lim_{t\to\infty}\lim_{L\to\infty} \langle \mathrm{GS}_\pm | \sigma_1^z(t)\, \sigma_1^z |\mathrm{GS}_\pm\rangle^c = 0. \tag{6.6}$$

Indeed, in the latter case, the two ground Bethe states have different magnetization, and therefore cannot contribute to the form-factor series of the autocorrelation function since the matrix elements of the operator $\sigma_1^z$ between states of different magnetization always vanish. On the other hand, if at $h_- = -h_+$ one considers as above the mean value in a superposition $|\widetilde{\mathrm{GS}}_\pm\rangle$ (6.5) of these two ground states which corresponds to an eigenstate of the spin-flip operator, one obtains

$$\lim_{t\to\infty}\lim_{L\to\infty} \langle \widetilde{\mathrm{GS}}_\pm | \sigma_1^z(t)\, \sigma_1^z |\widetilde{\mathrm{GS}}_\pm\rangle^c = \lim_{L\to\infty} \left| \langle \widetilde{\mathrm{GS}}_\pm | \sigma_1^z |\widetilde{\mathrm{GS}}_\mp\rangle \right|^2, \tag{6.7}$$

where the contributing form factor,

$$\langle \widetilde{\mathrm{GS}}_\pm | \sigma_1^z |\widetilde{\mathrm{GS}}_\mp\rangle = \frac{1}{2}\big( \langle \mathrm{GS}_+ | \sigma_1^z |\mathrm{GS}_+\rangle - \langle \mathrm{GS}_- | \sigma_1^z |\mathrm{GS}_-\rangle \big), \tag{6.8}$$

is effectively given by half of the discontinuity of the boundary magnetization (6.3).

The fact that we have to consider the superposition of Bethe states (6.5) is somehow the counterpart of the fact that, for even $L$ at the point $h_- = h_+$, the boundary root in the ground state is delocalized between the two edges and contributes only with a factor $1/2$ to the boundary magnetization: it can therefore be seen as a "superposition" of the two boundary roots which characterize the ground state for $h_- > h_+$ or $h_- < h_+$ respectively.

## 7  Conclusion

In this paper we have shown that the physics of the open XXZ chain at zero temperature and in the antiferromagnetic regime $\Delta > 1$ is strongly influenced by the presence of its boundary modes. In the language of Bethe ansatz these modes correspond to isolated complex Bethe roots converging exponentially fast with $L$ towards a zero of the boundary factor, and can be understood as excitations that are exponentially pinned at one of the two edges of the chain. We have shown that for chains of even size there exists a regime, $|h_\pm| < h_{\mathrm{cr}}^{(1)}$, in which such a boundary root is present both in the Bethe solutions for the ground state and for the first excited state. The values of the two boundary magnetic fields determine at which edge the corresponding boundary excitation is localized in the ground state. As a consequence of this localization, the value of the spin magnetization at one of the boundaries of the chain also depends indirectly on the value of the magnetic field at the opposite edge, even in the thermodynamic limit: it presents in particular a discontinuity in this limit at $h_- = h_+$. Moreover we have shown that, when the two boundary fields are equal ($h_- = h_+ = h$ with $|h| < h_{\mathrm{cr}}^{(1)} = \Delta - 1$), the spectrum is gapped in the thermodynamic limit and the ground state is doubly degenerate up to exponentially small corrections in $L$. In this case, the boundary root in the ground state (or in the quasi-ground state) is delocalized between the two edges and contributes only with a factor $1/2$ to the boundary magnetization; furthermore, the spin-spin autocorrelation function on one of the two edges relaxes at large time to a finite value, given by the contribution of the boundary root to the boundary magnetization, or equivalently by half of the discontinuity of this boundary magnetization at $h_- = h_+$.

Note that such an effect due to the change of localization of an isolated boundary root in the ground state in the regime $|h_{\pm}| < h_{cr}^{(1)}$ is specific to chains of even length. For chains of odd length, the discontinuity of the boundary magnetization that can be observed at $h_- = -h_+$ (even at finite size) is simply the consequence of a crossing of levels in the finite-size spectrum between a state of magnetization $+1/2$ and a state of magnetization $-1/2$: at exactly $h_- = -h_+$, the finite-size spectrum of chains of odd length is indeed doubly degenerate due to an exact $\mathbb{Z}_2$ symmetry. For chains of even length we no longer have this exact double degeneracy at finite size, in particular for the ground state, but also for some higher excited states, and one may wonder whether the presence of the boundary root can be understood as a possible signature of the remaining quasi-degeneracy that survives in this case in the thermodynamic limit.

It would be desirable to establish a more direct relation between the boundary root and the strong zero mode found in [12] in the case $h_- = h_+ = 0$. In particular few questions can be immediately formulated. Is the strong zero mode related to the creation operator of the boundary excitation corresponding to the boundary root at one edge of the chain? And what happens at finite temperatures? Does the degeneracy observed in [12] at all energies in the spectrum, which is due to the strong zero mode, translate into the presence of a boundary root also for finite temperature states? Are there two degenerate representative thermal states (via the thermodynamic Bethe ansatz [55]) distinguished by two opposite deviations of the boundary root, as it is the case for the ground state? We postpone the study of these interesting questions to future works.

It would also be interesting to investigate supersymmetric properties of the open chain (see for example [56, 57]), and to understand how to formulate an ensemble of states in the presence of boundary roots. Namely how to determine for example the steady state after a quantum quench [58] (Generalised Gibbs Ensemble) close to the edge of the chain. It is indeed evident that the value of the conserved charges that are extensive in the system must be supplied with the information about the boundary [59] which at the moment is not clear how to include in the steady state of an interacting system.

# Acknowledgements

The work of S.G. is supported by a SENESCYT-IFTH fellowship from the Government of Ecuador. J.D.N. is supported by the Research Foundation Flanders (FWO) and in the early stage of the work by the LabEx ENS-ICFP:ANR-10-LABX-0010/ANR-10-IDEX-0001-02 PSL*. V.T. is supported by CNRS. V.T. gratefully acknowledges support from the Simons Center for Geometry and Physics, Stony Brook University, at which some of the research for this paper was performed during the program *Exactly Solvable Models of Quantum Field Theory and Statistical Mechanics*. Numerical calculations were performed using the ITensor C++ library, and the QuSpin Python exact diagonalization package [60].

# A  The Bethe equations in logarithmic form and allowed quantum numbers for the real roots

In this appendix we consider the Ising limit $\Delta \to +\infty$, i.e. $\zeta \to +\infty$, of the logarithmic Bethe equations (4.11). More precisely, we suppose that $\Delta$, and therefore $\zeta$, are large but finite, and we write the logarithmic Bethe equations at leading order in $\zeta$. At leading order in $\zeta$ the counting function (4.12) becomes linear, which enables us to determine the allowed quantum numbers $n_j$ for the real roots.

It is easy to see from the expressions (4.13)-(4.16) that, for $\alpha \in \mathbb{R}$,

$$p(\alpha) \underset{\zeta \to +\infty}{\longrightarrow} 2\alpha, \tag{A.1}$$

$$\theta(\alpha) \underset{\zeta \to +\infty}{\longrightarrow} -2\alpha, \tag{A.2}$$

$$\Theta(\alpha, \lambda_k) \underset{\zeta \to +\infty}{\longrightarrow} \begin{cases} -4\alpha & \text{if} \quad |\operatorname{Im}(\lambda_k)| = o(\zeta), \\ 0 & \text{if} \quad \zeta = o(|\operatorname{Im}(\lambda_k)|), \end{cases} \tag{A.3}$$

and that

$$g(\alpha) \underset{\zeta \to +\infty}{\longrightarrow} -2(\tilde{\delta}_+ + \tilde{\delta}_-)\alpha, \tag{A.4}$$

where, for $\sigma \in \{+,-\}$,

$$\tilde{\delta}_\sigma = \begin{cases} 1 & \text{if} \quad h_\sigma < h_{\text{cr}}^{(1)} \text{ or } h_\sigma > h_{\text{cr}}^{(2)} \quad (\text{i.e. for } \tilde{\xi}_\sigma < \zeta/2), \\ -1 & \text{if} \quad h_{\text{cr}}^{(1)} < h_\sigma < h_{\text{cr}}^{(2)} \quad (\text{i.e. for } \tilde{\xi}_\sigma > \zeta/2). \end{cases} \tag{A.5}$$

Let us now consider a solution $\{\lambda\} \equiv \{\lambda_1, \dots, \lambda_N\}$ of the logarithmic Bethe equations (4.11). Let $n_w$ be the number of wide roots $\lambda_k$ such that $\zeta = o(|\operatorname{Im}(\lambda_k)|)$. Then, if $\alpha \in \mathbb{R}$,

$$\hat{\xi}_L(\alpha|\{\lambda\}) \underset{\zeta \to +\infty}{\sim} \frac{2M}{L}\alpha, \tag{A.6}$$

where

$$M = L - N + n_w + 1 - \frac{\tilde{\delta}_+ + \tilde{\delta}_-}{2}, \tag{A.7}$$

so that the logarithmic Bethe equations (4.11) for each real root $\lambda_j$ become, at leading order in $\zeta$:

$$\lambda_j \underset{\zeta \to +\infty}{\sim} \frac{\pi n_j}{2M}. \tag{A.8}$$

Since the allowed real solutions are such that $0 < \lambda_j < \frac{\pi}{2}$ (we recall that we have to discard the obvious solutions 0 and $\frac{\pi}{2}$, see footnote 2), the integers $n_j$ associated with real roots can then take only the possible values

$$n_j \in \{1, 2, \dots, M-1\}. \tag{A.9}$$

Hence we have to distinguish different cases.

**Case A.** Both boundary fields $h_+$ and $h_-$ are *not* in the interval delimited by the two critical fields $h_{\text{cr}}^{(1)}$ and $h_{\text{cr}}^{(2)}$ ($h_\pm \notin [h_{\text{cr}}^{(1)}, h_{\text{cr}}^{(2)}]$).

1. There are $L - N + n_w - 1$ possible quantum numbers for the real roots.

2. The maximum number of real Bethe roots for a solution in the sector $N = \frac{L}{2}$ is $N-1$. Such a solution therefore contains an additional isolated complex root, which may correspond either to one of the two possible boundary roots $\alpha_{\text{BR}}^\sigma$ (4.9) with $\sigma \in \{+,-\}$, or to a wide root. The corresponding quantum numbers $n_j$ ($j = 1, \dots N-1$) for the real roots are such that

   (a) $\{n_1, \dots, n_{N-1}\} = \{1, \dots N-1\}$ (there is no hole) if the complex root is a boundary root $\alpha_{\text{BR}}^\sigma$; this is possible only if the corresponding field $h_\sigma$ is *not* between $-h_{\text{cr}}^{(2)}$ and $-h_{\text{cr}}^{(1)}$ (since in that case $\operatorname{Im}(\alpha_{\text{BR}}^\sigma) = o(\zeta)$);

   (b) $\{n_1, \dots, n_{N-1}\} \subset \{1, \dots, N\}$ (there is one hole) if the complex root is a wide root.

Other types of solutions in the sector $N = \frac{L}{2}$ contain more holes, except the solution with $N - 2$ real roots and a pair of bulk close roots (i.e. from [49] a 2-string), which has to be compared with the solution 2b since it also contains one hole.

3. In the sector $N = \frac{L}{2} - 1$, there exists a solution with $N$ real roots (and therefore no complex root) with quantum numbers to be distributed within the set $\{1, \ldots, N + 1\}$. Hence this solution contains a hole at some position $h \in \{1, \ldots, N + 1\}$. Other possible solutions in that sector or in sectors $N < \frac{L}{2} - 1$ contain more holes.

**Case B.** One of the fields is in the interval delimited by the two critical fields $h_{\text{cr}}^{(1)}$ and $h_{\text{cr}}^{(2)}$ and the other is not.

1. There are $L - N + n_w$ possible quantum numbers for the real roots.

2. The maximum number of real Bethe roots for a solution in the sector $N = \frac{L}{2}$ is $N$. It corresponds to a full set of adjacent quantum numbers $j = 1, \ldots, N$ (no hole and no complex root). Other types of solutions with complex roots in that sector contain one or more hole(s).

3. Solutions in sectors $N < \frac{L}{2}$ contain at least two holes.

**Case C.** Both boundary fields are in the interval delimited by the two critical fields $h_{\text{cr}}^{(1)}$ and $h_{\text{cr}}^{(2)}$ ($h_{\text{cr}}^{(1)} < h_{\pm} < h_{\text{cr}}^{(2)}$).

1. There are $L - N + n_w + 1$ possible quantum numbers for the real roots.

2. In the sector $N = \frac{L}{2}$, there exists a solution with $N$ real roots (and therefore no complex root) with quantum numbers to be distributed within the set $\{1, \ldots, N + 1\}$. Hence this solution contains a hole at some position $h \in \{1, \ldots, N + 1\}$. Other possible solutions in that sector, i.e. with some complex roots, contain two or more holes.

# B  Controlling the finite-size corrections in the large $L$ limit

In this appendix, we explain how to control the finite-size corrections to the integral over the density which come from sums over real Bethe roots in the large $L$ limit.

Let $\{\lambda\} \equiv \{\lambda_1, \ldots, \lambda_N\}$ be a solution of the Bethe equations (4.4). We suppose that this solution corresponds to an infinite number of real roots (i.e. of order $L$), with a finite number of complex roots and a finite number of holes in the thermodynamic limit. The logarithmic equation for the real roots can be written as in (4.18), in terms of the positions $h_1, \ldots, h_n$ of the holes in the adjacent set of quantum numbers for the real roots, with $M$ given by (A.7) that we suppose to be of the same order as $L$. Note that the counting function $\widehat{\xi}(\alpha) \equiv \widehat{\xi}(\alpha | \{\lambda\})$ associated with this set of Bethe roots, which is defined as in (4.12), satisfies the following properties for $\alpha \in \mathbb{R}$:

$$\widehat{\xi}(-\alpha) = -\widehat{\xi}(\alpha), \tag{B.1}$$

$$\widehat{\xi}(\alpha + \pi) = \widehat{\xi}(\alpha) + \frac{2M}{L}\pi, \tag{B.2}$$

$$\widehat{\xi}(0) = 0, \qquad \widehat{\xi}\left(\frac{\pi}{2}\right) = -\widehat{\xi}\left(-\frac{\pi}{2}\right) = \frac{M\pi}{L}. \tag{B.3}$$

Moreover,

$$\widehat{\xi}'(\alpha) \underset{L \to \infty}{\longrightarrow} \pi\rho(\alpha) > 0, \tag{B.4}$$

so that $\widehat{\xi}$ is an increasing, and hence invertible function for $L$ large enough (see the argument in the footnote of [50]). We can therefore introduce the inverse images $\check{\lambda}_j$ of $\frac{\pi j}{L}$ for $j \in \{1, \ldots, M-1\}$:

$$\widehat{\xi}(\check{\lambda}_j | \{\lambda\}) = \frac{\pi j}{L}, \qquad j \in \{1, \ldots, M\}, \tag{B.5}$$

which defines in particular the hole rapidities $\check{\lambda}_{h_k}$ for $k \in \{1, \ldots, n\}$ (recall that $\check{\lambda}_j$ coincides with the real root $\lambda_j$ if $j \neq h_1, \ldots, h_n$).

## B.1 From the sums over the real roots to integrals

PROPOSITION B.1. *Let $f$ be a $C^\infty$ $\pi$-periodic even function on $\mathbb{R}$. Let $\{\lambda\} \equiv \{\lambda_1, \ldots, \lambda_N\}$ be a solution of the Bethe equations defined as above, and let $\widehat{\xi}(\alpha) \equiv \widehat{\xi}(\alpha | \{\lambda\})$ be the corresponding counting function. Then, the sum of all the values $f(\lambda_j)$ corresponding to the real roots $\lambda_j$, $j \in \{1, \ldots, M-1\} \setminus \{h_1, \ldots, h_n\}$, can be replaced by an integral in the large $L$ limit according to the following rule:*

$$\frac{1}{L} \sum_{\substack{j=1 \\ j \neq h_1, \ldots, h_n}}^{M-1} f(\lambda_j) = \frac{1}{2\pi} \int_{-\frac{\pi}{2}}^{\frac{\pi}{2}} f(x) \widehat{\xi}'(x) \, dx - \frac{f(0) + f(\frac{\pi}{2})}{2L} - \frac{1}{L} \sum_{j=1}^{n} f(\check{\lambda}_{h_j}) + O(L^{-\infty}), \tag{B.6}$$

*where $O(L^{-\infty})$ stand for exponentially small corrections in $L$.*

*Proof.* The proof can be done with similar arguments as in [47] (see also [50]), adapted here to the case of the open chain and of general low-energy states.

Since $f$ is $\pi$-periodic,

$$\frac{1}{2M} \sum_{k=-M+1}^{M} f\left(\frac{\pi k}{2M}\right) = \frac{1}{2M} \sum_{k=1}^{2M} f\left(\frac{\pi k}{2M}\right) = \frac{1}{\pi} \int_{-\frac{\pi}{2}}^{\frac{\pi}{2}} f(x) \, dx + O(M^{-\infty}), \tag{B.7}$$

and since $f$ is even,

$$\frac{1}{M} \sum_{k=1}^{M-1} f\left(\frac{\pi k}{2M}\right) = \frac{1}{2M} \sum_{k=-M+1}^{M} f\left(\frac{\pi k}{2M}\right) - \frac{f(0) + f(\frac{\pi}{2})}{2M}$$

$$= \frac{1}{\pi} \int_{-\frac{\pi}{2}}^{\frac{\pi}{2}} f(x) \, dx - \frac{f(0) + f(\frac{\pi}{2})}{2M} + O(M^{-\infty}). \tag{B.8}$$

We now make a change of variables using the function $\widetilde{\xi}$ defined from the counting function $\widehat{\xi}$ as

$$\widetilde{\xi}(\alpha) = \frac{L}{2M} \widehat{\xi}(\alpha), \tag{B.9}$$

which is still odd and invertible and satisfies, instead of (B.2) and (B.3), the properties

$$\widetilde{\xi}(\alpha + \pi) = \widetilde{\xi}(\alpha) + \pi, \qquad \widetilde{\xi}(0) = 0, \qquad \widetilde{\xi}\left(\frac{\pi}{2}\right) = -\widetilde{\xi}\left(-\frac{\pi}{2}\right) = \frac{\pi}{2}. \tag{B.10}$$

Hence, the function $f \circ \widetilde{\xi}^{-1}$ is also even and $\pi$-periodic, so that we have

$$\frac{1}{M} \sum_{k=1}^{M-1} f(\check{\lambda}_k) = \frac{1}{M} \sum_{k=1}^{M-1} f \circ \widetilde{\xi}^{-1}\left(\frac{\pi k}{2M}\right)$$

$$= \frac{1}{\pi} \int_{-\frac{\pi}{2}}^{\frac{\pi}{2}} f \circ \widetilde{\xi}^{-1}(x) \, dx - \frac{f \circ \widetilde{\xi}^{-1}(0) + f \circ \widetilde{\xi}^{-1}(\frac{\pi}{2})}{2M} + O(M^{-\infty})$$

$$= \frac{1}{\pi} \int_{-\frac{\pi}{2}}^{\frac{\pi}{2}} f(\mu) \widetilde{\xi}'(\mu) \, d\mu - \frac{f(0) + f(\frac{\pi}{2})}{2M} + O(M^{-\infty}). \tag{B.11}$$

Multiplying by $M/L$ and setting appart the contributions of the holes from the ones of the real roots we obtain (B.6). $\qquad\square$

COROLLARY B.1. *Let $f$ be a $\mathcal{C}^\infty$ $\pi$-periodic function on $\mathbb{R}$. Then, with the same notations as in Proposition B.1,*

$$
\frac{1}{L}\sum_{\substack{j=1\\j\neq h_1,\dots,h_n}}^{M-1}\left[f(\lambda_j)+f(-\lambda_j)\right]=\frac{1}{\pi}\int_{-\frac{\pi}{2}}^{\frac{\pi}{2}}f(x)\widehat{\xi}'(x)\,dx-\frac{f(0)+f(\frac{\pi}{2})}{L}
$$
$$
-\frac{1}{L}\sum_{j=1}^{n}\left[f(\check{\lambda}_{h_j})+f(-\check{\lambda}_{h_j})\right]+O(L^{-\infty}). \quad\text{(B.12)}
$$

*Let $g$ be a $\mathcal{C}^\infty$-function such that $g'$ is $\pi$-periodic. Then*

$$
\frac{1}{L}\sum_{\substack{j=1\\j\neq h_1,\dots,h_n}}^{M-1}\left[g(\lambda_j)+g(-\lambda_j)\right]=\frac{1}{\pi}\int_{-\frac{\pi}{2}}^{\frac{\pi}{2}}g(x)\widehat{\xi}'(x)\,dx-\frac{g(\frac{\pi}{2})+g(-\frac{\pi}{2})+2g(0)}{2L}
$$
$$
-\frac{1}{L}\sum_{j=1}^{n}\left[g(\check{\lambda}_{h_j})+g(-\check{\lambda}_{h_j})\right]+O(L^{-\infty}). \quad\text{(B.13)}
$$

*Proof.* (B.12) is a direct consequence of (B.6).

If $g'(x)$ is $\pi$-periodic then $g(x)-c_g x$ is also $\pi$-periodic, where

$$
c_g=\frac{1}{\pi}\int_{-\frac{\pi}{2}}^{\frac{\pi}{2}}g'(x)\,dx=\frac{g(\frac{\pi}{2})-g(-\frac{\pi}{2})}{\pi}=\frac{g(y+\pi)-g(y)}{\pi},\qquad\forall y. \quad\text{(B.14)}
$$

Hence one can apply (B.12) to $g(x)-c_g x$,

$$
\frac{1}{L}\sum_{k=1}^{M-1}\left[g(\check{\lambda}_k)+g(-\check{\lambda}_k)\right]=\frac{1}{L}\sum_{k=1}^{M-1}\left[g(\check{\lambda}_k)-c_g\check{\lambda}_k+g(-\check{\lambda}_k)+c_g\check{\lambda}_k\right]
$$
$$
=\frac{1}{\pi}\int_{-\frac{\pi}{2}}^{\frac{\pi}{2}}\left[g(x)-c_g x\right]\widehat{\xi}'(x)\,dx-\frac{g(0)+g(\frac{\pi}{2})-c_g\frac{\pi}{2}}{L}+O(L^{-\infty})
$$
$$
=\frac{1}{\pi}\int_{-\frac{\pi}{2}}^{\frac{\pi}{2}}g(x)\widehat{\xi}'(x)\,dx-\frac{c_g}{\pi}\int_{-\frac{\pi}{2}}^{\frac{\pi}{2}}x\,\widehat{\xi}'(x)\,dx
$$
$$
-\frac{g(\frac{\pi}{2})+g(-\frac{\pi}{2})+2g(0)}{2L}+O(L^{-\infty}), \quad\text{(B.15)}
$$

and the second integral vanishes due to the fact that $\widehat{\xi}'$ is an even function. $\qquad\square$

## B.2 Finite size corrections to the counting function

We can in particular apply (B.13) to transform the sum over real roots in the definition (4.12) of the counting function:

$$
\widehat{\xi}(\alpha) = p(\alpha) + \frac{g(\alpha)}{2L} - \frac{\theta(2\alpha)}{2L} + \frac{1}{2\pi} \int_{-\frac{\pi}{2}}^{\frac{\pi}{2}} \theta(\alpha - \mu) \widehat{\xi}'(\mu) \, d\mu + \frac{1}{2L} \sum_{k \in \mathcal{Z}} \Theta(\alpha, \lambda_k)
$$
$$
- \frac{\theta(\alpha - \frac{\pi}{2}) + \theta(\alpha + \frac{\pi}{2}) + 2\theta(\alpha)}{4L} - \frac{1}{2L} \sum_{j=1}^{n} \left[ \theta(\alpha - \check{\lambda}_{h_j}) + \theta(\alpha + \check{\lambda}_{h_j}) \right] + O(L^{-\infty}), \quad \text{(B.16)}
$$

in which $\mathcal{Z}$ is the set of indices corresponding to the complex roots (i.e. $\mathrm{Im}(\lambda_k) \neq 0$ if $k \in \mathcal{Z}$). Deriving (B.16), we obtain the following integral equation for $\widehat{\xi}'$:

$$
\widehat{\xi}'(\alpha) = p'(\alpha) + \frac{g'(\alpha)}{2L} - \frac{\theta'(2\alpha)}{L} + \frac{1}{2\pi} \int_{-\frac{\pi}{2}}^{\frac{\pi}{2}} \theta'(\alpha - \mu) \widehat{\xi}'(\mu) \, d\mu - \frac{\theta'(\alpha) + \theta'(\alpha + \frac{\pi}{2})}{2L}
$$
$$
+ \frac{1}{2L} \sum_{k \in \mathcal{Z}} \Theta'(\alpha, \lambda_k) - \frac{1}{2L} \sum_{j=1}^{n} \left[ \theta'(\alpha - \check{\lambda}_{h_j}) + \theta'(\alpha + \check{\lambda}_{h_j}) \right] + O(L^{-\infty}). \quad \text{(B.17)}
$$

Hence, the expression (B.16) of the counting function can be decomposed in terms of the different contributions of the real roots, the complex roots and the holes as in (4.20). In (4.20), $\widehat{\xi}_0(\alpha)$ is the common contribution of the "Fermi sea" of real roots. It is an odd function, and its derivative is defined as the solution of the integral equation

$$
\widehat{\xi}_0'(\alpha) + \int_{-\frac{\pi}{2}}^{\frac{\pi}{2}} K(\alpha - \mu) \widehat{\xi}_0'(\mu) \, d\mu = p'(\alpha) + \frac{g'(\alpha)}{2L} - \frac{\theta'(2\alpha)}{L} - \frac{\theta'(\alpha) + \theta'(\alpha + \frac{\pi}{2})}{2L}. \quad \text{(B.18)}
$$

Note that $\widehat{\xi}_0'(\alpha)$ can itself be decomposed as

$$
\widehat{\xi}_0'(\alpha) = \pi \rho(\alpha) + \frac{1}{L} \widehat{\xi}_{\mathrm{open}}'(\alpha), \quad \text{(B.19)}
$$

where $\rho$ is the density (4.8) solution of (4.7), and where $\widehat{\xi}_{\mathrm{open}}'$ is the correction due to the $1/L$ terms in (B.18), which is defined as the solution to the integral equation

$$
\widehat{\xi}_{\mathrm{open}}'(\alpha) + \int_{-\frac{\pi}{2}}^{\frac{\pi}{2}} K(\alpha - \mu) \widehat{\xi}_{\mathrm{open}}'(\mu) \, d\mu = \frac{g'(\alpha)}{2} - \theta'(2\alpha) - \frac{\theta'(\alpha) + \theta'(\alpha + \frac{\pi}{2})}{2}. \quad \text{(B.20)}
$$

The function $\widehat{\xi}_\mu$, which corresponds to the contribution to the counting function of an excitation (an additional complex root or a hole at position $\mu$) with respect to the above Fermi sea of real roots, is also an odd function with derivative being the solution of the integral equation:

$$
\widehat{\xi}_\mu'(\alpha) + \int_{-\frac{\pi}{2}}^{\frac{\pi}{2}} K(\alpha - \beta) \widehat{\xi}_\mu'(\beta) \, d\beta = \frac{\Theta'(\alpha, \mu)}{2}. \quad \text{(B.21)}
$$

This latter can easily be computed in Fourier modes by using the following lemma:

LEMMA B.1. *Let $\varphi'(z, \gamma)$ be the function (4.17) with $\gamma > 0$. Then, for $x, y \in \mathbb{R}$,*

$$
\varphi'(x + iy, \gamma) = \sum_{k=-\infty}^{+\infty} \varphi_k(y, \gamma) e^{2ikx}, \quad \text{(B.22)}
$$

*with*

$$\varphi_k(y,\gamma) = \frac{1}{\pi}\int_{-\frac{\pi}{2}}^{\frac{\pi}{2}} \frac{\sinh(2\gamma)}{\sin(x+i(\gamma+y))\sin(x-i(\gamma-y))} e^{-2ikx}\,dx$$

$$= \sum_{\sigma=\pm} 2\,\mathrm{sgn}(\gamma+\sigma y)\,\mathrm{H}(k(y+\sigma\gamma))\,e^{-2k(y+\sigma\gamma)}, \tag{B.23}$$

*where* H *denotes the Heaviside function and* sgn *the sign function.*

We obtain that $\widehat{\xi}'_\mu(\alpha) = \frac{1}{2}\big[\check{\xi}'_\mu(\alpha)+\check{\xi}'_{\bar\mu}(\alpha)\big]$, with

$$\check{\xi}'_\mu(\alpha) = \begin{cases} -\displaystyle\sum_{k=-\infty}^{+\infty} \frac{e^{-|k|\zeta}}{\cosh(k\zeta)}\cos(2k\mu)\,e^{2ik\alpha} & \text{if } |\mathrm{Im}(\mu)| < \zeta, \\[2ex] 2\displaystyle\sum_{k=-\infty}^{\infty} e^{|k|\zeta}\,\sinh(|k|\zeta)\,e^{2i|k|\,\mathrm{sgn}(\mathrm{Im}\,\mu)\mu}\,e^{2ik\alpha} & \text{if } |\mathrm{Im}(\mu)| > \zeta. \end{cases} \tag{B.24}$$

## B.3   Finite-size corrections to the energy

We now apply the results of the previous subsections so as to compute the energy (2.17) associated with a given solution $\{\lambda\} \equiv \{\lambda_1,\dots,\lambda_N\}$ for large $L$ up to exponentially small corrections in $L$.

Formula (4.28) is a direct consequence of (B.6) and (4.20). In this expression, the common contribution $E_0$ of the real roots is

$$E_0 = h_+ + h_- + \frac{L}{2\pi}\int_{-\frac{\pi}{2}}^{\frac{\pi}{2}} \varepsilon_0(\mu)\,\widehat{\xi}'_0(\mu)\,d\mu - \frac{\varepsilon_0(0)+\varepsilon_0(\frac{\pi}{2})}{2}, \tag{B.25}$$

and $\varepsilon(\mu)$ is the dressed energy of an excitation with rapidity $\mu$, defined as in (4.29), in terms of the bare energy (2.18) and of the correction to the counting function due to the root $\mu$, see (B.24)-(B.24). We can compute the expression of (4.29) in Fourier modes, by using Lemma B.1 and the expression (B.24) of $\widehat{\xi}'_\mu(\alpha)$:

$$\varepsilon(\mu) = \frac{\varepsilon_\mu + \varepsilon_{\bar\mu}}{2}, \tag{B.26}$$

where

$$\varepsilon_\mu = -2\sinh\zeta\left[\varphi'(\mu,\zeta/2) + \frac{1}{2\pi}\int_{-\frac{\pi}{2}}^{\frac{\pi}{2}} \varphi'(\beta,\zeta/2)\,\widehat{\xi}'_\mu(\beta)\,d\beta\right], \tag{B.27}$$

$$= \begin{cases} -2\sinh\zeta\displaystyle\sum_{k\in\mathbb{Z}} \frac{e^{2ik\mu}}{\cosh(k\zeta)} = -2\pi\sinh\zeta\,\rho(\mu) & \text{if } |\mathrm{Im}(\mu)| < \zeta/2, \\[2ex] 2\sinh\zeta\displaystyle\sum_{k\in\mathbb{Z}} \frac{e^{2ik[\mu-i\,\mathrm{sgn}(\mathrm{Im}\,\mu)\zeta]}}{\cosh(k\zeta)} = -2\pi\sinh\zeta\,\rho(\mu) & \text{if } \zeta/2 < |\mathrm{Im}(\mu)| < \zeta, \\[2ex] 4\,\mathrm{sgn}(\mathrm{Im}\,\mu)\sinh\zeta\displaystyle\sum_{k\in\mathbb{Z}} \sinh(k\zeta)\,e^{2i|k|\,\mathrm{sgn}(\mathrm{Im}\,\mu)\mu} = 0 & \text{if } |\mathrm{Im}(\mu)| > \zeta, \end{cases}$$

in which $\rho$ is the ratio of Theta functions (4.8). Here we have notably used the quasi-periodicity property $\rho(\mu\pm i\zeta) = -\rho(\mu)$.

In particular, the dressed energy of a hole with rapidity $\check{\lambda}_h \in (0,\frac{\pi}{2})$ is given by (4.32), whereas the dressed energy of the boundary root (4.9) is given by (4.33).

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
