# Peer review of "Open XXZ chain and boundary modes at zero temperature"

_SciPost Physics, doi:SciPost Phys. 7, 023 (2019)_

## Round 2 · Referee Report · Anonymous (Referee 1) · 2019-2-21

Strengths

  1. New results.
  2. Exact treatment.
  3. Good presentation.
  4. Correcting earlier mistakes in the literature.

Weaknesses

  1. Insufficient physical explanations.
  2. Treatment of earlier literature is a bit rushed.

Report

This paper deals with the boundary XXZ chain with diagonal boundary field. It discusses the nature of the ground state at even length and it has very interesting results. In certain cases it corrects mistakes made in some papers in the literature, including a classic paper [15].

Nevertheless I have certain comments about the presentation. The main issue is about the meaning of the results and establishing the connection to experiments and to earlier results in the literature.

This work deals with the ground state energy, and ground state boundary magnetization. Also, it discusses the boundary spin autocorrelation, which is important for non-equilibrium problems. The work defines the ground state as it should be: the lowest energy state in any finite volume. Nevertheless it discusses cases when $h_1 = h_L$ and there will be two degenerate states in the TDL, with an exponentially decaying (with volume) gap. Let us denote the true finite volume ground states by $GS_1$, and the second state as $GS_2$. The work deals with discontinuities in certain quantities, for example in the boundary magnetization. It is defined as
$<GS_1 | \sigma_{1,z} | GS_1 >$
It is noted that this quantity is discontinuous, and it depends also on the value of the magnetic field on the other side of the chain.

The statements are correct and the formulas are mathematically precise. Nevertheless the physical interpretation is not correct in my opinion. The authors explain that the reason for the discontinuities is that when we keep $h_1$ fixed and change $h_L$ to cross $h_1$, the boundary excitation ``jumps'' from one end to the other end. Now on a big chain there is clearly no instantenous jump and rearrangement. This discontinuity is an artifact of the mathematical definitions, but it has no immediate physical meaning. Its origin is that $GS_1$ and $GS_2$ are exchanged when we change one of the magnetic fields to cross the other one. There is a physical consequence though: the spin-autocorrelation, which is measurable, but any sudden jump in $\sigma_{1,z}$ as we change the magnetic field at the other end is not measurable.

Also, it is quite questionable what is relevant for experiments. When we cool a system down, it is hard to imagine that we would actually get to the $GS_1$ mean values in any situation. Instead, lowering the temperature would result in the average

$\frac{1}{2}(<GS_1 | \sigma_{1,z} | GS_1 > + <GS_2 | \sigma_{1,z} | GS_2 > )$

And this average is a continuous function of any of the magnetic fields. It is difficult to imagine that any measurement would detect the discontinuity investigated by the authors. Only the effect in the autocorrelation is easily detected.

The connection to the finite T results of [50,51] is a little bit rushed, but it should be certainly discussed. It is claimed that [50,51] can be valid only when there is free boundary condition at the other end. On the other hand, the authors explain in the Conclusion, that they don't know yet what happens at finite T.

From the results of this manuscript it seems to me that the authors might be correct that [50,51] (and possibly other works too) were wrong with regard to the boundary magnetization in the ground state, in the regimes discussed here. However, [50,51] take the $T\to 0$ limit of the finite $T$ formulas, which were obtained already in the $L\to\infty$ limit. It seems to me that the two limits $T\to 0$ and $L\to\infty$ do not commute in this case. [50,51] does the TDL first and $T\to 0$ afterwards, and the present manuscript does $T\to 0$ first (by selecting the true ground state in finite volume) and doing the TDL afterwards.

I think that this difference should be discussed. Also, it should be discussed which one might be more relevant for experimental situations.

The Conclusion leaves open the discussion of the finite $T$ case. I doubt that any such phenomenon would survive at finite $T$, but it is certainly worthwhile to investigate.

Also, it would be important to discuss the order of limits for the autocorrelation function. The hard statement is about the $t\to\infty$ limit in the cases when $h_1=h_L$. What happens here when we change only one of the fields, and how does it relate to the size of the system, and the order of limits?

One more problem with the present work is that it discusses only the even lenghts. Given the amount of mathematical precision in the definitions, it is a bit strange to focus only on even lengths. How would the situation look like at odd lengths? Again, at any finite $T$ we would expect that the length does not matter, but it is known that for the ground states it does matter. I am not so sure what should be suggested here: asking for a major revision to do the odd length case is perhaps too much. Nevertheless it should be mentioned and at least a little bit discussed.

Requested changes

  1. Discuss the odd length case.
  2. Discuss the physical meaning of the results.
  3. Discuss the exchange of limits with regards to the finite $T$ case. (or try to convince me if I am wrong)
  4. Discuss the order of limits with respect to the autocorrelation function.
  5. Formula (1.8) can not be correct in this form. If the chain is periodic, and there is only a global field, then nothing distinguished the odd sites and even sites, so the formula could be $(-1)^j$ but also $(-1)^{j+1}$. This should be somehow corrected, perhaps with a staggered field going to zero, or in any other way.
  6. There are a few small misprints, the text should be carefully read once more.

---

## Round 2 · Referee Report · Anonymous (Referee 2) · 2019-3-15

Strengths

The paper is mathematically robust and clear, and considers a non trivial problem. They obtained an unexpected result. In a special region of the model, the boundary magnetization depends on the magnetic field applied in one end of the chain. This is a surprise since the quantum chain is gaped.

Weaknesses

(opitional) Perhaps more physical discussions of the resuls would be desired

Report

The authors study the XXZ open quantum chain in the antiferromagnetic regime,
in the presence of magnetic fields coupled with the spins at the boundaries
of the lattice. Using the quantum inverse scattering method with boundaries
they were able to connect the expected double degeneracy of the ground state
(since the model has an antiferromagnetic order) with the existence of
a Bethe complex root (boundary root), related to an excitation localized at the
boundaries of the chain. As a result of their calculation they show that
the boundary magnetization at one end of the lattice (in the bulk limit)
depends on the value of the magnetic field at the other end. This is a
surprise.

The paper in my opinion is interesting from both, mathematical and physical
perspectives, and I recommend its acceptance for publication.

Requested changes

1)The authors should state in the abstract and several places that the model are considered in the anti ferromagnetic regime ($\Delta >1$ is not enough for that).

2) The integrability of the model were obtained in their Re.[21], this should be mentioned in starting Section 2.

---

## Round 3 · Referee Report · Anonymous · 2019-6-4

Report

The authors addressed my questions, and I think that the manuscript has improved considerably. Nevertheless I have some further questions. I am sure that the article can be published soon, once these remaining questions are settled.

-The authors added a complete new discussion about the odd length chain. I think this is a great improvement to the paper. Quite interestingly, footnote 9 explains that the thermodynamic limit should be taken with a boundary magnetic field which changes sign between odd and even length chains. Would the authors agree that this is a consequence of the anti-ferromagnetic nature of the chain? Can this be seen perhaps in the Ising limit?

-The authors are explaining very clearly that the odd length case is different, because there is a level crossing even at finite $L$. On the other hand, in the even length case there is a level avoidance, and the numerics is shown on the right of Fig. 3. However, from this figure it seems to me that in the $L\to\infty$ limit we would indeed get a level crossing, because clearly the gap has to smoothen out, it becomes exponentially small. I think that this should be mentioned: based on the $L\to\infty$ values the odd and even length cases are not so much different, given that the magnetic field is changed alternatingly. This is also a nice physical picture, that we don't really get different behaviour for an infinitely long chain, apart from the issue of the anti-ferromagnetic ordering.

-Fig 3. is very informative and useful. Nevertheless I would like to see a finite volume data of the GS boundary magnetization too. This is presented in Fig 15, but there the finite volume data is only used to confirm the infinite volume predictions. Or at least it seems to me that this is the role. I think that it would be much more pedagogical if some finite volume data for $\sigma_1^z$ would be added already around Fig 3. It would be nice to see data from two $L$ values, perhaps the $L=12$ and some smaller value. Such that we could see the closing of the gap like on the right of Fig 3 now, and the appearance of the discontinuity like on Fig 15. I stress again that these figures are already quite good and informative, but the understanding would be much better if they would be presented close to each other, with two $L$ values, with continuous curves. Also, perhaps these figures could be presented earlier. Now Section 1. has Figs. 1 and 2, and they are very informative. But why not present the other data here, before the discussion of the complicated Bethe Ansatz solution? It is always useful to have the simple physical picture in mind, before going to the technical details.

Requested changes

1. Discuss a bit more the nature of the TDL, and the alternating boundary field.
2. Discuss a bit more clearly that the level avoidance becomes a level crossing eventually. The physical insights should be added to the Conclusions too.
3. Reposition some of the figures somehow, and add a bit more numerical data, for better visibility and understanding.
4. A typo: page 2 has ''two decoupled Majoranas fermions''

---

## Round 3 · Author Response

We thank the referees for their careful reading of the manuscript and their interesting comments. We have indeed implemented the two requested changes suggested in report 2. Concerning the many insightful comments of report 1, they deserve some more explanations.

  • The sentence «  the boundary magnetization becomes a discontinuous function of the other field at h− = h+, point at which the boundary root jumps from one edge of the chain to the other » was supposed to be an image describing the significant change in the mathematical description of the ground state (defined as the lowest energy state in finite volume) when crossing this point: the boundary root changes localization in its mathematical description as a solution of the Bethe equation (from being exponentially close to a zero associated to the boundary parameter at one end of the chain, it becomes exponentially close to the zero associated to the boundary parameter at the other end of the chain, as it becomes clear from our study of the ground state in section 4, and this also corresponds to a change of localization of the corresponding contribution to the wave function), and this change of localization is in fact responsible for the discontinuity that we observe here. This is not really due to a crossing of level since the degeneration is not exact at h_-=h_+ for L even (so there is no crossing of level for L finite in this case, see Figure 3 that we have slightly modified to make this point more clear). This change of localization of the boundary root with respect to the two zeroes of the boundary factor corresponds of course to a rearrangement of all the Bethe roots, but the latter is continuous when crossing this point (since the two zeroes coincide at this point). Nevertheless, since it seems that this sentence can be misunderstood, we have slightly modified it, as well as a few sentences in the paragraph before, so as to make it more precise.

  • "It is difficult to imagine that any measurement would detect the discontinuity investigated by the authors ». We are not expert in experimental measurements and, of course, since the discontinuity that we computed for the boundary magnetization may be smoothened by the temperature, it may not be easy to measure it experimentally (and we never pretended it should be). However, it seems to us that, in order to cool down the system to one of its ground state, it is sufficient to choose one of the magnetic fields at the edges to be larger (or smaller) than the other and then cool down the chain. This way the system is gapped, there is no degeneracy and the ground state 1 (or 2) has a finite energy difference with the other. We agree that this may be difficult to do in practice when h_- ~ h_+ (the temperature should be kept smaller than |h_- - h_+|) but in principle the protocol is clear.

  • Concerning the discussion about finite temperature, we hope it is clear that we do not consider the temperature case in the present paper. This is indeed an interesting problem that we mention as a possible further development in the conclusion. In particular, our study of the boundary magnetization in the ground state is performed strictly at zero temperature. Therefore, we do not pretend to correct anything about references [50,51] (which are now [53,54] in the new version): indeed, these references should in fact not appear in the footnote on page 28, and we thank the referee for pointing out this misprint, we have corrected this. We mentioned these references only by completeness since the same quantity was computed there in the temperature case. In particular, in [51] (now [53]), the zero-temperature limit of the boundary magnetization has been recovered (in the case of a zero magnetic field at the other end).

  • Concerning the question of whether the thermodynamic and the zero-temperature limits commute or not, it is difficult to answer this question without having investigated the temperature case. Nevertheless, the authors of [51] (now [53]) have investigated the zero-temperature limit of the expression they have obtained for the boundary magnetization at finite-temperature by taking the thermodynamic limit first, and they have recovered the result (including the discontinuity) that had been previously obtained at zero-temperature for the case of a zero-boundary magnetic field at infinity. This seems to indicate that, at least in the case considered in [51], these limits do commute.

  • « What happens here when we change only one of the fields, and how does it relate to the size of the system, and the order of limits ? » We thank the referee for this very good question. What we have found is that the form factor of the sigma_1^z operator between the two ground states decays exponentially in system size whenever the two fields are chosen to be different. Therefore in order to have a finite t->infty limit for the auto-correlation one should take first h_- -> h+ and then the thermodynamic limit. We have commented this in the manuscript, and added the computation of the form factor for different fields.

  • We focused on the even length case because it is the only case where there is an interesting quasi (and not exact) degeneracy at h_-=h_+. In the odd length case, there is no degeneracy at h_-=h_+ unless both are zero. There is however an exact degeneracy for h_-= - h_+, which is also present for finite L and can be seen as a consequence of the spin flip symmetry of the chain, and the two degenerate ground Bethe states are in this case in different magnetization sectors. It is obvious that in this case the discontinuity of the boundary magnetization is due to a crossing of levels of these two ground Bethe states with different magnetization, and therefore this case seemed to us a priori not so interesting from the physical point of view. It is nevertheless interesting to compare what happens in the even and odd length cases, since the appearance of the discontinuity is due to very different microscopic mechanisms. Hence, as suggested by the referee, we have added a section devoted to the odd L case, in which we redo the computation in this case and explain the differences with respect to the even L case. We have also commented this case in introduction and in conclusion.

  • Finally, we thank the referee for pointing out the misprint in formula (1.8). Indeed in the bulk case, the spontaneous staggered magnetisation can be observed either by applying a small staggered field or a small field on one of the sites (or at the boundary). We have nevertheless preferred to remove the mention to this quantity, since finally we do not find it so significant for our study. We have preferred to add instead a small comment about the odd L case that we now also study in our paper.

Resubmission 1901.10932v4 on 8 July 2019

You are currently on this page

Resubmission 1901.10932v3 on 27 May 2019

---

## Round 4 · Author Response

We thank the referee for his/her comments. However we mostly disagree on the suggestions that he/she formulated in order to improve our manuscript. In the following we reply to each comment.

-"The authors added a complete new discussion about the odd length chain. I think this is a great improvement to the paper. Quite interestingly, footnote 9 explains that the thermodynamic limit should be taken with a boundary magnetic field which changes sign between odd and even length chains. Would the authors agree that this is a consequence of the anti-ferromagnetic nature of the chain? Can this be seen perhaps in the Ising limit?"

We have added an extra comment in footnote 9 stressing that indeed this is due to the antiferromagnetic nature of the chain and that it can be also easily understood in the Ising limit.

-"The authors are explaining very clearly that the odd length case is different, because there is a level crossing even at finite
L. On the other hand, in the even length case there is a level avoidance, and the numerics is shown on the right of Fig. 3. However, from this figure it seems to me that in the L to infinity
limit we would indeed get a level crossing, because clearly the gap has to smoothen out, it becomes exponentially small. I think that this should be mentioned: based on the
L to infinity values the odd and even length cases are not so much different, given that the magnetic field is changed alternatingly. This is also a nice physical picture, that we don't really get different behaviour for an infinitely long chain, apart from the issue of the anti-ferromagnetic ordering."

We believe we have substantially explained the different nature of odd or even L. The even L case displays a degeneracy that closes exponentially with the system size while in the L odd case the degeneracy is exact due to Z_2 symmetry. We have stressed that we indeed recover the same behaviour in the thermodynamic limit for the quantities we have computed. This is also commented in the conclusion. We do not see reasons to stress this more.

-"Fig 3. is very informative and useful. Nevertheless I would like to see a finite volume data of the GS boundary magnetization too. This is presented in Fig 15, but there the finite volume data is only used to confirm the infinite volume predictions. Or at least it seems to me that this is the role. I think that it would be much more pedagogical if some finite volume data for
would be added already around Fig 3. It would be nice to see data from two
values, perhaps the
and some smaller value. Such that we could see the closing of the gap like on the right of Fig 3 now, and the appearance of the discontinuity like on Fig 15. I stress again that these figures are already quite good and informative, but the understanding would be much better if they would be presented close to each other, with two
L values, with continuous curves. Also, perhaps these figures could be presented earlier. Now Section 1. has Figs. 1 and 2, and they are very informative. But why not present the other data here, before the discussion of the complicated Bethe Ansatz solution? It is always useful to have the simple physical picture in mind, before going to the technical details."

We have added a comment on the relation between the closing of the gap and the discontinuity in the spontaneous magnetisation in the caption of figure 11, where the boundary magnetization for L even is numerically plotted at finite size versus its analytical value in the thermodynamic limit. This comment refers directly to Fig. 3 so that the reader can more easily make the connection. We believe that our manuscript has already many figures and all the aspects are covered, we do not see a reason to add extra figures. The boundary magnetisation is plotted in figure 11 and figure 15 and we believe this figures are enough for our publication. We moreover think that the position of the figures is adequate in the text to illustrate our analytical results. We hope that the referee will understand that further modifications will go against the format we wish to have for our manuscript.

---

## Editorial Decision

published